# Modelling the Inorganic Bromine Partitioning in the Tropical Tropopause Layer over the Eastern and Western Pacific Ocean.

Maria A. Navarro[1], Alfonso Saiz-Lopez[2], Carlos A. Cuevas[2], Rafael P. Fernandez[3], Elliot Atlas[1], Xavier Rodriguez-Lloveras[2], Douglas Kinnison[4], Jean-Francois Lamarque[4], Simone Tilmes[4], Troy Thornberry[5,6], Andrew Rollins[5,6], James W. Elkins[5], Eric J. Hintsa[5,6], and Fred L. Moore[5,6]

[1] Department of Atmospheric Sciences, RSMAS, University of Miami, Miami, Florida, USA
[2] Department of Atmospheric Chemistry and Climate, Institute of Physical Chemistry Rocasolano, CSIC, Madrid, Spain
[3] National Research Council (CONICET), FCEN-UNCuyo, UTN-FRM, Mendoza, Argentina
[4] Atmospheric Chemistry Observation & Modeling Laboratory, National Center for Atmospheric Research, Boulder, Colorado, USA
[5] National Oceanic & Atmospheric Administration, Earth System Research Laboratory, Boulder, Colorado, USA
[6] Cooperative Institute for Research in Environmental Science, University of Colorado, Boulder, Colorado, USA

*Correspondence to*: Maria A. Navarro (mnavarro@rsmas.miami.edu)

**Abstract.** The stratospheric inorganic bromine burden ($Br_y$) arising from the degradation of brominated very short-lived organic substances ($VSL_{org}$), and its partitioning between reactive and reservoir species, is needed for a comprehensive assessment of the ozone depletion potential of brominated trace gases. Here we present modelled inorganic bromine abundances over the Pacific tropical tropopause based on aircraft observations of $VSL_{org}$ of two campaigns of the Airborne Tropical TRopopause EXperiment (ATTREX 2013 carried out over the eastern Pacific and ATTREX 2014 carried out over the western Pacific) and chemistry-climate simulations (along ATTREX flight tracks) using the specific meteorology prevailing. Using the Community Atmosphere Model with Chemistry (CAM-Chem), we model that BrO and Br are the daytime dominant species. Integrated across all ATTREX flights, BrO represents ~ 43 % and 48 % of daytime $Br_y$ abundance at 17 km over the western and eastern Pacific, respectively. The results also show zones where Br/BrO >1 depending on the solar zenith angle (SZA), ozone concentration and temperature. On the other hand, BrCl and $BrONO_2$ were found to be the dominant night-time species with ~ 61% and 56 % of abundance at 17 km over the western and eastern Pacific, respectively. The western-to-eastern differences in the partitioning of inorganic bromine are explained by different abundances of ozone ($O_3$), nitrogen dioxide ($NO_2$), total inorganic chlorine ($Cl_y$) and the efficiency of heterogeneous reactions of bromine reservoirs (mostly BrONO2 and HBr) occurring on ice-crystals.

## 1 Introduction

The role of bromine in stratospheric ozone depletion has been discussed in several studies (Brinckmann et al., 2012;Daniel et al., 1999;Fernandez et al., 2017;Hossaini et al., 2015;Prather and Watson, 1990;Salawitch et al., 2005;Sinnhuber et al., 2009;Wofsy et al., 1975). Many of these discuss the contribution of brominated very short-lived organic substances ($VSL_{org}$)

like bromoform ($CHBr_3$), dibromomethane ($CH_2Br_2$), bromochloromethane ($CH_2BrCl$), dibromochloromethane ($CHBr_2Cl$) and bromodichloromethane ($CHBrCl_2$), in addition to long-lived halons and methyl bromide, as an important source of stratospheric bromine. The reaction mechanisms of $VSL_{org}$ that lead to the formation of inorganic bromine ($Br_y$) involve complex sets of reactions that have been described in previous modelling studies (Krysztofiak et al., 2012;Ordóñez et al.,

2012;Hossaini et al., 2010). The chemistry is initiated mainly via reaction with OH and by photolysis and leads to the formation of several unstable organic halogenated radicals. The fate of these radicals, and their mechanisms of reaction, are controlled by $NO_x$ conditions and OH levels. In polluted environments (high $NO_x$ regime) the radicals react to produce $NO_2$ or halogenated nitrates that can decompose, experience oxidation or photolysis, washout or react on surfaces. In "clean" environments (low $NO_x$ regime) the radicals undergo a series of cross reactions (including reaction with $HO_2$) leading to the

formation of several different products that can continue reacting with OH (or Cl), washout, or photo-dissociate to form $Br_y$ species as the end product (Krysztofiak et al., 2012).

The implementation of the complex chemistry of very short-lived species in current models like the Community Atmospheric Model with Chemistry (CAM-Chem) has been simplified by assuming that $Br_y$ is immediately formed from photo-oxidised $VSL_{org}$ (Ordóñez et al., 2012). However, the challenge to simulate $VSL_{org}$ observations and to quantify their degradation

products ($Br_y = Br + BrO + HOBr + BrONO_2 + HBr + BrCl + 2Br_2 + BrNO_2$) relies on other atmospheric processes that could modify this chemistry. For example, the location and timing of emissions, the transport dynamics and dehydration processes in the tropical tropopause layer (TTL) (Liang et al., 2010), and the occurrence of heterogeneous recycling reactions on sea-salt aerosol and ice-crystals (Fernandez et al., 2014) affect the total amount of inorganic bromine that can be injected to the stratosphere. A recent study by Navarro et al. (2015), which examined the TTL over the Pacific Ocean, showed the impact of

strong convective events on the chemistry of brominated species. The estimates of the contribution of very short-lived substances to total stratospheric bromine over the tropical eastern and western Pacific, showed a similar amount of $VSL_{org}$ over both regions. Nevertheless, a smaller amount of $Br_y$ was observed over the western Pacific due to the influence of the South Pacific Convection Zone (SPCZ) and the path of typhoon Faxai in 2014.

As tropical circulation, assisted by convection, could enhance the vertical transport of $VSL_{org}$, the $VSL_{org}$ degradation products

could reflect changes in their abundances and chemical speciation. A previous study by Fernandez et al. (2014) calculated that $VSL_{org}$ provided an annual average tropical stratospheric injection of total bromine of approximately 5 ppt (parts per trillion), where ~ 3 ppt entered the stratosphere as a product gas, while ~2 ppt entered as a source gas. In the Fernandez et al.(2014) study, the distribution of $Br_y$ species showed Br and BrO to be the dominant inorganic species during daytime, particularly over the tropical western Pacific, which they suggested was a hot spot with increased stratospheric bromine

injection (~3.8 ppt $Br_y$ and ~3.8 ppt $VSL_{org}$). This study also introduced the concept of the "tropical ring of atomic bromine", a photochemical phenomenon that extends in the tropics from approximately 15 to 19 km where the abundance of Br atoms is favoured due to low temperatures (<200K) and low $O_3$ abundances (<100 ppb). Similarly, the estimates of stratospheric $Br_y$ inferred from measurements of BrO during the NASA- Airborne Tropical TRopopause EXperiment (ATTREX) deployment over the eastern Pacific, showed approximately 3 to 5 ppt $Br_y$ for potential temperatures between 350 and 400 K in the TTL

(Werner et al., 2017). These previous studies, as well as many others (e.g., Dorf et al., 2008; Brinckmann et al., 2012;Liang et al., 2010, etc.), highlighted the importance of $Br_y$ product gas transportation from the lower troposphere into the TTL and lower stratosphere as well as its impact on ozone.

Our study focuses mainly on the difference in modelled $Br_y$ concentrations in the TTL over the Pacific throughout the ATTREX campaign flight tracks, and examines temporal and spatial distributions of $Br_y$. Based on the reliable representation of the observed $VSL_{org}$ by the CAM-Chem model (Navarro et al., 2015), and to further investigate the chemistry of bromine tracers in the TTL, we estimate the partitioning of $Br_y$ over the tropical eastern and western Pacific during 2013 and 2014, respectively. From this case study analysis, we also complement the study of the diurnal $Br_y$ speciation in the TTL, and the Br/BrO ratio distribution in the upper troposphere-lower stratosphere (UTLS) modelled by Fernandez et al. (2014) and Saiz-Lopez and Fernandez, (2016). This paper is organized as follows: Section 2 briefly describes the campaign, the methods used for the observations of trace gases, and the characteristics of the model simulations. Section 3 discusses the major findings regarding the amount of $Br_y$ and its partitioning within the TTL, as well as its relevance for the formation of the proposed tropical ring of atomic bromine (Fernandez et al., 2014;Saiz-Lopez and Fernandez, 2016). In addition, Section 3 discusses the results of a sensitivity test in the model where water-ice aerosol reactions are deactivated. Section 4 summarizes and concludes this study.

## 2 Methods:

### 2 .1 Observations

### 2 .1. 1 ATTREX Campaign

ATTREX, an airborne campaign focused on the chemical and physical processes in the TTL (~13-18 km), took place over the Pacific Ocean during boreal winter (Jensen et al., 2015). In 2013, six flights were conducted over the eastern and central Pacific, targeting the area between 187 and 268 °E and 11 °S to 34 °N (hereafter referred to as the eastern Pacific (EP)). In 2014, eight flights were carried out over the western Pacific, covering the area between 120 and 165 °E and 11 °S to 35 °N (hereafter referred to as the western Pacific (WP)). On board the NASA Global Hawk, the University of Miami deployed the Global Hawk Whole Air Sampler (GWAS) to collect more than 900 samples at different locations along the flight tracks. From these measurements, 436 observations along the flight track (388 during the day and 48 at night) over the WP and 309 flight points (152 during the day and 157 at night) over the EP were used to simulate $Br_y$ partitioning (Fig 1). Due to the logistics of the missions, the distribution of sample points according to solar zenith angle (SZA) was different between WP and EP. During ATTREX 2014 (WP), flights occurred mostly during daylight hours, as the plane took off at early local morning (~17:00 UTC, 3:00 am local time). Also, flights lasted approximately 20 hours which reduced the amount of night-time samples, particularly at high SZA. The flight tracks were oriented in different directions, mostly west-east, north-south, and even a local circle flight, following a different path from and to Andersen Air Force Base (13.5 °N, 144.9 °E). For these reasons, samples showed a high

density of observations at 50° SZA, and no samples beyond 130° SZA for WP. In contrast, during ATTREX 2013 (EP), flights also took off during early morning (~15:00 UTC, 7:00 am local time), but lasted between 22 to 24 hours. In addition, the flight tracks were oriented in southern and south-western direction and returned to Edwards Air Force Base (34.9 °N, 243.8 °E) over the same path. Hence, for the EP, the number of samples was evenly distributed along the entire range of SZAs.

**2 .1. 2 VSL$_{org}$ Observations**

Measurements of VSL$_{org}$, were carried out with the GWAS during the deployments and were used here to evaluate the model. Detailed methodology and implementation of this instrument is described in Navarro et al. (2015). Briefly, the GWAS contained 90 custom-made, 1.3 L stainless steel canisters with electrically operated solenoid valves. Two metal bellows pumps (Senior Aerospace) flow ambient air through a custom inlet at 2 to 8 liters per minute depending on the altitude. The instrument
is fully automated and controlled from the ground through an Ethernet interface. During the ATTREX mission, samples were collected along the Global Hawk flight track at different altitudes by closing the exhaust valves (Parker-Hannifin) and opening the solenoid valve (Parker-Hannifin) of the selected sample canister. When the desired pressure was reached the canister valve was closed and the manifold continued flushing. After sample collection, canisters were analysed using a high-performance gas chromatograph (Agilent Technology 7890A) and mass spectrometer with mass selective, flame ionization and electron
capture detectors (Agilent Technology 5975C).

The VSL$_{org}$ bromine budget and the vertical distribution of these species were presented in Navarro et al. (2015), which also included estimates of the organic *vs.* inorganic bromine fraction over the EP and WP. At the tropopause level (~17 km), the modelled estimates of the organic bromine fractions derived from very short-lived species were similar for the entire Pacific (3.84 ± 0.64 and 3.18 ± 1.49 ppt from WP and EP, respectively). However, the inorganic fraction inferred from VSL$_{org}$
measurements showed 3.02 ± 1.90 ppt of Br$_y$ over the EP and 1.97 ± 0.21 ppt over WP (Navarro et al., 2015). Based on these results, we performed model simulations to look into the variability and distribution of the Br$_y$ partitioning, and used ozone concentrations measured with the National Oceanic & Atmospheric Administration's Ozone system (NOAA-O$_3$) during ATTREX 2013, and the Unmanned aircraft system Chromatograph for Atmospheric Trace Species (UCATS-O$_3$) during ATTREX 2014 to also evaluate the model.

**2 .1. 2 O$_3$ Observations**

Measurements of ozone were carried out with dual-beam UV photometers. These instruments utilize two identical absorption cells, a mercury lamp, an ozone scrubbing catalyst, and two detectors that measure 253.7-nm UV radiation transmitted through the absorption cell. At this wavelength, the ozone absorption cross-section is well known, and thus the ozone number density can be readily calculated by Beer's Law. Since the two absorption cells are identical, virtually continuous measurements of
ozone are made by alternating the ambient air sample and ozone scrubbed sample between the two of them. At a fast collection

rate (from 2 Hz at < 200hPa to 0.5 Hz at 500hPa for NOAA- $O_3$ and 0.2 Hz for UCATS-$O_3$), the minimum detectable concentration of ozone corresponds to 1 ppb or less at STP. Based on an intercomparison between the two instruments during ATTREX 2013, the UCATS-$O_3$ instrument was 4.8±0.8 ppb higher than NOAA-$O_3$ in the regions of interest between 50 and 250 ppb, and approached zero near 500 ppb. The NOAA-$O_3$ instrument was unavailable on ATTREX-2014.  To merge the

measurements taken over different time scales, these high-rate measurements of ozone were averaged to match the sample collection times of each GWAS sample (~30 – 90 sec) (Blake et al., 1997;Blake et al., 2001;Blake et al., 2003;Blake et al., 1999;Blake et al., 2004;Schroeder et al., 2014), and then the merged data  were compared to CAM-Chem outputs (Kormann et al., 2003;Olson et al., 2012).

## 2 .2 Modelling: CAM-Chem configuration

The estimates of VSL$_{org}$, $O_3$ and the Br$_y$ partitioning were carried out with the CAM-Chem model, a 3-D chemistry climate model included into the CESM framework (Community Earth System Model, version 1.1.1) (Lamarque et al., 2012). This model includes a complete photochemistry, wet and dry deposition, and heterogeneous chemistry on sea-salt aerosols and ice particles (Fernandez et al., 2014;Ordóñez et al., 2012). The current setup is based on the bromocarbon emission inventory of Ordoñez et al., (2012), which includes time-dependent geographically distributed sources of $CHBr_3$, CH2Br$_2$, $CH_2BrCl$,

$CHBr_2Cl$, $CHBrCl_2$ and $CH_2IBr$. We do not consider here chlorocarbon sources such as $CH_2Cl_2$ and $C_2Cl_4$ since those species live long enough to be injected almost entirely as source gases to the stratosphere and contribute little to the tropospheric inorganic chlorine (Cl$_y$) loading (Hossaini et al., 2015), though this situation may change based on current trends (Oram et al., 2017). Additional Br$_y$ and Cl$_y$ sources from sea-salt heterogeneous dehalogenation in the lower troposphere are parameterized (Ordóñez et al., 2012;Fernandez et al., 2014). Prescribed volume mixing ratios of long-lived chlorofluorocarbons (CFCs) and

halons at the surface, as well as surface concentrations of anthropogenic $CO_2$, $CH_4$, $N_2O$ and other ozone precursors, are based on the inventory of Meinshausen et al. (2011). Global emissions of important ozone precursors (NO$_x$, CO, VOCs) were obtained through a harmonization exercise of reactive emissions between years 2000 and 2005 for different RCP (Representative Concentration Pathways) scenarios (Meinshausen et al., 2011;Lamarque et al., 2011). It is worth noting that all inorganic halogen species (i.e., Cl$_y$ and Br$_y$) are not constrained but explicitly solved at each timestep.  Losses in CAM-

Chem are parameterized following a large-scale precipitation scavenging algorithm that includes a physical treatment of scavenging through improvements in the formulation of the removal in sub-grid-scale cloudy environments, and includes washout as well as ice phase uptake of soluble inorganic bromine species (each of them with an independent Henry's Law constant) within the water column (see Neu and Prather, 2012 and Fernandez et al., 2014 for details).

Model simulations were run in specified dynamics mode (SD) using meteorological fields prevailing at the time of the

campaigns. A spatial resolution of 1° (longitude) x 1° (latitude) with 56 vertical levels (from the surface to ~ 3.5 hPa) and a temporal resolution of 30 min were used. Model hourly output was sampled at exactly the same times and locations as the ATTREX measurements, without performing either spatial or temporal averaging on model grids. Once each independent

flight track was extracted from the model output, all atmospheric quantities were averaged into 1 km altitude bins, standard deviations were calculated, and the model was compared with measured data. The chemistry of $Br_y$ species between day and night was distinguished by disregarding the samples collected during twilight (total solar zenith angle between 80° and 100°). In addition, local estimates of ozone, nitrogen dioxide and inorganic chlorine concentrations were carried out over the EP and

WP along the flight tracks to understand the chemistry that leads to the specific bromine partitioning.

## 3 Results and discussion

The current modelling study was conducted as part of the work described by Navarro et al. (2015), and it follows the same methodology and statistical analysis for discrete variables. At the time of the model runs and analyses, only ozone and $VSL_{org}$ abundances were available to validate model performance, since BrO and $NO_2$ measurements from the ATTREX mission,

now published by Werner et al. (2017), were not available. Thus, once the model performance during ATTREX campaign is evaluated in Sect. 3.1, we proceed to the CAM-Chem modelling case study to determine the $Br_y$ partitioning (Sect. 3.2) and efficiency of heterogeneous recycling reactions (Sect- 3.3) on the mostly unexplored eastern and western Pacific TTL.

## 3. 1 CAM-Chem model evaluation

The first step of this study was to evaluate the CAM-Chem model chemistry and performance for our intended application.

For the evaluation, we used the NOAA-$O_3$, UCATS-$O_3$, and organic bromine species from GWAS measurements taken during the ATTREX campaign, since BrO and $NO_2$ measurements from the ATTREX mission were still under examination by the time of this analysis. Figure 2 shows the measured and model mixing ratios of $O_3$ in the UTLS for the WP and EP. The NOAA and UCATS $O_3$ mixing ratios averaged into 1 km altitude bins ranged from 46 ppb at 14 km to 166 ppb at 18km over the WP, and 46 ppb to 179 ppb from 14 to 18 km over EP. CAM-Chem simulations estimated 67 ppb to 196 ppb from 14 to 18 km

over WP, and 100 ppb at 14 km and 243 ppb at 18 km over EP. The model reproduces well the variability of measured ozone with altitude in both the WP and EP. However, the different convective processes are not accurately represented in both regions, which leads to an offset in the EP between measurements and simulated ozone values. The vertical profile of organic bromine species from GWAS measurements can be found in Navarro et al. (2015).

Figure 3 shows the correlation between average measurements and model outputs of the 1 km of altitude bins for $O_3$, $CHBr_3$

and $CH_2Br_2$ over the WP and EP, as well as the linear regression equations with and associated uncertainties in slopes and intercepts. Excellent correlation ($R^2 > 0.97$) between $O_3$ measurements and CAM-Chem estimates for both EP and WP were observed, with slopes not differing significantly from unity. However, small offsets (27.1 ppb in WP and 54.5 ppb in EP) reflect bias in the model. In addition, good agreement was observed between GWAS measurements and model simulations of $CHBr_3$ ($R^2 = 0.84$ for WP, $R^2 = 0.81$ for EP), and $CH_2Br_2$ ($R^2 = 0.95$ for WP, $R^2 = 0.90$ for EP), as shown in figure 3 and the

previous work of Navarro et al. (2015).

## 3. 2 $Br_y$ partitioning

The vertical distribution of inorganic species showed a slight variability with altitude (Fig 4). Over both the WP and the EP, BrO and Br are the most abundant species during daytime hours from 14 to 18 km. However, below 16 km, the amount of HBr present over the EP tends to be slightly larger than that of atomic Br. For the WP and the EP during night-time hours, BrCl and $BrONO_2$ are the dominant species, which is in agreement with Fernandez et al. (2014). Mixing ratios of BrCl closely compete with $BrONO_2$, particularly at 15 and 18 km over the WP. Over the EP, $BrONO_2$ dominates the $Br_y$ species over the entire range of altitude from 14 to 18 km at night. During daylight hours, the total $Br_y$ burden increases from 1.49 to 2.43 ppt between 14 km and 18 km over the WP and from 1.82 to 2.97 ppt over the EP within the same altitude range. Similarly, night-time $Br_y$ ranges from 1.40 to 2.27 ppt and 1.82 to 2.99 ppt for the WP and EP, respectively. This indicates that the total $Br_y$ atmospheric burden is equivalent within the diurnal cycle. Note that within the same altitude range, the model output for total $VSL_{org}$ decreases from 5.72 ppt at 14 km to 2.53 ppt at 18 km over the WP, and from 3.90 ppt at 14 km to 1.97 ppt at 18 km over the EP, which indicates the dominant role played by VSL photodecomposition in controlling the $Br_y$ loading in the tropical UTLS (Navarro et al., 2015). Our calculated mean vertical distributions for the EP are at the lower edge of the ranges described by Werner et al. (2017). They report a range for BrO between $0.5 \pm 0.5$ ppt at the bottom of the TTL to about 5 ppt at $\theta = 400$ K, consistent with an inferred increase of $Br_y$ from a mean of $2.63 \pm 1.04$ ppt to $5.11 \pm 1.57$ ppt.

At the tropopause level (~17 km) and integrated over all flights and SZAs, the inorganic partitioning showed ~ 43 % (0.79 ppt) of abundance of BrO during daylight and ~61 % of BrCl (0.94 ppt) during night-time over WP. On the other hand, 48 % (1.43 ppt) of $Br_y$ is presented as BrO during daylight and 56 % (1.41 ppt) as $BrONO_2$ at night-time over EP (Fig 4). Atomic bromine is the second most abundant species during the day, with mean daytime values of 0.64 ppt and 0.57 ppt for the WP and EP, respectively. During the night, $BrONO_2$ (0.29 ppt) and BrCl (0.34 ppt) are the $2^{nd}$ most abundant species over the WP and EP, respectively. It is worth noting that even when the maximum inorganic chlorine levels for individual flights are larger in the EP, BrCl is not the dominant night-time reservoir, while in the WP, where BrCl dominates, maximum $Cl_y$ mixing ratio is almost half the concentration predicted for the EP (see Table 1 and Figures 5 and 6). Maximum $Cl_y$ abundances averaged for all flights within each region show < 85 pptv in the WP and < 182 pptv for the EP (see Fig. 7), with a global mean tropical annual $Cl_y$ mixing ratio of 50 pptv in agreement with previous reports (Marcy et al., 2004; Fernandez et al., 2014;Hobe et al., 2011;Jurkat et al., 2014;Mébarki et al., 2010). This can be explained considering the faster vertical transport occurring in the WP region, which decreases the contribution from photochemical decomposition of VSL chlorocarbons (Saiz-Lopez and Fernandez, 2016). Note that within the TTL, HCl dominates the $Cl_y$ partitioning, with modelled mixing ratios up to 1 order of magnitude larger than those found for HOCl and $ClONO_2$ (see Fig. 10 in Fernandez et al., 2014). Further knowledge of the complete partitioning between inorganic chlorine species is beyond the scope of this work.

In order to understand the chemistry that led to these modelled abundances, the concentrations of $Br_y$ products (Br, BrO, HOBr, $BrONO_2$, HBr, and BrCl), as well as modelled mixing ratios of the dominant reactants: $O_3$, $NO_2$, and $Cl_y$, were studied as a function of the SZA for the entire range of altitude (14 to 18 km). Figures 5 and 6 compare the $Br_y$ partitioning as well as

the modelled $O_3$, $NO_2$ and $Cl_y$ abundances along all flights in the EP and WP, respectively. Here, it can be clearly observed how the dominant species changes from BrO during daytime, to $BrONO_2$ or BrCl during night-time. Figure 7 shows the mean abundances for all species including all flights in the WP and EP. Even though the mean results do not simulate differences observed in each flight, they are representative and illustrative of the average state of the tropical upper atmosphere within the

EP and WP in the presence and absence of sunlight, and should provide relevant information about the dominant processes occurring in each region. During daylight, the high average $O_3$ (up to ~ 190 ppb), compared to average $Cl_y$ and $NO_2$ ($Cl_y$ up to ~ 84 ppt, $NO_2$ up to ~ 33 ppt) (Fig 7), led to the rapid formation of BrO over the WP (Fig 7a). As the SZA increases, a decrease of photolysis (particularly at SZA ~ 100-120°, Fig 7b) allows the heterogeneous reaction of inorganic chlorine (mostly HCl) and bromine reservoirs (HOBr and $BrONO_2$) to increase the production of BrCl during the night. Note that the accumulation

of night-time BrCl is more evident during ATTREX 2014 due to larger ice-crystal surface areas and lower $NO_x$ levels prevailing over the WP (Fig 7a).

The scenario over the EP is slightly different as levels of $NO_2$ and $O_3$ are larger, while the Surface Area Density of ice-crystals (SAD-ICE) is reduced (see Section 3.3). A statistical analysis of CAM-Chem $NO_2$ during daylight over the EP is presented in Fig 8.  Our average range of $NO_2$ mixing ratios is approximately $15 \pm 6$ ppt at 14 km, with slightly higher values over the

tropopause, $22 \pm 24$ ppt at 17 km. These estimates are within 1 standard deviation agreement with the $NO_2$ values presented by Stutz et al., 2017 and Werner et al., 2017  from observations made during ATTREX 2013 over the EP. Their $O_3$ scaling technique allowed retrieval of $NO_2$ concentrations of $15 \pm 15$ ppt in the TTL, with excursions of 70 - 170 ppt in the mid-latitude lower stratosphere associated with older stratospheric air intrusions. However, previous studies have shown large associated uncertainties in $NO_2$ measurements based on remote sensing instruments, which also depends on the individual

observation geometries and instrument operation times (e.g. 30% of total relative error of $NO_2$ measurements below 25Km (Weidner et al., 2005), and 50 % for satellite measurements from SAGE II  bellow 25 km (Bauer et al., 2012)).

Hence, the EP daytime average concentrations of ozone (up to ~ 300 ppb), $Cl_y$ (max $Cl_y$ ~ 181 ppt) and $NO_2$ (max $NO_2$ ~ 48 ppt) are almost twice as high as those over the WP (Fig 7d), while SAD-ICE levels are up to 1 order of magnitude smaller (see Fernandez et al., 2014). Higher concentrations of ozone were associated with enhanced production of BrO (Fig 7c). Meanwhile,

during dark hours, the higher $NO_2$ concentrations and the slower rate of heterogeneous reactions of bromine reservoirs (see Section 3.3) lead to the formation of $BrONO_2$ and the reduction of BrCl levels over the EP (Fig 7c). These results are in good agreement with the partitioning of $Br_y$ found by Werner et al. (2017) where BrO is the dominant daylight species over EP, and the estimates of Fernandez et al. (2014), which suggested BrO and $BrONO_2$ as the dominant species in the TTL over the entire tropics during daytime and night-time, respectively.

Note that the predicted differences in $Cl_y$ abundance can reach factors as much as 5 times larger for the EP if individual flights are considered (e.g., max. Cly ~500 ppt for RF01, RF03 and RF04 performed in the EP during ATTREX-2013, while max. $Cl_y$ for all flights except RF07 (< 400 pptv) remain below 100 ppt). However, the night-time BrCl abundance is still larger in the WP, representing more than 90% of the night-time $Br_y$ partitioning for flights RF02 and RF04 (see Figure 5). For these cases, BrCl mixing ratios between 1 and 2 pptv are modelled within air-parcels with a very low $Cl_y$ abundance (of the order of

6 ppt). In order to understand this unexpected behaviour, we performed a sensitivity simulation neglecting the inter-halogen heterogeneous recycling occurring on upper tropospheric ice-crystals, see Sect. 3.3 below.

### 3. 2. 1 Tropical ring of atomic Br: indications from this case study

Our results also indicate that, integrated over all SZAs, the second most abundant species over both EP and WP during daytime is atomic Br. This species plays a fundamental role in the formation of the proposed inhomogeneous tropical ring of atomic Br, a natural atmospheric phenomenon that comes from the rapid photochemical equilibrium between BrO and Br under conditions of low $O_3$ and temperatures (Fernandez et al., 2014;Saiz-Lopez and Fernandez, 2016). According to our simulations, during daytime hours, the average mixing ratios of Br remain below 0.75 ppt over the EP and WP along the entire range of altitudes (i.e., between 14 to 18 km) (Fig 4). However, modelled atomic Br abundances surpass BrO mixing ratios at low SZA (close to noontime) and low ozone abundances (below 100 ppb, Fig. 7b and 7d). These observations contrast with the variability observed in BrO, which varies from 0.61 ppt to 1.19 ppt over the WP and from 0.72 ppt to 1.43 ppt over the EP (Fig 4), depending on $O_3$ and $NO_2$ background levels. Also, Br shows a smooth variation during the day, slightly decreasing its abundance as the SZA increases, while the temporal evolution of BrO is more variable, mostly under the higher $NO_x$ levels prevailing in the EP (Fig. 7). A closer inspection of each separate flight (Figs. 5 and 6) reveals the large inhomogeneity of the tropical rings of atomic bromine. In the EP, modelled Br surpasses BrO mixing ratios at 60º SZA for flights RF04 and RF06, but as the remaining flights sampled larger BrO mixing ratios, the mean EP abundances shown in Fig. 7c shows Br/BrO > 1 only at 20º SZA. Similarly, the mean results shown in Fig. 7a for the WP show BrO > Br at all times, but RF02 and RF03 show the ratio Br/BrO is larger than one at 50º SZA. This highlights the importance of considering non-averaged (both spatially and temporally) model output to determine the concentration of photochemical reactive species or other atmospheric quantities such as the Br/BrO ratio.

In contrast to the study of Werner et al., 2017, which focused on ATTREX measurements taken over the EP and used an $O_3$-scaling technique to retrieve their results, our model calculations support the prediction that the Br/BrO ratio could become larger than the unity, particularly in the tropical UT and TTL of both the eastern and western Pacific Ocean during daylight hours, particularly at low SZA. This enhancement of Br atoms in the tropical tropopause layer has also been identified in other studies and seems to be a consistent feature in global models including a complete treatment of halogen chemistry in the troposphere (Chen et al., 2016;Holmes et al., 2006;Schmidt et al., 2016). Nevertheless, the combination of ozone concentrations and temperatures plays a fundamental role in the distribution of both species. Thus it is expected to find a spatially irregular pattern in the conditions that favour Br/BrO > 1.

Figure 9 shows the distribution of the Br/BrO ratio over the WP and EP, and its correlation with ozone concentrations and temperatures. The results in the figure are based on the average 1-km binned data for all track flights, although equivalent conclusions can be drawn for each separate flight transect. Over the EP, Br/BrO > 1 is predicted in discrete air masses, particularly at SZA between 40° and local noon. Cold temperatures and low $O_3$ concentrations enhance the prevalence of

atomic Br for those times and locations. Indeed, Saiz-Lopez and Fernandez. (2016), determined that Br becomes the dominant species for $O_3 < 100$ ppb and $T < 200$ K. Br/BrO ratios lower than 1 are observed near the tropopause for SZA between 40° and 55° under higher concentrations of ozone and at warmer temperatures. The Br/BrO distribution over the WP seems to be more inhomogeneous with an overall higher Br/BrO ratio than over the EP. The lower ratio of Br/BrO in the EP compared to

WP could be due to the higher levels of $O_3$ modelled in this region (Fig. 2). In any case, the presence of the tropical rings of Br occurs as a patchy distribution of bromine atoms superimposed on a background BrO curtain. In general, the Br/BrO ratio peaks at 17 km, very close to the upper limit of the TTL, highlighting the importance of determining Br abundances in order to address the total amount of $Br_y$ injected to the stratosphere.

As the magnitude of the $Br_y$ reactive species (Br and BrO) depends on changes in the oceanic sources and vertical transport,

the Br/BrO ratio will vary according to season and geographical region. Hence, during strong convective events over the WP (e.g., air masses tracked from the SPCZ) the amount of Br and BrO remained similar to each other. These results are in good agreement with Fernandez et al. (2014) who suggested Br/BrO > 1 during strong convective periods over the WP warm pool region.

Reservoir species like HBr and HOBr were the third most dominant $Br_y$ components. These two species contribute to the

formation of BrCl through heterogeneous recycling reactions (see Ordóñez et al., 2012), particularly at night due to the absence of photolysis reactions and the presence of traces of chlorine. As suggested by Fernandez et al. (2014), higher abundances of $Cl_y$ could result from the decomposition of very short-lived chlorocarbons or the subsidence of HCl from the stratosphere to the TTL, driving the night-time chemistry in areas where $Cl_y > Br_y$.

### 3. 3 Heterogeneous reactions: impact of water-ice recycling on $Br_y$ speciation/distribution

Heterogeneous recycling reactions of reservoir species on ice-crystals are relevant at UTLS altitudes. Thus, a sensitivity test was carried out to determine the influence of water-ice aerosols on the partitioning of inorganic species. Equations (1) to (6) shows the chlorine, bromine and inter-halogen tropospheric heterogeneous reactions that occur on ice-crystals (for a complete description of the implementation of heterogeneous reactions in CAM-Chem, see Table S1 in supplementary online material of Fernandez et al., 2014. )

$$BrONO_2 \rightarrow HOBr + HNO_3 \qquad (1)$$
$$ClONO_2 \rightarrow HOCl + HNO_3 \qquad (2)$$
$$HOCl + HCl \rightarrow Cl_2 + H_2O \qquad (3)$$
$$HOCl + HBr \rightarrow BrCl + H_2O \qquad (4)$$
$$HOBr + HCl \rightarrow BrCl + H_2O \qquad (5)$$
$$HOBr + HBr \rightarrow Br_2 + H_2O \qquad (6)$$

Including heterogeneous reactions in the chemical mechanism changes the relative partitioning between $Br_y$ species, and consequently, the abundance of the dominant species that controls the effective removal is altered. Thus, turning on and off heterogeneous reactions will change the bromine sinks within the UT and TTL, as the relative efficiency of effective washout for each independent $Br_y$ species are different in the model (i.e., individual Henry's Laws constant are considered for each species). Figure 10 shows mixing ratios at different altitudes when water-ice aerosol reactions were deactivated simultaneously. Relative to the results from the complete mechanism (Fig. 4), at 17 km the absence of ice-crystal reactions increases the total inorganic fraction by 7% and 12% over the EP during the day and night, respectively. On the other hand, $Br_y$ increases by 29% and 40% over the WP during day and night, respectively. This relative increase of total $Br_y$ is mainly due to changes in the amount of HBr during the day and an enhancement of both HBr and $BrONO_2$ during the night. As BrCl is only produced by equations (4) and (5), it does not accumulate during night hours within the sensitivity study where heterogeneous reactions have been turned off (Fig. 10). The mixing ratios of all other species remain very similar as shown in Figure 4 and Figure 10. Our model results show that turning off heterogeneous reactions reduces the total amount of $Br_y$ washed out at 17 km by ~0.5 pptv and ~0.3 pptv for the WP and EP, respectively. This value is of the same magnitude but opposite direction to the results obtained by Aschmann et al., (2011). HBr is highly soluble and it would be expected that a relative increase in HBr partitioning would imply a more efficient washout. But, as explained by Aschmann et al. (2011), it is possible that a significant part of the adsorbed HBr at high altitudes can re-evaporate within the TTL (and eventually reach the stratosphere) before being washed out. This is because the removal process does not occur immediately and residence times are large in the TTL. Indeed, they found a local HBr maximum at around 17 km within an equivalent sensitivity simulation that neglected heterogeneous activation for HBr.

Another characteristic of the sensitivity test is the clear absence of BrCl during night-time as water-ice aerosol reactions of HOBr and HCl are suppressed. Compared to the base case, the absence of BrCl makes $BrONO_2$ the dominant species at night over the WP, with 53% at 17 km, compared to 61% BrCl in the base case. Thus, neglecting ice-recycling reactions (1) to (6) prevents the heterogeneous conversion of $BrONO_2$ to BrCl, and gas-phase bromine nitrate (which is formed mainly by the termolecular reaction of $BrO + NO_2 + M$ during twilight) remain as the dominant $Br_y$ species during the night for both EP and WP regions. But when the heterogeneous recycling reactions are activated, the model predicts that the recycling efficiency depends mostly on the total SAD-ICE prevailing in the upper troposphere. Even under very low $Cl_y$ concentrations (between 10 and 20 ppt), if SAD-ICE is present in the TTL, the night-time reservoir partitioning is shifted to BrCl. Fernandez et al., (2014) found tropospheric SAD-ICE levels within the western pacific upper TTL to be some of the largest within the tropics, suggesting that BrCl abundance should peak in this region of the Pacific. The smaller impact of turning off heterogeneous reactions in the EP can be explained by the less efficient inorganic bromine recycling occurring on the smaller SAD-ICE prevailing in this region. However, the heterogeneous chemistry of bromine species with water-ice aerosols still requires further research as their atmospheric surfaces are highly dynamic. Indeed, Hobe et al., (2011) suggested that the coupling of chlorine and nitrogen compounds in the tropical UTLS may not be completely understood, which would also impact on the bromine burden. The presence of cirrus ice clouds, at the cold temperatures of the equatorial and mid-latitude UTLS, facilitates

the conversion of bromine reservoir species to more photochemically active forms, which play an important role in the oxidative capacity of this region of the atmosphere.

## 4 Summary and Conclusions

Our estimates of $Br_y$ partitioning in the TTL over the Pacific Ocean showed that mostly BrO and to a lesser extent atomic Br are the dominant species during daytime hours, while BrCl and $BrONO_2$ are predicted to dominate the TTL $Br_y$ at night-time over the WP and EP. The difference in the partitioning of $Br_y$ during the diurnal cycle between the WP and the EP could be explained by the changes in the abundance of $O_3$, $NO_2$ and $Cl_y$ in these two regions of the Pacific, as well as by the efficiency of heterogeneous reactions that could modify this chemistry, mostly during the night. Table 1 summarizes the results found at 17 km.

Reactive species like atomic Br become the dominant $Br_y$ species in patchy regions of the Eastern and Western Pacific TTL during daylight, following the large inhomogeneity of ozone abundances within these regions strongly influenced by deep convection. The low ozone and cold conditions, in combination with the rapid photochemical equilibrium between BrO and Br, favour Br/BrO > 1 for patchy regions of the TTL and are consistent with previous results about the proposed tropical ring of atomic bromine. The CAM-Chem output along the ATTREX flights indicates that Br and BrO alternate as the dominant daytime species indicating a large inhomogeneity for the tropical ring of Br, mainly due to the large ozone/T variability of the air parcels within the convective tropical WP and EP. Improved field data for the identification and complete speciation of bromine species are needed to continue evaluating this hypothesis.

This model study contributes to the growing database of reactive halogen estimates based on halocarbon observations. The variable photodecomposition of $VSL_{org}$, the transport of inorganic degradation products from the lower troposphere into the TTL, as well as the efficiency of heterogeneous reactions involving ice aerosols, play an important role in the overall upper tropospheric $Br_y$ loading and the consequent stratospheric bromine injection. However, further research on the organic/inorganic bromine fraction, as well as its distribution between reactive and reservoir species, is needed in different areas of the globe and at different heights to reduce the uncertainty of the amount of $Br_y$ that enters the stratosphere, and to properly constraint the global bromine budget.

## Acknowledgments

This work was supported by NASA Grant NNX10AO83A S08 and NASA Atmospheric Composition Modelling and Analysis Program Activities (ACMAP), grant/cooperative agreement number NNX11AH90G.

We gratefully acknowledge the support of Eric Jensen and Lenny Pfister, principal investigators of the ATTREX campaign. We thanks the engineers, technicians and pilots of NASA Armstrong Flight Research Center, and NASA-ESPO Project Management, as well as S. Schauffler, R. Lueb, R. Hendershot and S. Gabbard for technical support in the field, and V. Donets, X. Zhu and L. Pope for GWAS data analysis. The National Center for Atmospheric Research (NCAR) is funded by the

National Science Foundation NSF. Computing resources (ark:/85065/d7wd3xhc) were provided by the Climate Simulation Laboratory at NCAR's Computational and Information Systems Laboratory (CISL), sponsored by the NSF and other agencies. The CESM project (which includes CAM-Chem) is supported by the NSF and the Office of Science (BER) of the US Department of Energy. RPF would like to thanks CONICET and FCEN-UNCuyo/UTN-FR Mendoza for financial support.

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

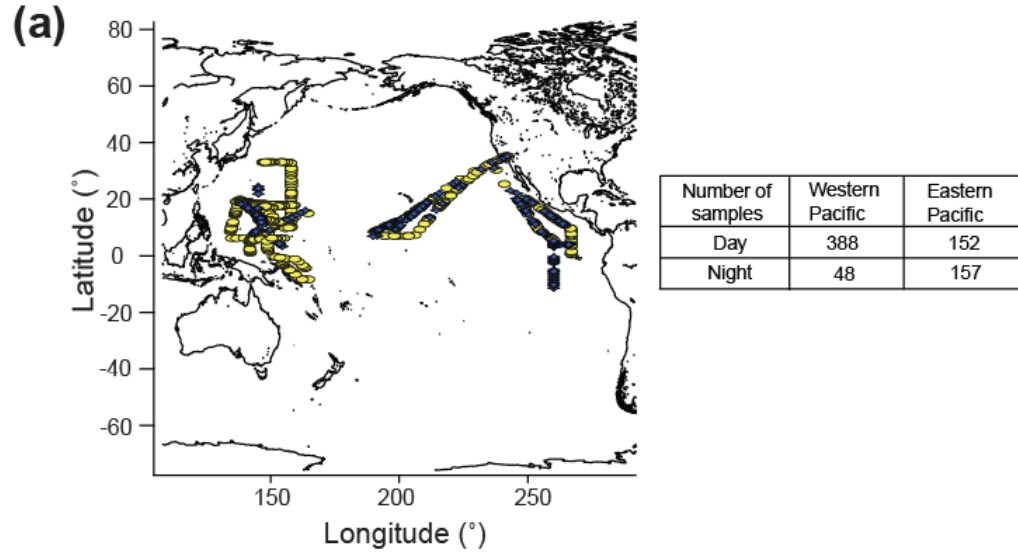

| Number of samples | Western Pacific | Eastern Pacific |
|---|---|---|
| Day | 388 | 152 |
| Night | 48 | 157 |

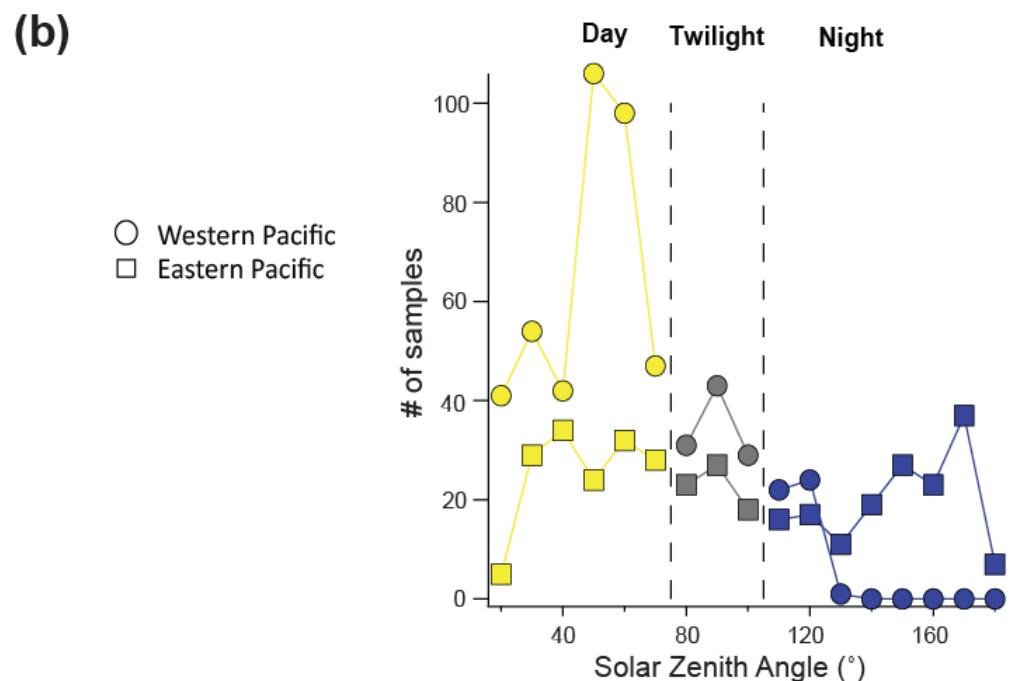

**Figure 1: (a) Location of GWAS observations along ATTREX flight tracks taken at different altitudes over WP and EP. Yellow filled circles represent the samples taken during the day and blue filled circles the samples taken at night. (b) Sample density of GWAS observations arranged by solar zenith angle over the western and eastern Pacific. Yellow filled circles represent samples taken over WP during daylight, while blue filled circles are the samples taken over WP during night-time. Yellow filled squares represent the samples taken over EP during daylight, while blue filled squares are the samples taken over EP during night-time. Grey circles and squares are the samples taken during twilight over WP and EP, respectively.**

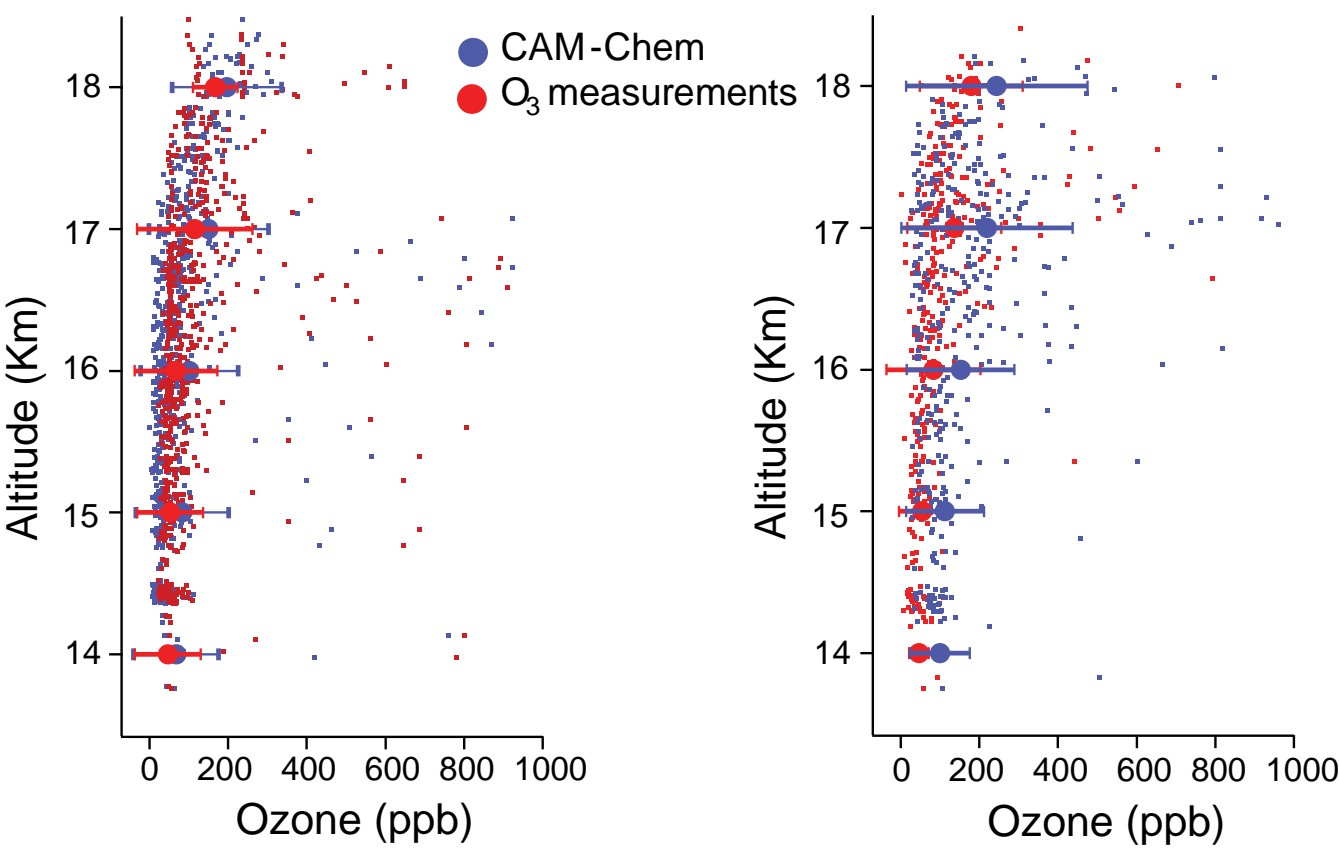

**Figure 2: NOAA-UCATS O₃ observations for all flights (blue dots) and CAM-Chem simulations along ATTREX flight tracks (red dots). Filled circles represent the mixing ratios averaged into 1 km altitude bins ± 1 sd over WP and EP to illustrate the altitude variation of ozone. Range of the bins are: 14 km= 13.5 to 14.5 km, 15 km= 14.5 to 15.5 km, 16 km= 15.5 to 16.5 km, 17 km= 16.5 to 17.5 km, 18 km= 17.5 to 18.5 km. Similar figures for CH₂Br₂ and CHBr₃ are in Navarro et al., (2015).**

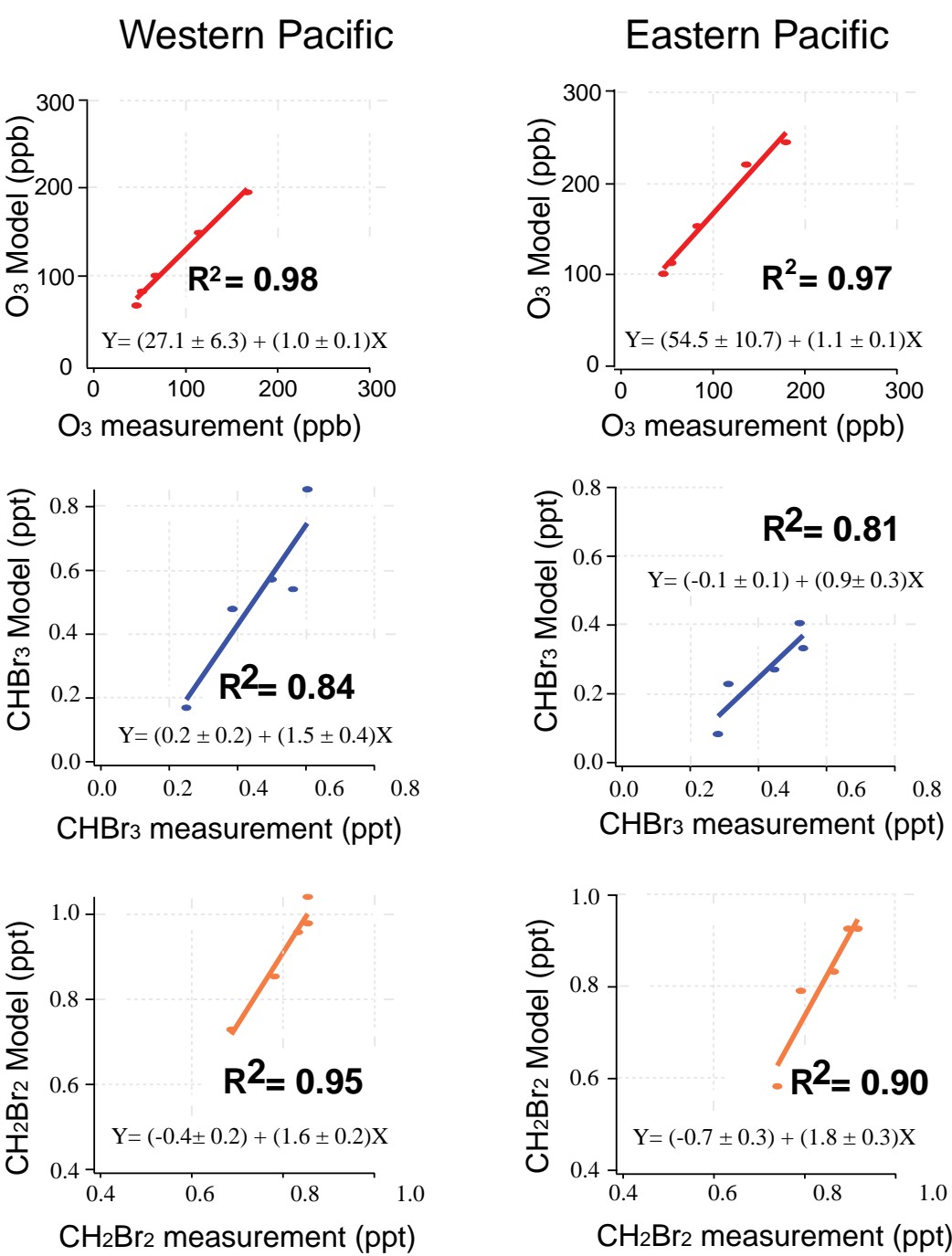

**Figure 3: Correlations between measured and modelled mixing ratios of $O_3$, $CHBr_3$ and $CH_2Br_2$ over the WP and EP, using the 1 km average bins. Filled circles are the mixing ratios averaged into 1 km altitude bins (same as in figure 1 for $O_3$, and figure 1 in Navarro et al., (2015) for $CHBr_3$ and $CH_2Br_2$). Solid lines represent the linear regression analysis of the 1 km average bins including coefficient of determination ($R^2$) and regression equation (Y).**

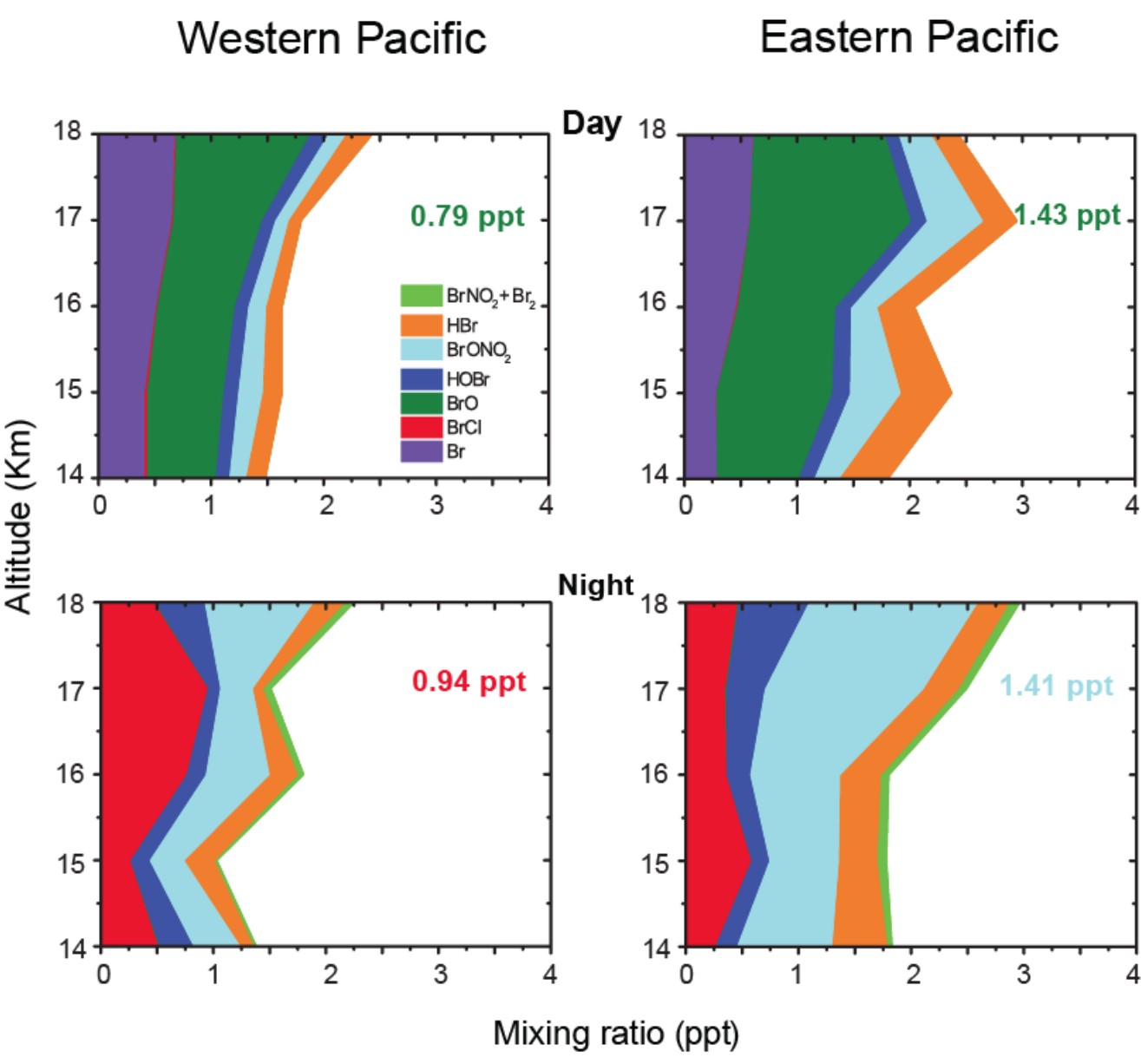

**Figure 4: Model calculations of inorganic bromine partitioning along ATTREX flight tracks over the WP and EP using CAM-Chem with prevailing atmospheric conditions (specified meteorology) during the ATTREX campaign. Numbers inside the boxes represent the mixing ratios of BrO (green), BrCl (red) and BrONO₂ (light blue) at 17 km.**

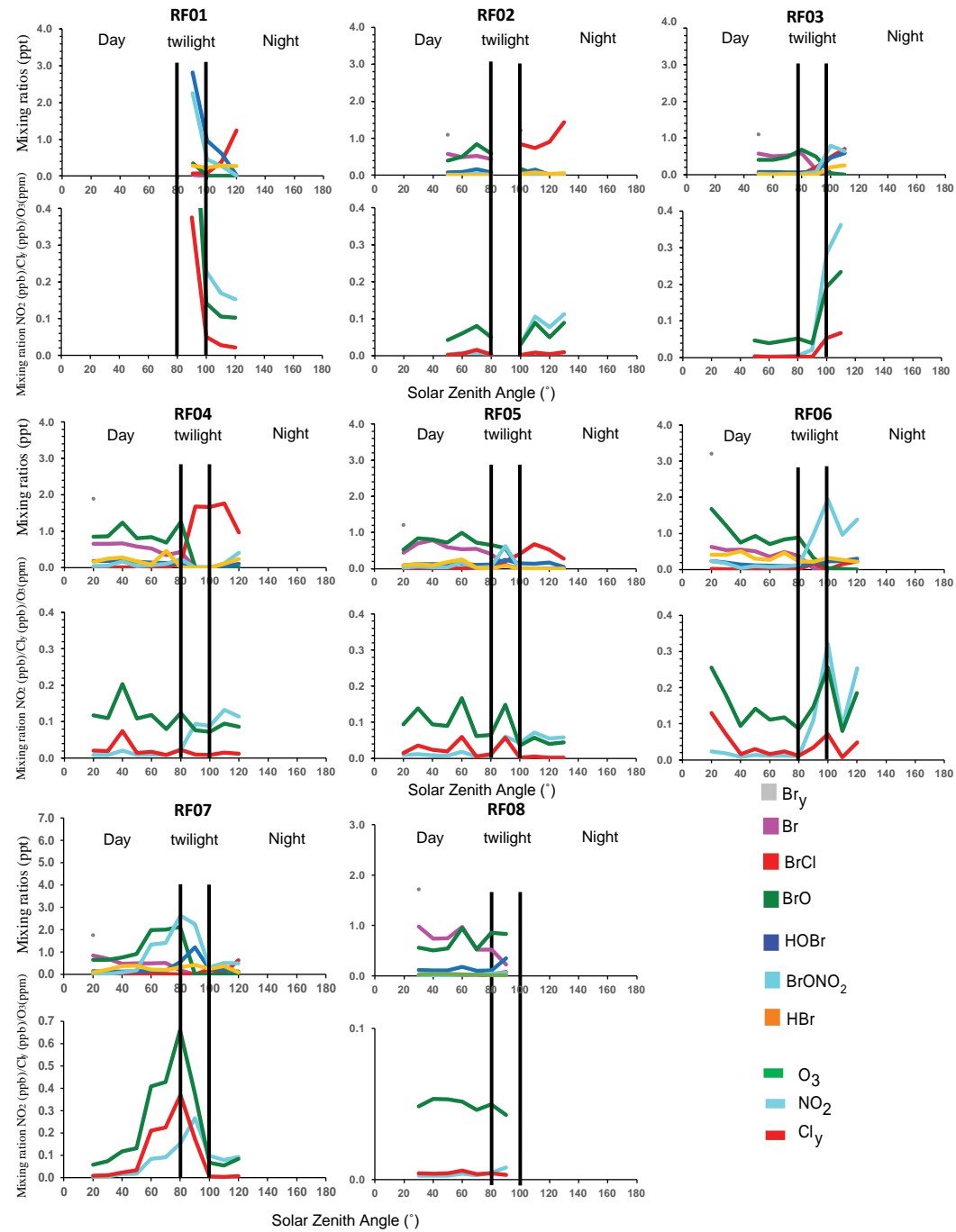

**Figure 5: Inorganic bromine species and main reactants of the inorganic chemistry sampled at exactly the same times and locations as the ATTREX flights developed over the western Pacific. Each separate panel show SZA dependent results for flights RF01, RF03, RF04, RF05, RF06, RF07 and RF08. Only output sampled between 14 and 18 km is considered.**

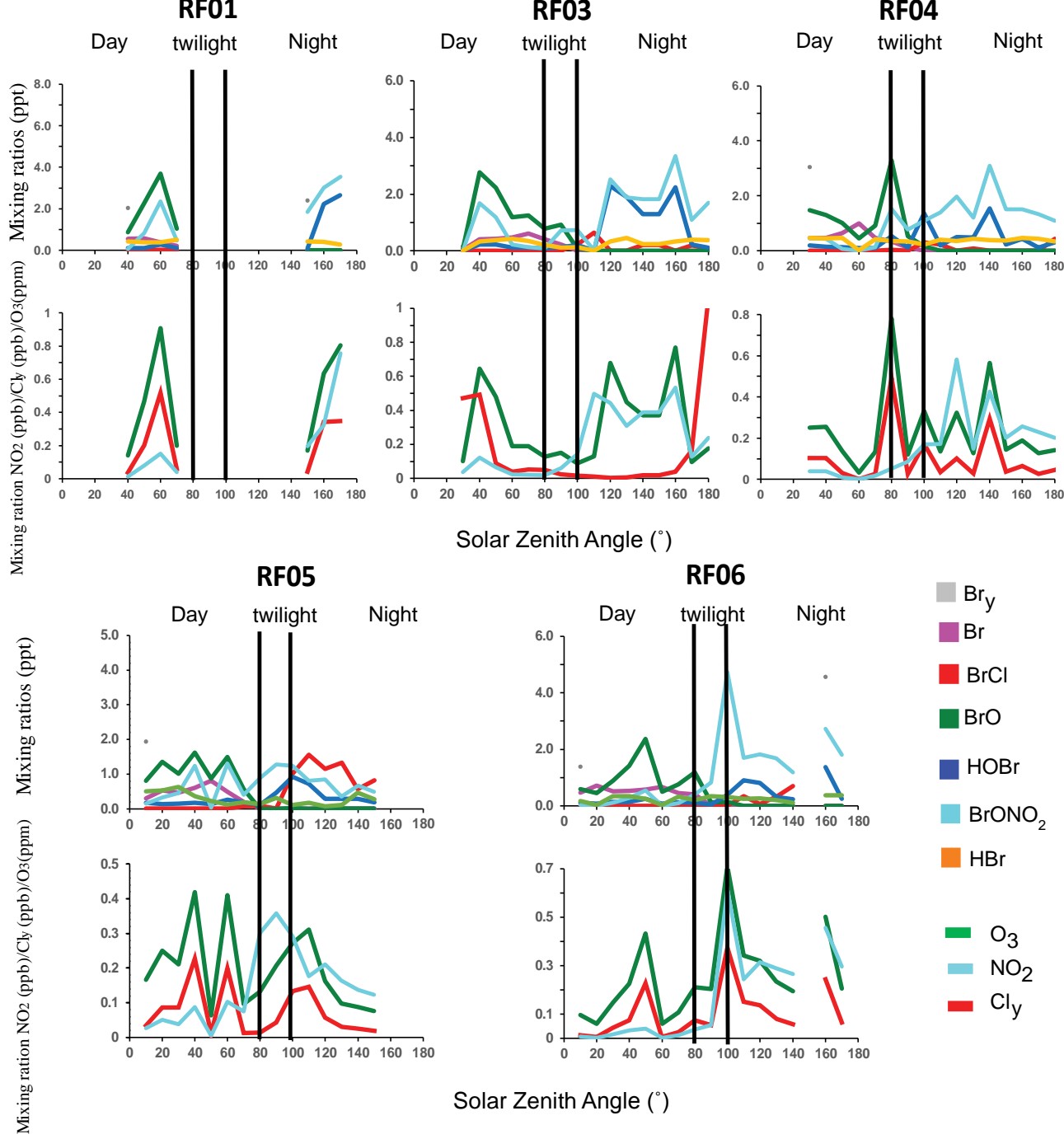

**Figure 6: Inorganic bromine species and main reactants of the inorganic chemistry sampled at the same times and locations as the ATTREX flights over the eastern Pacific. Each separate panel show SZA dependent results for flights RF01, RF03, RF04, RF05 and RF06. Only output sampled between 14 and 18 km is considered.**

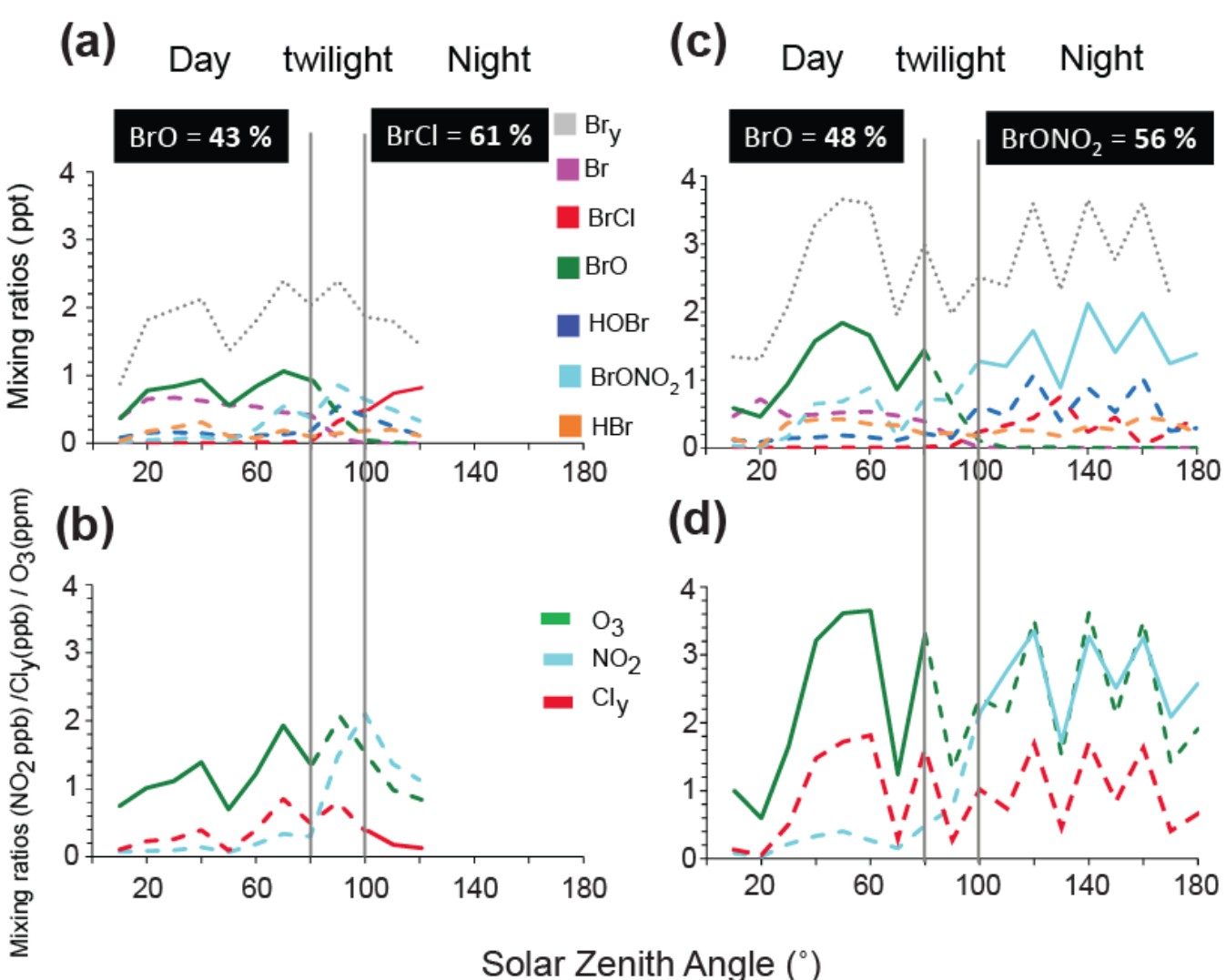

**Figure 7: Average of inorganic bromine species (top panel) and main reactants of the inorganic chemistry (bottom panel) for all ATTREX flights developed over the western Pacific (a and b) and eastern Pacific (c and d). The output from each flight has been sampled only between 14 and 18 km, and that the average has been performed within ± 5 SZA bins. Black boxes indicate the percentage of the dominant Br$_y$ species for day and night at 17 km.**

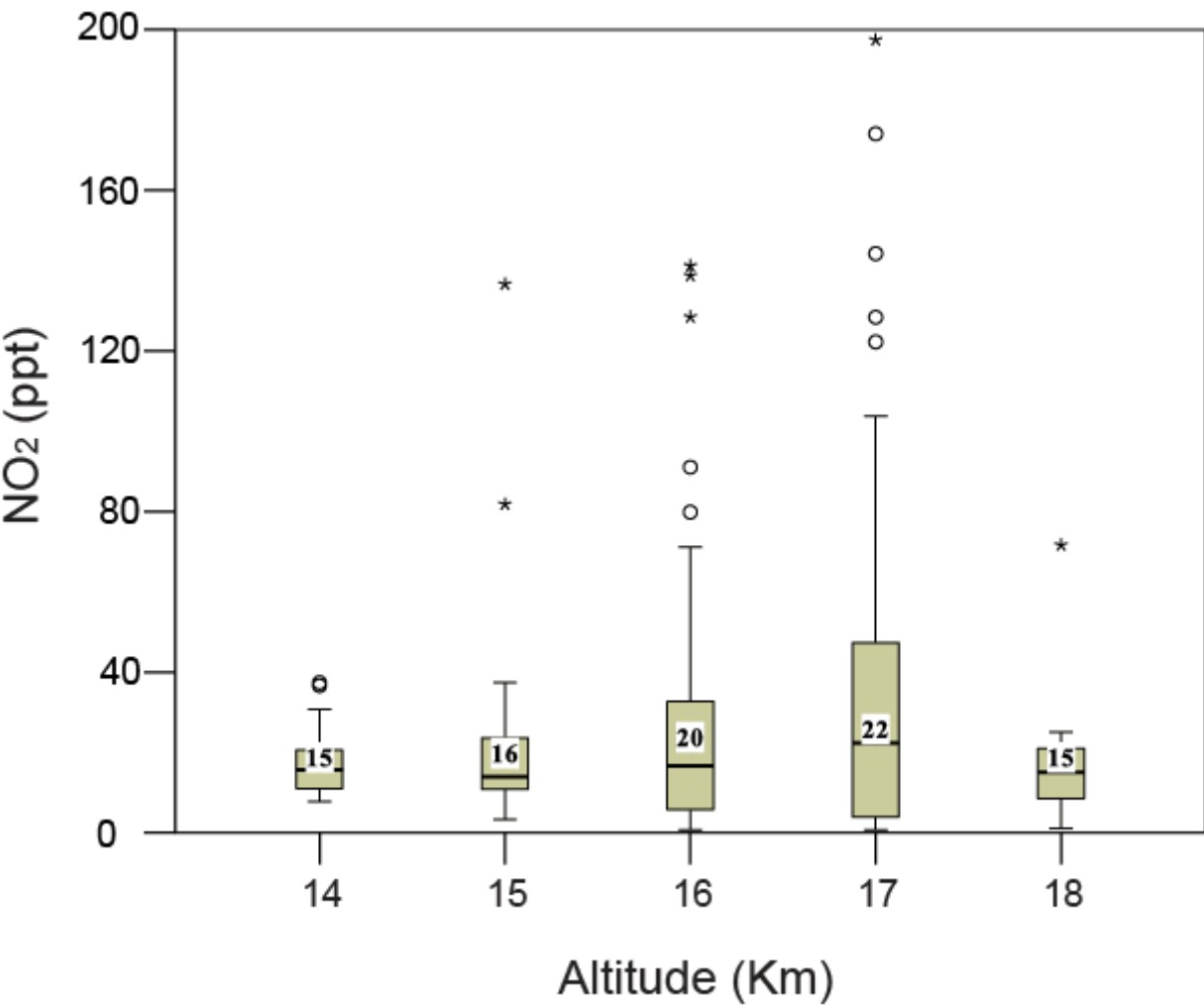

**Figure 8: Boxplot of CAM-Chem NO₂ estimates as a function of altitude during daylight over the eastern Pacific. The filled boxes display the minimum, first quartile (Q1), median (Q2), third quartile (Q3), and maximum values. Black line across the interior of the box represents the median (Q2) of NO₂ for the corresponding altitude. The bottom of the box is the first quartile (Q1) or the 25th percentile; the top of the box is the third quartile (Q3) or the 75th percentile. T-bars extended from the boxes represent the largest and the smallest non-outlier values. Circles and stars are the outliers defined according to the interquartile range (IQR = Q3-Q1): Circles are the "out values" or the NO₂ mixing ratios greater than 1.5 times the interquartile range, and starts are "the extreme values" or the NO₂ mixing ratios greater than 3 times the interquartile range. The numbers inside the box represent the average NO₂.**

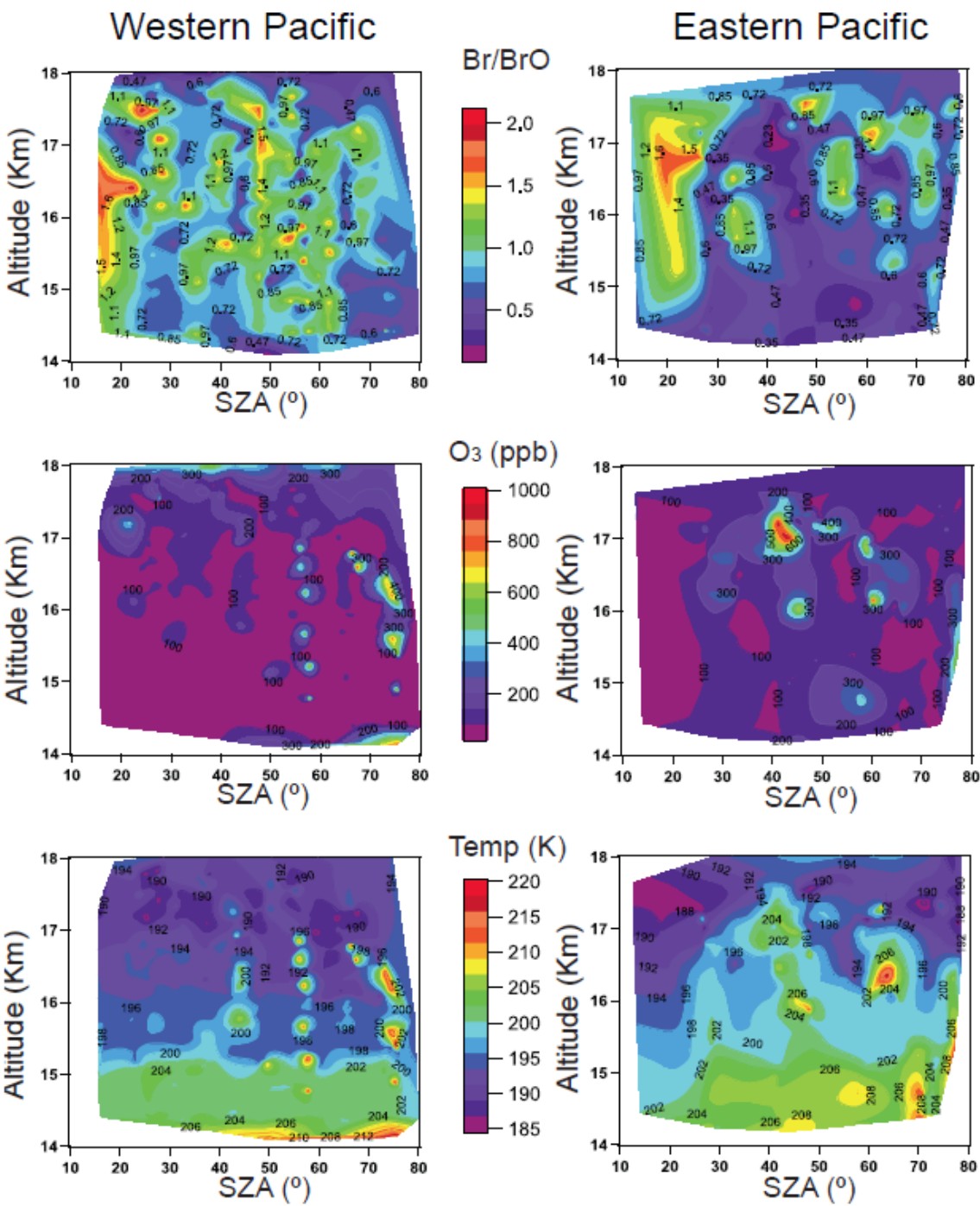

**Figure 9: CAM-Chem model estimates of the daylight Br/BrO ratio (top panel), ozone (middle panel) and temperature (bottom panel) as a function of SZA over the WP and EP.**

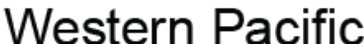

**Figure 10: Inorganic bromine partitioning along ATTREX flight tracks over the western and eastern Pacific. Similar to Fig 4, but ice aerosol recycling reactions in the CAM-Chem chemical mechanism were turned off in this simulation.**

**Table 1: Summary of the CAM Chem model output of $Br_y$ species at the Tropical Tropopause level (~17km)**

|  | Species (time) | WP (ppt) | EP (ppt) |
|---|---|---|---|
| Mean | BrO (day) | 0.79 | 1.43 |
|  | Br (day) | 0.64 | 0.57 |
|  | BrCl (night) | 0.94 | 0.34 |
|  | BrONO2 (night) | 0.29 | 1.41 |
| #Range | $O_3$ (day)* | 35-500 | 35-900 |
|  | $NO_2$ (day) | 1-227 | 0.7-343 |
|  | $Cl_y$ (day) | 1-515 | 1-969 |
|  | Temp (day)** | 188-192 | 188-206 |

# Range defined as min and max values, *units in ppb, **units in K.