# Peer review of "Modelling the Inorganic Bromine Partitioning in the Tropical Tropopause Layer over the Eastern and Western Pacific Ocean."

_Atmospheric Chemistry and Physics, 2016_

## Referee Comment (RC1) · K. Pfeilsticker (Referee) · 26 Jan 2017

**Review for manuscript acp-2016-031**

The manuscript of Navarro et al., reports on Cam-Chem **(**Community Atmosphere Model with Chemistry**)** modelling of the partitioning of inorganic bromine in the tropical tropopause layer (TTL) over the eastern and western Pacific Ocean. The modelling is compared with (averaged) observations of some key species, i.e. of the in-situ measured brominated source gases and $O_3$ from which to the partitioning of inorganic bromine is concluded. Comparisons of measured with and modelling in particular for the yet underexplored TTL are per-se important and interesting. However, based the already-published literature and state knowledge of this field, the paper has major flaw in its present state. My criticism of the present study is based on 5 major deficits (2 more general and 3 more specific comments including one related remark, #3), which are detailed in the following:

**I.)     Methodological deficits of the study**

1. Using (spatial and temporal) averages for fast reacting species (radicals) in photochemical calculations:

   For the modelling of the bromine partitioning, averaged in-situ $O_3$ (together with bromine released from brominated VLSL) is used. Averaging over space and time of concentrations of photochemical reactive species (Figure 1) however is dangerous, since it may lead to incorrect and spurious results for the inferred quantities (for example concentration ratios). In order to see this please consider the rapidly established steady state of [Br] and [BrO] (both being a function of space x and time t) as well as of some (radical) species (e.g. $O_3$, OH, $HO_2$,..) at daytime, which is established through

$$J_{BrO}(x,t) \cdot [BrO(x,t)] = k(T) \cdot [O_3(x,t)] \cdot [Br(x,t)] + \dots$$

or

$$\frac{[BrO(x,t)]}{[Br(x,t)]} = \frac{k(T) \cdot [O_3(x,t)]}{J_{BrO}(x,t)} + \dots$$

where in the present context irrelevant and missing terms are abbreviated by ..... Evidently in order for the equation to make sense k(T), $O_3$(x,t), $J_{BrO}$(x,t) need to be local (i.e. measured or calculated) quantities in the photochemical calculations. When using instead space and/or time-averaged quantities (the overbars denote either space or time averaging), the above mentioned equation would instead read as

$$\overline{J_{BrO}(x,t) \cdot [BrO(x,t)]} = \overline{k(T)} \cdot \overline{[O_3(x,t)]} \cdot \overline{[Br(x,t)]} \cdot + \dots$$

$$\frac{\overline{[BrO(x,t)]}}{\overline{[Br(x,t)]}} = \frac{\overline{k(T)} \cdot \overline{[O_3(x,t)]}}{J_{BrO}(x,t)} + \overline{\dots}$$

It can easily be seen, however, that the [BrO]/[Br] ratio calculated from averaged (space or time) quantities and from local quantities generally differ

$$\overline{\frac{[BrO(x,t)]}{[Br(x,t)]}} = \overline{\frac{\overline{k(T)} \cdot \overline{[O_3(x,t)]}}{\overline{J}_{BrO}(x,t)}} + \overline{\dots}$$

$$\neq \frac{[BrO(x,t)]}{[Br(x,t)]} = \left[\frac{k(T) \cdot [O_3(x,t)]}{J_{BrO}(x,t)} + \dots\right]$$

and accordingly only the latter gives the right answer for the photochemically established [BrO]/[Br] ratios in the atmosphere. In conclusion, when using space and time-averaged ozone concentrations (from the manuscript it is not clear as to whether (k(T), $J_{BrO}$, [Br] and [BrO] were also averaged in the same manner or not, but the answer is somewhat irrelevant to my argument), the modelled [BrO]/[Br] ratio may depart more or less from the actual atmospheric [BrO]/[Br] ratio.

Further when inspecting the ozone concentrations measured by the NOAA instrument in the TTL during ATTREX, it can be seen that actual ozone concentrations may vary by up to a factor of 10 (mostly with height, less in the horizontal in the TLL, see Figure 1) and so the [BrO]/[Br] should cover a similar dynamical range (keeping all the other parameters the same, see Figure 8 in Fernandez et al., (2014)), a behaviour not really recovered when using 1 km binned averages for ozone (Figure 1).

As consequence, the modelled [BrO]/[Br] may not well represent actual [BrO]/[Br] ratio met in the atmosphere, and as thus may not really provide a meaningful information to reader.

Finally, the ozone measured by the NOAA instrument and plotted in Figure 1 (right panel) appears to be spikier (due to any reason, but this could also be visual illusion) than the same ozone plotted in Figures 3 to 8 (panel c) in Werner et al., (2017) for the Eastern Pacific.

2. Averaging (over the space and time domains) concentrations for longer lived species:

For some selected measurements (which ones?) 6 $O_3$ averages and corresponding averages of $CHBr_3$, and $CH_2Br_2$ (out of in total 745 in-situ samples from the EP according to the information provided in Figures 2 and 3a) are inter-compared with the respective model predicted parameters. Averaging over time (or space) for species of different photochemical lifetimes is somewhat problematic.

In order to see this let's consider species of different photochemical lifetimes $\tau_i$ (i = 1, 2, 3…) with a common timescale against atmospheric transport $\tau_m$. Here remember that in general photochemical and dynamical time scales for individual air masses are distributed in space and time (e.g., Waugh and Hall, 2002; Waugh, 2009; for TTL distributions of $O_3$ see c.f., Pan et al., 2014). For the moment, however I skip these complications. Then the joint timescale for photochemical processing and transport is given by

$$\tau_{eff,i} = \left(\frac{1}{\tau_m} + \frac{1}{\tau_i}\right)^{-1}$$

where for the sake simplicity, it is assumed that both photochemical and dynamical processes lead to exponentially decaying concentrations. With these simplifications in mind, the time averaged concentration is then obtained from

$$\overline{c(x)} = \frac{\int\limits_0^\infty c(x,t) \cdot \exp(-\frac{t}{t_{eff,i}}) \cdot dt}{\int\limits_0^\infty \exp(-\frac{t}{t_{eff,i}}) \cdot dt} = -\frac{1}{t_{eff,i}} \cdot \int\limits_0^\infty c(x,t) \cdot \exp(-\frac{t}{t_{eff,i}}) \cdot dt$$

(which is a Laplace transform of c(x,t)). Averaging samples using an appropriate kernel (here exp(- t/t_{eff, i}) is of course different from the (geometrical) average taken over individual samples of $c_k$(x,t), i.e.

$$\overline{c(x,t)} = \frac{1}{m} \cdot \sum_{k=1}^m c_k(x,t)$$

since in the latter calculation any kernel (whether appropriate, or not see below) to calculated averages is discarded.

While for photochemical processes an exponential decay is a reasonable assumption, for dynamical processes in the atmosphere, it is certainly not a good assumption due to the turbulent transport (2-D in the stratosphere). Accordingly, the kernel for dynamical averages (often also called probability density functions, or pdf) does not follow an exponential but rather a power law (e.g., Min et al., 1996; Pierrehumbert and Yang, 1993; Minschwaner et al., 1996; Seo and Bowman, 2000; and for the statistics of actual field data of $O_3$, ClO, and others e.g., see Tuck et al., 2003; Tuck, 2008; Pan et al., 2014). As a consequence, the resulting air mass age spectrum (from which the average age can approximately calculated) is then (approximately) represented by Γ-type functions for the concentrations, which again depend on the time and location in the atmosphere (Hall and Plumb, 1994, Waugh and Hall, 2002).

In consequence, the comparison of modelled and measured averages (for ozone in Figure 1 and 2, and ozone CHBr_3 and CH_2Br_2 shown in Figure 2) does not really make sense, if the pdfs for the atmospheric and modelled samples are not the same in a statistical sense. To put it into simple terms, when averaging over (limited) samples one has to prove that the sampling from the real atmosphere and from the modelled atmosphere are made from the same statistical distributed event in space-time manifold in order for comparisons to make any sense. So certainly the way that the measured and modelled parameters are averaged deserves much more attention in the manuscript.

Finally, noteworthy is that averages over temporally and spatially distributed 'fluctuations' only give the same result for the inferred moments (averages, variance, et cetera) if the system is ergodic, which unfortunately in atmospheric dynamics is mostly not the case. Moreover, the samples need to be huge in order to fulfil one requirement of the central limiting theorem (CLT), that both samples (taken from the atmosphere and the model) converge to the same pdf (given they are the same which needs separately to be proven).

**A remark**

3. Comparing remotely sensed and modelled concentrations:

Moreover, the kernels to calculate averages (and used further on in inter-comparison exercises, see below) in remote sensing applications and in inverse modelling are strongly instrument and measurement-dependent (Rodgers, 2000). Fortunately, they often mask the above described effects due to their limited spatial or time resolution, i.e. their inherent averaging. In fact, in the latter applications these 'kernels' are called 'averaging kernels

(AK)' of the observation and in colloquial English the averaging kernels can be called the 'glasses' by/through which the remote sensing observations were made. So the characteristics of 'the glasses' need to be considered in some way in inter-comparison exercises with modelled quantities (see below).

For some examples of actual AKs, please inspect Figures 5 and 10 (for the weighting of the probed concentrations in the horizontal) in Stutz et al., (2016), Figure 3.5 in Rodgers (2000), Figure 1 (below), or any other study on remote sensing. Chapter 3 in Rodgers, (2000) also discusses the different error sources of the traditional inversion methods used in remote sensing and inverse modelling. It also describes how remotely sensed quantities (here called $c_o(i)$, where i is the retrieval grid number somehow representing the vertical resolution of the measurement) need to be compared with modelled results ($c_m(i)$), i.e. by comparing the inferred $c_o(i)$ with the product $AK \cdot c_m(i)$, where AK is a tensor, of which the columns (or rows) a filled with the individual averaging kernels, displayed for example in Figure 1.

[Figure]

Figure 1: Calculated Averaging Kernels (AK) to infer BrO profiles (Rodgers, 2000) from limb observations during NASA ATTREX using optimal inversion (Stutz et al., 2016, and Werner et al, 2017).

.

In order to avoid these complications using traditional inversion methods for the interpretation of remotely sensed quantities (and in the particular case those arising from multiple scattering due to the a priori unknown spatial distribution and optical properties of aerosol and cloud particles), Stutz et al., (2016) describes a novel (scaling) method for the interpretation data. In effect, the scaling method uses additional information gained from simultaneously in-situ measured gases (i.e. $O_3$) in order to assist the interpretation of remotely sensed $NO_2$, and BrO in the TTL.  Therefore, the scaling method has to be considered as a hybrid method (since it uses information collected by remote sensing and in-situ measurement), which comes with some advantages (and disadvantages) over traditional remote sensing methods. For example, it provides a higher accuracy than methods purely relying on remotely sensed information. Evidently the major disadvantage of the scaling method arises from the need of in-situ information of the probed air masses, i.e. it is suitable for applications from satellites, or high flying balloons. Further the scaling method still requires to carefully consider (by RT calculations simulating the observations) in order simulate how the information (the measured absorption) is obtained.

Accordingly, when applying the scaling technique to their remotely sensed data, Stutz et al., (2016) and Werner et al., (2017) actually simulated each individual observation by modelling the actual RT (and the predicted absorption of the targeted species) by considering instrumental and other details of the measurements as well as predicted curtains of the targeted species, obtained from CTM modelling (TOMCAT/SLIMCAT). This approach (as in any traditional remote sensing application) thus carries over to the analysis any relevant instrumental and observation-related features in the forward modelling of the observation. Evidently, the scaling method (as any traditional inversion method) then allows very close inter-comparisons of the predicted quantities (e.g. trace gas concentrations) with the observations, including a correct attribution of the fraction of the

measured absorption (or slant column) to parts of the atmosphere not directly probed by the observation, however only if the averaging kernels are appropriately considered.

Here please also note that the latter approach to inter-compare remotely sensed data and CTM modelling is not new at all, but e.g., it has been used by our group for more than 2 decades. Further using the scaling method, the calculation of absolute concentrations is achieved using a simultaneously in-situ measured and remotely sensed gas (e.g., $O_3$), together with an appropriate consideration (by RT modelling) of the different sensitivities for detection of the targeted and scaling gas (see equation 14 in Stutz et al., 2016). In effect, the accuracy of the inferred quantities is arguably much better (Stutz et al., 2016) than only relying on remotely sensed quantities for the retrieval of concentrations. Accordingly check your statement on page 6 (lines 14 and 15) for correctness.

**II.)     Comparison with available measured data**

Further, I'm really curious why the authors did not attempt to compare their modelling work with actual measured $NO_2$ and BrO data (potentially) available to the first author for more than a year and which now have been published (Werner et al, 2017). However, when using remotely sensed data in inter-comparison exercises, the kernel for horizontal averaging (see Figure 10 in Stutz et al. (2016)) has to be appropriately taken into account for the modelled data (see my remark #3 above). Further, given that the Werner et al., (2017) manuscript (which the first author of the present article co-authored) was submitted earlier (July 17, 2016) than the present manuscript (Nov. 18, 2016), the statement on page 4 (line 18) is not well based. By being more specific, the lack of a tight comparison of the modelled results with existing measured data give rise to some more deficits of the present study:

4. Simulated $NO_2$:

   For the Eastern Pacific TTL, the CAM-Chem model predicts $NO_2$ between 0.7 – 343 ppt at daytime (Table 1 and Figure 6). No reasons are provided for the elevated $NO_2$ in the TTL over the EP, except that modelled air masses are affected by 'pollution'. However, no other indication (neither from, for example, measured CO during NASA-ATTREX (UCATS) nor any further evidence inferred from the model) is provided that in fact polluted air masses were reaching the TTL over the EP in early 2013. In fact, the $NO_2$ mixing ratios reported by Werner et al., (2017) were < 20 ppt in the TTL, and they agree well (within the error bars +-/10 ppt) with the predictions of the TOMCAT/SLIMCAT simulations assuming no contribution from 'pollution'. In all these respects, and in particular with respect to the discussion provided above under point #3, the statement on page 6 (lines 14 and 15) is not well founded.

   Accordingly, in any further study information has to be provided why for the EP TTL the Cam-Chem model predicts $NO_2$ concentrations much large than observed. In addition, coherent evidences both from observations and modelling has to be provided (for example from $CO/O_3$ and $CH4/O_3$) that indeed the TTL over the EP is affected by 'pollution'.

5. Simulated inorganic bromine, its partitioning and spatial patchiness:

   A major part of the study is devoted to model the bromine partitioning. First, I found it hard understand why the model does not really reproduce the increase in total inorganic bromine with increasing height (potential temperature) within the TTL, mainly caused by the destruction of brominated VSLS. This is somehow curious since the bromine concentrations at the lower boundary reported by Navarro et al., (2015) (page 3, line 18 and 19; VSLS; 3.84 ± 0.64 and 3.18 ± 1.49 ppt from WP and EP, respectively, and inorganic bromine 3.02 ± 1.90 ppt of Bry over the EP and 1.97 ± 0.21 ppt over WP) are in reasonable agreement with the data for the EP TTL, reported by Werner et al., (2017). Moreover, in

the Cam-Chem model inorganic bromine (in gaseous form) barely increase from ~2 ppt (from the lower boundary at 14 km) to ~3 ppt at 18 km (Figures 4 and 8), in stark contrast with the observations presented in the Werner et al., study for the upper levels of the TTL over EP. Here, depending on the flight, inorganic bromine ranges from (2.63 ± 1.04) ppt (range from 0.5 ppt to 5.25 ppt) to 5.1 ±1.57 ppt (at $\Theta$ = 390 - 400 K) to 6.74±1.79 ppt (at $\Theta$ > 400 K), in agreement with the measured destruction of brominated VSLS species (Navarro et al., 2015, and Figure 14 in Werner et al., 2017). So the obvious question is: Does the model either not efficiently destroy the brominated VSLS, and/or does the missing bromine reside in/on particles? If the latter is the case, the bromine up-taken by particles need to be rather large (2 – 3 ppt) in order close the bromine budget. So some information has to be provided how the bromine budget is closed in the model, and in particular on how much bromine is up-taken by the particles.

Next even though the modelled absolute amount of gaseous inorganic bromine likely may not affect the $Br_y$ partitioning, the modelled [Br]/[BrO] (cited: (1) …. the modelled Br/BrO maximizes at 17 km from page 7, line 7 to 17 and in Figure 7 and (2) …..that Br/BrO may become as large as 2 in the TTL of the EP, see Figure 7) deviates from expectations based on the amount of ozone and its increase with height (see Figure 1 left panel, and Figure 3 to 8 in Werner et al., 2017), and the modelled bromine partitioning in the TTL as function of ozone (Fernandez et al., 2014, Figure 8). In fact, these findings largely contrast with early findings based on the Br/BrO ratio in TTL (at 17 km) during daytime c.f., by Fernandez et al., 2014 (Figure 1, left panel where Br/BrO < 0.6 at 17 km during tropical noon), Schmidt et al., (2016) (Figure 1), or lately the model results presented in Werner et al., (2017) (inspect Figure 3 – 8, Br/BrO < 0.6 at 17 km). Reasons for this discrepancy, including a discussion how the averaging of the ozone and the source gas concentrations and of other quantities impacts the modelled Br/BrO ratio (see points 1 and 2 above) certainly need to be addressed in any future study.

Finally, the model predicts a certain patchiness (on spatial scales of some hundred kilometres) of the modelled Br/BrO ratio at 17 km for the EP (and WP), with [Br]/[BrO] ratios ranging from below < 0.5 to about 2. No further reason for this patchiness is provided in the manuscript. If air masses entrained by mesoscale convection into the TTL are responsible for this patchiness, then it also needs to be seen in other gases (e.g. CO, $CH_4$…), but again no evidence for this is provided in the manuscript. The predicted patchiness also contrasts with measured $O_3$, $NO_2$, and BrO, in particular since the remote sensing measurements can easily resolve horizontal variations of the measured quantities on the hundreds of kilometre scale (e.g., Stutz et al., (2016) figure 9, and Werner et al., (2017), figures 3 to 8). Further, since at daytime a rapid steady state is established between Br and BrO (see above) as function of the solar illumination and $O_3$ concentration, it is difficult to infer from measured data any reason for the predicted patchiness in the [Br]/[BrO] ratio.

6. Error and uncertainties:

   Finally, as an experimentalist who devotes 85 % of his efforts in the interpretation of data to get a handle on a reliable (thus justifiable) errors and uncertainties of the measured quantities, I always find it curious if studies lack a proper discussion of errors and uncertainties of the presented results. In modelling studies, this could for example be done by (1) inspecting respective Jacobians of the relevant quantities, (2) investigate differences in the modelled fields from 'on and off' runs, and (c) perform ensemble runs et cetera. So also in this respect, the present study largely lacks this requirement for robust science.

**Summary**

Given the above described methodological deficits (points 1 and 2), the lacking comparison of the modelled results with actual measured data (points 4 and 5), and the lacking discussion of errors and uncertainties (point 6), unfortunately it is impossible to recommend the manuscript for publication in the present form.

**Additional references:**

Fernandez et al., Bromine partitioning in the tropical tropopause layer: implications for stratospheric injection, Atmos. Chem. Phys., 14, 13391-13410, doi:10.5194/acp-14-13391-2014, 2014.

Hall, T. M., and R. A. Plumb, Age as a diagnostic of stratospheric transport, JGR, 99, 1059 - 1070, 1994.

Minschwaner et al., Bulk properties of isentropic mixing into the tropics in the lower stratosphere, JGR, 101, 9433-9436, 1996.

Min et al., Levy stable distributions for velocity and velocity difference in systems of vortex elements, Phys. Fluids, 8, 1169, 1996.

Navarro et al., Airborne measurements of organic bromine compounds in the Pacific tropical tropopause layer, PNAS, 112, 51, 2015.

Pan, L. L., L. C. Paulik, S. B. Honomichl, L. A. Mucnchak, J. Bian, H. B. Selkirk, and H. Vömel (2014), Identification of the tropical tropopause transition layer using the ozone-water vapor relation-ship, J. Geophys. Res. Atmos., 119, 3586–3599, doi:10.1002/2013JD020558.

Pierrehumbert, R. T., and H. Yang, Global chaotic mixing on isentropics surfaces J, Atmos. Sci.,5 0, 2462-2480, 1993.

Rodgers, C., Inverse methods for atmospheric sounding, World Scientific, Singapore, New Jersey, London, Hongkong, 2000.

Schmidt et al., Modelling the observed tropospheric BrO background: Importance of multiphase chemistry and implications for ozone, OH, and mercury, JGR, 121, doi:10.1002/2015JD024229, 2016.

Seo, K.-H. and K.P. Bowman, Lévy flights and anomalous diffusion in the stratosphere, JGR, 105, D10, 12295–12302, 2000.

Stutz et al., A New Differential Optical Absorption Spectroscopy Instrument to Study Atmospheric Chemistry from a High-Altitude Unmanned Aircraft, AMTD, 148, doi:10.5194/amt-2016-251, http://www.atmos-meas-tech-discuss.net/amt-2016-251/, 2016.

Tuck et al., Law of mass action in the Arctic lower stratospheric polar vortex January–March 2000: ClO scaling and the calculation of ozone loss rates in a turbulent fractal medium, JGR, 108, NO. D15, 4451, doi:10.1029/2002JD002832, 2003

Tuck A., Atmospheric Turbulence: a molecular dynamics perspective, Oxford University Press, 2008.

Werner, B., Stutz, J., Spolaor, M., Scalone, L., Raecke, R., Festa, J., Colosimo, S. F., Cheung, R., Tsai, C., Hossaini, R., Chipperfield, M. P., Taverna, G. S., Feng, W., Elkins, J. W., Fahey, D. W., Gao, R.-S., Hintsa, E. J., Thornberry, T. D., Moore, F. L., Navarro, M. A., Atlas, E.,

Daube, B. C., Pittman, J., Wofsy, S., and Pfeilsticker, K.: Probing the subtropical lowermost stratosphere and the tropical upper troposphere and tropopause layer for inorganic bromine, Atmos. Chem. Phys., 17, 1161-1186, doi:10.5194/acp-17-1161-2017, 2017.

Waugh, D. W. & Hall, T. M. Age of stratospheric air: Theory, observations and models. Rev. Geophys. 40, 1-10, 2002.

Waugh D., The age of stratospheric air, Nature Geoscience, 2, Jan.2., 2009.

---

## Referee Comment (RC2) · Anonymous Referee #2 · 29 Jan 2017

Review of: Modelling the Inorganic Bromine Partitioning in the Tropical Tropopause over the Pacific Ocean by Navarro et al.

General Remarks: This paper uses global model stimulations to examine the inorganic bromine (Bry) budget of the TTL, building on the work of Navarro et al. (2015). In that work, the authors (a) presented measured (and modelled) vertical profiles of brominated very short-lived substances (VSLS), such as CHBr3 and CH2Br2, from recent NASA ATTREX flights, and (b) used a model that reproduces the observations well (CAM-Chem), to estimate the contribution of VSLS to Bry in the TTL (highlighting the significance of that contribution).

In the present work, the same approach is adopted as above, though the focus is more on understanding the modelled Bry speciation in the TTL, the Bry diurnal cycle, and

differences between the West and East Pacific (where the ATTREX missions sampled). The model results from this work show that BrO and Br are the most abundant daytime species, while BrCl and BrNO3 are more important at night. The authors also discuss differences in modelled Bry partitioning between the West and East Pacific, and briefly the sensitivity to heterogeneous processes on ice.

Overall, this paper is an interesting case study that provides an (incremental) advance on our understanding of Bry partitioning in the TTL over the Pacific. In the absence of new BrO measurement data being included in the manuscript, this advance is somewhat subtle when viewed alongside the modelling study of Fernandez et al. (2014, ACP) that also used CAM-Chem to look at TTL bromine partitioning, in some detail. I have outlined three major areas below that should be addressed before publication.

Major Comments: 1. The authors should ensure that the Introduction clearly sets out which of the broad model findings have come before, in order to help determine what the main motivation and purpose of this paper is. For example, the model results on zones where the Br/BrO ratio is >1 in the UTLS are interesting, though have been discussed previously by Fernandez et al. (2014, ACP) and Saiz-Lopez and Fernandez (2016, GRL). The same can be said about the analysis of the Bry diurnal cycle and Bry speciation in the TTL, and their sensitivity to heterogeneous processes. Is the advance here that this is simply a CAM-Chem case study for the ATTREX campaign period? If so, that is fine, but the measurements of BrO (and NO2) from ATTREX would very much strengthen the paper and help corroborate the modelled fields. In the first paragraph of Results and Discussion, it is noted that "BrO and NO2 measurements from the ATTREX mission were still under examination by the time of this analysis". Is this still the case? It strikes me that it is quite odd that these data are not included here.

2. The most novel aspect of this work is the examination of differences between the W and E Pacific. The discussion of chlorine could be improved in this regard. If differences in Cly between the two regions can impact local Bry partitioning, some discussion on how well constrained the actual Cly simulation over the WP (average up to 84

ppt Cly in daylight) and EP (up to 181 ppt Cly in daylight) is needed. At the very least some more details of the chlorine simulation could be given. More broadly, I would suggest that the title of the paper should reflect that the emphasis of the paper is on the differences between W and E Pacific.

3. The writing is quite awkward in many places and the paper would benefit from a very thorough check/read through. In addition, although the paper is compact, it would benefit from some sub-headings, particularly in the Results and Discussion section (e.g. Model-measurement O3 comparison, Diurnal cycle in Bry partitioning, or something similar).

Other points:

*Abstract* Sometimes the "E" in Eastern (Pacific) and the "W" in Western (Pacific) are capitalized and sometimes they are not. Please be consistent throughout manuscript (including in figures and captions).

*Introduction* P1, L33: "Bry" is defined early on in the manuscript but "inorganic bromine" is used in numerous places after that. I suggest changing the latter to the former where appropriate throughout the manuscript.

P3, L2: Struck me that introducing the "proposed tropical ring of atomic bromine" here is odd. Could you not make mention of these papers earlier in the Introduction?

P3, L3: "section 3" — "Section 3"

*Methods* I would separate out Methods into 2.1 Observations and 2.2 Modelling. This section seems quite unstructured in its present form. The Modelling section needs more details as to the ozone precursor emissions that were used in CAM-Chem + some brief information of the chlorine simulation.

P3, L12: Has "very short-lived organic substances" not already been defined as VSLorg? Also, as the focus of this work is on VSLS, it would be better if some of the gases listed (e.g. CHBr3 etc.) are actually referred to earlier in the Introduction (maybe

around line 31).

P3, L17: The two sentences beginning "At the tropopause level" seems out of place. Is this not motivation/background for the present study and should it not appear in the Introduction?

P4, L4: CAM-Chem has already been defined.

P4, L21: "for both, the WP and EP respectively" — "for both the WP and EP".

P5, L7, Sentence beginning "During ATTREX" to the end of the paragraph. This text describes the different sampling times/paths of the observations/flights and would be better placed in Section 2. In the current location it disrupts the flow of results. Similarly, consider moving Figure 3 to the Measurements section.

---

## Author Comment (AC1) · 16 Mar 2017

Responses have been uploaded also as a supplement pdf file. Please refer to this file as the "plain text" option do not show equations or reactions added to the responses.

Review of: Modelling the Inorganic Bromine Partitioning in the Tropical Tropopause over the Pacific Ocean by Navarro et al.

The manuscript of Navarro et al., reports on Cam-Chem (Community Atmosphere Model with Chemistry) modelling of the partitioning of inorganic bromine in the trop-ical tropopause layer(TTL) over the eastern and western Pacific Ocean. The modelling is compared with (averaged) observations of some key species, i.e. of the in-situ mea-sured brominated source gases and O3 from which to the partitioning of inorganic bromine is concluded. Comparisons of measured with and modelling in particular for

the yet underexplored TTL are per-se important and interesting. However, based the already-published literature and state knowledge of this field, the paper has major flaw in its present state. My criticism of the present study is based on 5 major deficits (2 more general and 3 more specific comments including one related remark, #3), which are detailed in the following:

Response: We thank Prof. Pfeilsticker for his comments. We really appreciate and welcome any feedback that could be used to improve our manuscript.

I.) Methodological deficits of the study 1. Using (spatial and temporal) averages for fast reacting species (radicals) in photochemical calculations: For the modelling of the bromine partitioning, averaged in-situ O3 (together with bromine released from brominated VLSL) is used. Averaging over space and time of concentrations of photochemical reactive species (Figure 1) however is dangerous, since it may lead to incorrect and spurious results for the inferred quantities (for example concentration ratios). In order to see this please consider the rapidly established steady state of [Br] and [BrO] (both being a function of space x and time t) as well as of some (radical) species (e.g. O3, OH, HO2,..) at daytime, which is established through eq. (1) (see eq in pdf version) where in the present context irrelevant and missing terms are abbreviated by .... . Evidently in order for the equation to make sense k(T), O3(x,t), JBrO(x,t) need to be local (i.e. measured or calculated) quantities in the photochemical calculations. When using instead space and/or time averaged quantities (the overbars denote either space or time averaging), the above mentioned equation would instead read as eq. (2) (see eq in pdf version) It can easily be seen, however, that the [BrO]/[Br] ratio calculated from averaged (space or time) quantities and from local quantities generally differ eq. (3) (see eq in pdf version) and accordingly only the latter gives the right answer for the photochemically established [BrO]/[Br] ratios in the atmosphere. In conclusion, when using space and time-averaged ozone concentrations (from the manuscript it is not clear as to whether (k(T), JBrO, [Br] and [BrO] were also averaged in the same manner or not, but the answer is somewhat irrelevant to my argument), the modelled [BrO]/[Br]

[Figure]

ratio may depart more or less from the actual atmospheric [BrO]/[Br] ratio.

Response: We agree with the reviewer regards the different results eq. (1) and (2) can give depending on the spatial or temporal average applied to either modelled and/or measured data. In doing so, please note the importance of assuming that the "irrelevant and missing terms" for the case of the [Br]/[BrO] ratio are negligible compared to the dominant production and loss channels for atomic bromine in the atmosphere. Only if those irrelevant terms can be neglected, then eq. (1) and (2) can be written in its simple ratio form (i.e., dependent only on the photodissociation rate constant JBrO and on the pseudo first-order reaction rate k(T)*[O3]). Having said this, we would like to make the following points clear: - Photochemical production of bromine atoms in the Upper Troposphere (UT) is dominated by BrO photolysis (see reaction in pdf version), while chemical losses occur mainly through the bimolecular thermal reaction (see reaction in pdf version), which itself represent 98% of the total atomic Br losses, (see Saiz-Lopez and Fernandez, GRL, (2016) for details). The much rapid reactivity of these two channels (respect to the neglected terms in eq. (1)) allows the establishment of a rapid pseudo steady-state between Br and BrO. Thus, if we accept that a rapid steady-state is reached between these two species, and also agree that neglected terms are irrelevant for this case, then it is evident that Br and BrO abundances must be related by a mathematical expression which considers only the reaction rates connecting those species. - We do not mention at all the explicit relation between the [Br]/[BrO] ratio and JBrO, k(T) and O3 in this manuscript. We only mentioned the relevance of computing the [Br]/[BrO] ratio in relation to the proposed tropical rings of atomic halogens, whose drivers are described in a preceding paper (Saiz-Lopez and Fernandez, GRL, 2016). Anyhow, and being aware of the averaging issues mentioned by the reviewer, in that work we performed spatial and temporal averages of Br and BrO abundances, as well as to all rate constants affecting atomic bromine production and losses, and found an excellent correlation between instantaneous (e.g., hourly) and averaged (e.g., monthly) modelled output. Indeed, Fig. 3 in Saiz-Lopez and Fernandez, GRL, (2016), shows the vertical profile of the [Br]/[BrO] ratio as well as the JBrO/K(t)[O3] ratio obtained with CAM-Chem for an equivalent setup simulation as the one used in this work. The main panels show either annual values for the tropical (20°N-20°S) average (Fig.3D) or monthly values within the Tropical Western Pacific (TWP, Fig.3E), while the inset panels show the hourly output modelled linear correlation (daytime masked) between[Br]/[BrO] and JBrO/K(t)[O3]. For the case of the Tropical region, r2 = 0.9782, while for the TWP, r2 = 0.99695, with ratio values spanning approximately from 0 to 3. Equivalent results were also obtained when individual model gridbox (lat,lon,z) were sampled, either hourly or monthly. - The model output for the present simulations was instantaneous (i.e. hourly, half-hour could have been the highest possible resolution within CAM-Chem). The model was run in Specified Dynamic (SD) mode (i.e., considering the current meteorology prevailing during the campaign) and was further sampled at the correspondent latitude, longitude, height and time "gridbox(lat,lon,z,t)" that best matched the ATTREX flight-track. Thus, nor spatial neither temporal averaging of [Br], [BrO] and its ratio [Br]/[BrO] or other atmospheric quantities in CAM-Chem (JBrO, K(T), T, O3, etc.) have been performed to extract the model output. As by the time of preparing this MS there were no other measurements available than ozone, we decided to present the validation of instantaneous O3 measurements for all independent flights (Figure 1, now Fig. 2 in the revised version) and then all atmospheric model variables were averaged into 1 km height bins so the output correspondent to each independent flight could be compared with each other. Being this one a modelling paper (as it is clearly stated in the title), we found appropriate to also perform the spatial-temporal mean of all flight-tracks, which are then used to present a more general representation of the modelled state of the atmosphere in the rather yet unexplored tropical upper troposphere (e.g. the Vertical Profiles shown in Figs. 4, 7 and 8). We are aware that these mean vertical profiles are not descriptive of each of the independent flights, but they are certainly representative and illustrative of the mean state of the tropical atmosphere within the Eastern and Western Pacific when sun photochemistry is turned on and off. - For the case of Fig. 5, which shows the temporal evolution (i.e., SZA dependent) of the dominant bromine species and the main inorganic reactants during the day, twilight

and night, we decided based on Prof. Pfeilsticker's comment, to present in addition to the mean temporal profile, the independent results for each specific flight. In this way, the changes in partitioning of the dominant species can be addressed directly in response to the current abundance of ozone, Cly and/or NO2 prevailing during each flight track. This helps, for example, to support the large inhomogeneity we suggested in previous studies (see Saiz- Lopez and Fernandez, GRL, (2016) for details) for the [Br]/[BrO] ratio, which is modelled to be larger than 1 at a fixed SZA for one of the flights but not for the others: for this cases, as highlighted by the reviewer, performing the spatial-temporal mean of all flights does not illustrate the intrinsic variability found on the abundance of ozone, bromine and all related short-lived quantities. Following the Prof. Pfeilsticker's comments and the above responses, we have modified the text in as follows: Page 5 line 20: "Model hourly output was sampled at exactly the same times and locations as the ATTREX measurements, without performing neither spatial nor temporal averaging on model grids. Once each independent flight track was extracted from the model output, all atmospheric quantities were averaged into 1 km altitude bins, to compare with measured data." Page 7 line 23: "Figure 5 compares the mean abundances observed in the EP and WP considering all flights. Even when these results are not descriptive of each of the independent flight, they are representative and illustrative of the mean state of the tropical upper atmosphere within the eastern and western Pacific in the presence and absence of sunlight. Equivalent results but for each independent flight are show in the Supplementary online material" Page 9 line 1: "A closer inspection on each independent flight (Figs. S1 and S2) reveals the large inhomogeneity of the tropical rings of atomic bromine. In the EP, Br surpass BrO mixing ratios at $60°$ SZA for flights RF04 and RF06, but as the remaining flights sampled larger BrO mixing ratios, the mean EP abundances shown in Fig. 5c shows Br/BrO > 1 only at $20°$ SZA. Similarly, the mean results shown in Fig. 5a for the WP show BrO > Br at all times, but RF02 and RF03 show the ratio Br/BrO to be larger than one at $50°$ SZA. This highlight the importance of considering non-averaged (both spatially and temporal) model output to determine the concentration of photochemical

reactive species or other atmospheric quantities such as the Br/BrO ratio." Page 9 line 14: "Figure 7 shows the distribution of the Br/BrO ratio over the WP and EP, and its correlation with ozone concentrations and temperatures. The results are based on the mean 1 km binned data for all track flights, although equivalent conclusions can be reached for each independent transect."

Further when inspecting the ozone concentrations measured by the NOAA instrument in the TTL during ATTREX, it can be seen that actual ozone concentrations may vary by up to a factor of 10 (mostly with height, less in the horizontal in the TLL, see Figure 1) and so the [BrO]/[Br] should cover a similar dynamical range (keeping all the other parameters the same, see Figure 8 in Fernandez et al., (2014)), a behaviour not really recovered when using 1 km binned averages for ozone (Figure 1). As consequence, the modelled [BrO]/[Br] may not well represent actual [BrO]/[Br] ratio met in the atmosphere, and as thus may not really provide a meaningful information to reader.

Response: The sensitivity of the [Br] and [BrO] abundances to ozone mixing ratio shown in Fig. 8 of Fernandez et al., (2014), was performed using a box-model constrained with many chemical parameters (not relevant to described here) and also constant temperature (T=190 K). Note that Fig. 9 of the same paper, shows an additional sensitivity of bromine abundances to temperature. As the bi-molecular thermal reaction (Br + O3—> Br, see equation in PDF version) decrease with increasing temperature (K(T)= 1.60 x 10-11 e(-780/T)), we do not expect the modelled [Br]/[BrO] ratio to cover a similar dynamical range than the ozone variations: it would also depend on the temperature change associated to the air parcels considered (K(T) changes a factor $\sim$ 1.5 between 190 and 210 K). Also note that in the current modelling approach, ATTREX ozone measurements has been used to validate CAM-Chem performance (Fig. 1), but individual measurements values have not been used to compute the [Br]/[BrO] ratio. The modelled [Br]/[BrO] ratio shown in Fig. 7A was computed considering CAM-Chem ozone (Fig. 7B) and temperature (Fig. 7C) fields. Modelled ozone abundances change between 100 ppb and 600 ppb and [Br]/[BrO] ratios between 0.35 and 2.0,

so the modeled range between maximum and minimum values span approximately a factor of 6. With regards to the last sentence, in any case, spatial/temporal average of the modelled [Br]/[BrO] ratio may depart more or less from the actual modelled JBrO/K(T)[O3] ratio, but not such a strong affirmation can be made respect to the atmospheric [Br]/[BrO]. There are no means we can compare here modelled [Br]/[BrO] with atmospheric [Br]/[BrO] ratios as that would have implied the simultaneous atmospheric measurements of atomic Br and BrO. Even when Br atoms (as well as atmospheric Bry) can be inferred from BrO measurements, this procedure also implies including a detailed chemical mechanism for bromine. Thus, any modelled ratio (such as [Br]/[BrO], [Br]/[Bry] or the more commonly used [BrO]/[Bry] partitioning) can be compared to atmospheric ratios as long as the chemical mechanism considered is appropriate to represent the chemistry of that specific portion of the atmosphere. We are quite confident that bromine chemistry in CAM-Chem is very well represented (as in many other global models) and that all the main chemical reactions reported in the literature are up-to-date in our setup. Thus, we found quite interesting to compute atmospheric ratios between the major species to establish which ones are the dominant species, and in this way, validate them against measurements to properly constrain chemistry-climate models.

Finally, the ozone measured by the NOAA instrument and plotted in Figure 1 (right panel) appears to be spikier (due to any reason, but this could also be visual illusion) than the same ozone plotted in Figures 3 to 8 (panel c) in Werner et al., (2017) for the Eastern Pacific.

Response: Measured ozone values have been processed as described in Section 2.1. We found quite difficult to compare the Vertical Profiles shown here in Figure 2 (note that Fig. 1 was shifted to Fig. 2 in the revised manuscript) with the temporal timeseries shown for each flight in Werner et al., 2017, Figs. 3-8c. In any case, the spiker representation of O3 measurements presented here might explain the large range between the maximum and minimum [Br]/[BrO] ratios you expected to find.

2. Averaging (over the space and time domains) concentrations for longer lived species: For some selected measurements (which ones?) 6 O3 averages and corresponding averages of CHBr3, and CH2Br2 (out of in total 745 in-situ samples from the EP according to the information provided in Figures 2 and 3a) are inter-compared with the respective model predicted parameters. Averaging over time (or space) for species of different photochemical lifetimes is somewhat problematic. In order to see this let's consider species of different photochemical lifetimes $\tau i$ (i = 1, 2, 3...) with a common timescale against atmospheric transport $\tau m$. Here remember that in general photochemical and dynamical time scales for individual air masses are distributed in space and time (e.g., Waugh and Hall, 2002; Waugh, 2009; for TTL distributions of O3 see c.f., Pan et al., 2014). For the moment, however I skip these complications. Then the joint timescale for photochemical processing and transport is given by where for the sake simplicity, it is assumed that both photochemical and dynamical processes lead to exponentially decaying concentrations. With these simplifications in mind, the time averaged concentration is then obtained from (which is a Laplace transform of c(x,t)). Averaging samples using an appropriate kernel (here exp(- t/teff, i) is of course different from the (geometrical) average taken over individual samples of ck(x,t), i.e. since in the latter calculation any kernel (whether appropriate, or not see below) to calculated averages is discarded. While for photochemical processes an exponential decay is a reasonable assumption, for dynamical processes in the atmosphere, it is certainly not a good assumption due to the turbulent transport (2-D in the stratosphere). Accordingly, the kernel for dynamical averages (often also called probability density functions, or pdf) does not follow an exponential but rather a power law (e.g., Min et al., 1996; Pierrehumbert and Yang, 1993; Minschwaner et al., 1996; Seo and Bowman, 2000; and for the statistics of actual field data of O3, ClO, and others e.g., see Tuck et al., 2003; Tuck, 2008; Pan et al., 2014). As a consequence, the resulting air mass age spectrum (from which the average age can approximately calculated) is then (approximately) represented by $\Gamma$-type functions for the concentrations, which again depend on the time and location in the atmosphere (Hall and Plumb, 1994, Waugh and

Hall, 2002). In consequence, the comparison of modelled and measured averages (for ozone in Figure 1 and 2, and ozone CHBr3 and CH2Br2 shown in Figure 2) does not really make sense, if the pdfs for the atmospheric and modelled samples are not the same in a statistical sense. To put it into simple terms, when averaging over (limited) samples one has to prove that the sampling from the real atmosphere and from the modelled atmosphere are made from the same statistical distributed event in space-time manifold in order for comparisons to make any sense. So certainly the way that the measured and modelled parameters are averaged deserves much more attention in the manuscript. Finally, noteworthy is that averages over temporally and spatially distributed 'fluctuations' only give the same result for the inferred moments (averages, variance, et cetera) if the system is ergodic, which unfortunately in atmospheric dynamics is mostly not the case. Moreover, the samples need to be huge in order to fulfil one requirement of the central limiting theorem (CLT), that both samples (taken from the atmosphere and the model) converge to the same pdf (given they are the same which needs separately to be proven).

Response: We appreciate the explanation but we believe that the concerns found by the reviewer in this point are not relevant for this study.

3. Comparing remotely sensed and modelled concentrations: Moreover, the kernels to calculate averages (and used further on in inter-comparison exercises, see below) in remote sensing applications and in inverse modelling are strongly instrument and measurement-dependent (Rodgers, 2000). Fortunately, they often mask the above described effects due to their limited spatial or time resolution, i.e. their inherent averaging. In fact, in the latter applications these 'kernels' are called 'averaging kernels (AK)' of the observation and in colloquial English the averaging kernels can be called the 'glasses' by/through which the remote sensing observations were made. So the characteristics of 'the glasses' need to be considered in some way in inter-comparison exercises with modelled quantities (see below). For some examples of actual AKs, please inspect Figures 5 and 10 (for the weighting of the probed concentrations in the

horizontal) in Stutz et al., (2016), Figure 3.5 in Rodgers (2000), Figure 1 (below), or any other study on remote sensing. Chapter 3 in Rodgers, (2000) also discusses the different error sources of the traditional inversion methods used in remote sensing and inverse modelling. It also describes how remotely sensed quantities (here called co(i), where i is the retrieval grid number somehow representing the vertical resolution of the measurement) need to be compared with modelled results (cm(i)), i.e. by comparing the inferred co(i) with the product AK· cm(i), where AK is a tensor, of which the columns (or rows) a filled with the individual averaging kernels, displayed for example in Figure 1. In order to avoid these complications using traditional inversion methods for the interpretation of remotely sensed quantities (and in the particular case those arising from multiple scattering due to the a priori unknown spatial distribution and optical properties of aerosol and cloud particles), Stutz et al., (2016) describes a novel (scaling) method for the interpretation data. In effect, the scaling method uses additional information gained from simultaneously in-situ measured gases (i.e. O3) in order to assist the interpretation of remotely sensed NO2, and BrO in the TTL. Therefore, the scaling method has to be considered as a hybrid method (since it uses information collected by remote sensing and in-situ measurement), which comes with some advantages (and disadvantages) over traditional remote sensing methods. For example, it provides a higher accuracy than methods purely relying on remotely sensed information. Evidently the major disadvantage of the scaling method arises from the need of in-situ information of the probed air masses, i.e. it is suitable for applications from satellites, or high flying balloons. Further the scaling method still requires to carefully consider (by RT calculations simulating the observations) in order simulate how the information (the measured absorption) is obtained. Accordingly, when applying the scaling technique to their remotely sensed data, Stutz et al., (2016) and Werner et al., (2017) actually simulated each individual observation by modelling the actual RT (and the predicted absorption of the targeted species) by considering instrumental and other details of the measurements as well as predicted curtains of the targeted species, obtained from CTM modelling (TOMCAT/SLIMCAT). This approach (as in any traditional

remote sensing application) thus carries over to the analysis any relevant instrumental and observation-related features in the forward modelling of the observation. Evidently, the scaling method (as any traditional inversion method) then allows very close inter-comparisons of the predicted quantities (e.g. trace gas concentrations) with the observations, including a correct attribution of the fraction of the measured absorption (or slant column) to parts of the atmosphere not directly probed by the observation, however only if the averaging kernels are appropriately considered. Here please also note that the latter approach to inter-compare remotely sensed data and CTM modelling is not new at all, but e.g., it has been used by our group for more than 2 decades. Further using the scaling method, the calculation of absolute concentrations is achieved using a simultaneously in-situ measured and remotely sensed gas (e.g., O3), together with an appropriate consideration (by RT modelling) of the different sensitivities for detection of the targeted and scaling gas (see equation 14 in Stutz et al., 2016). In effect, the accuracy of the inferred quantities is arguably much better (Stutz et al., 2016) than only relying on remotely sensed quantities for the retrieval of concentrations. Accordingly check your statement on page 6 (lines 14 and 15) for correctness.

Response: we thank the valuable information the reviewer gives us about remote sensing techniques and the scaling method, but our manuscripts is not based on them. We are not using the CAM-Chem model to compare with any remote sensing data. The manuscript is based on the discrete measurements taken with in situs GWAS, which CAM-Chem reproduced very well.

II.) Comparison with available measured data Further, I'm really curious why the authors did not attempt to compare their modelling work with actual measured NO2 and BrO data (potentially) available to the first author for more than a year and which now have been published (Werner et al, 2017). However, when using remotely sensed data in inter-comparison exercises, the kernel for horizontal averaging (see Figure 10 in Stutz et al. (2016)) has to be appropriately taken into account for the modelled data (see my remark #3 above). Further, given that the Werner et al., (2017) manuscript

(which the first author of the present article co-authored) was submitted earlier (July 17, 2016) than the present manuscript (Nov. 18, 2016), the statement on page 4 (line 18) is not well based. By being more specific, the lack of a tight comparison of the modelled results with existing measured data give rise to some more deficits of the present study: 4. Simulated $NO_2$: For the Eastern Pacific TTL, the CAM-Chem model predicts $NO_2$ between 0.7 – 343 ppt at daytime (Table 1 and Figure 6). No reasons are provided for the elevated $NO_2$ in the TTL over the EP, except that modelled air masses are affected by 'pollution'. However, no other indication (neither from, for example, measured CO during NASA-ATTREX (UCATS) nor any further evidence inferred from the model) is provided that in fact polluted air masses were reaching the TTL over the EP in early 2013. In fact, the $NO_2$ mixing ratios reported by Werner et al., (2017) were < 20 ppt in the TTL, and they agree well (within the error bars +-/10 ppt) with the predictions of the TOMCAT/SLIMCAT simulations assuming no contribution from 'pollution'. In all these respects, and in particular with respect to the discussion provided above under point #3, the statement on page 6 (lines 14 and 15) is not well founded. Accordingly, in any further study information has to be provided why for the EP TTL the Cam- Chem model predicts $NO_2$ concentrations much large than observed. In addition, coherent evidences both from observations and modelling has to be provided (for example from CO/O3 and CH4/ O3) that indeed the TTL over the EP is affected by 'pollution'. 5. Simulated inorganic bromine, its partitioning and spatial patchiness: A major part of the study is devoted to model the bromine partitioning. First, I found it hard understand why the model does not really reproduce the increase in total inorganic bromine with increasing height (potential temperature) within the TTL, mainly caused by the destruction of brominated VSLS. This is somehow curious since the bromine concentrations at the lower boundary reported by Navarro et al., (2015) (page 3, line 18 and 19; VSLS; $3.84 \pm 0.64$ and $3.18 \pm 1.49$ ppt from WP and EP, respectively, and inorganic bromine $3.02 \pm 1.90$ ppt of Bry over the EP and $1.97 \pm 0.21$ ppt over WP) are in reasonable agreement with the data for the EP TTL, reported by Werner et al., (2017). Moreover, in the Cam-Chem model inorganic bromine (in gaseous form) barely increase from ~2

ppt (from the lower boundary at 14 km) to ∼3 ppt at 18 km (Figures 4 and 8), in stark contrast with the observations presented in the Werner et al., study for the upper levels of the TTL over EP. Here, depending on the flight, inorganic bromine ranges from (2.63 ± 1.04) ppt (range from 0.5 ppt to 5.25 ppt) to 5.1 ±1.57 ppt (at Θ = 390 - 400 K) to 6.74±1.79 ppt (at Θ > 400 K), in agreement with the measured destruction of brominated VSLS species (Navarro et al., 2015, and Figure 14 in Werner et al., 2017). So the obvious question is: Does the model either not efficiently destroy the brominated VSLS, and/or does the missing bromine reside in/on particles? If the latter is the case, the bromine up-taken by particles need to be rather large (2 – 3 ppt) in order close the bromine budget. So some information has to be provided how the bromine budget is closed in the model, and in particular on how much bromine is up-taken by the particles. Next even though the modelled absolute amount of gaseous inorganic bromine likely may not affect the Bry partitioning, the modelled [Br]/[BrO] (cited: (1) . . .. the modelled Br/BrO maximizes at 17 km from page 7, line 7 to 17 and in Figure 7 and (2) . . ...that Br/BrO may become as large as 2 in the TTL of the EP, see Figure 7) deviates from expectations based on the amount of ozone and its increase with height (see Figure 1 left panel, and Figure 3 to 8 in Werner et al., 2017), and the modelled bromine partitioning in the TTL as function of ozone (Fernandez et al., 2014, Figure 8). In fact, these findings largely contrast with early findings based on the Br/BrO ratio in TTL (at 17 km) during daytime c.f., by Fernandez et al., 2014 (Figure 1, left panel where Br/BrO < 0.6 at 17 km during tropical noon), Schmidt et al., (2016) (Figure 1), or lately the model results presented in Werner et al., (2017) (inspect Figure 3 – 8, Br/BrO < 0.6 at 17 km). Reasons for this discrepancy, including a discussion how the averaging of the ozone and the source gas concentrations and of other quantities impacts the modelled Br/BrO ratio (see points 1 and 2 above) certainly need to be addressed in any future study. Finally, the model predicts a certain patchiness (on spatial scales of some hundred kilometres) of the modelled Br/BrO ratio at 17 km for the EP (and WP), with [Br]/[BrO] ratios ranging from below < 0.5 to about 2. No further reason for this patchiness is provided in the manuscript. If air masses entrained by mesoscale convection

into the TTL are responsible for this patchiness, then it also needs to be seen in other gases (e.g. CO, CH4...), but again no evidence for this is provided in the manuscript. The predicted patchiness also contrasts with measured O3, NO2, and BrO, in particular since the remote sensing measurements can easily resolve horizontal variations of the measured quantities on the hundreds of kilometre scale (e.g., Stutz et al., (2016) figure 9, and Werner et al., (2017), figures 3 to 8). Further, since at daytime a rapid steady state is established between Br and BrO (see above) as function of the solar illumination and O3 concentration, it is difficult to infer from measured data any reason for the predicted patchiness in the [Br]/[BrO] ratio.

Response: As answered to reviewer 2 (see major comment 1) the manuscript has been modified to emphasize the fact that this model study was performed simultaneously to the study published by Navarro et al., 2015. We prefer not to go into further details about the issue of data availability and, although we really thank Prof. Pfeilsticker for his insightful comments, we would also like to think that the soundness of our modelling paper is not solely based on how well or not we compare to measurements and TOM-CAT/SLIMCAT model results in Werner et al., 2017, as it seems to transpire throughout this review. Nevertheless, and when relevant, we have also modified the manuscript to state how our results compare to the work of Wener et al., 2017. A final point, so far and to the best of our knowledge, the BrO and NO2 measurements from ATTREX 2014 (Western Pacific) are still being reviewed. Werner et al., 2017 report measurements of BrO and NO2 from ATTREX 2013, although that paper was still in discussion by the time of our submission.

6. Error and uncertainties: Finally, as an experimentalist who devotes 85 % of his efforts in the interpretation of data to get a handle on a reliable (thus justifiable) errors and uncertainties of the measured quantities, I always find it curious if studies lack a proper discussion of errors and uncertainties of the presented results. In modelling studies, this could for example be done by (1) inspecting respective Jacobians of the relevant quantities, (2) investigate differences in the modelled fields from 'on and off'

runs, and (c) perform ensemble runs et cetera. So also in this respect, the present study largely lacks this requirement for robust science.

Response: This is a good point - in this work, we are using the model and chemical mechanism employed in the Navarro et al., 2015 paper. This mechanism was already tested, tuned and validated for tropical vertical profiles of speciated Bry (Fernandez et al., 2014; Ordoñez et al., 2012). The details about the development of the chemical mechanism, all sensitivity tests performed to tune the model along with uncertainties estimation can be found in those previous works.

Summary Given the above described methodological deficits (points 1 and 2), the lacking comparison of the modelled results with actual measured data (points 4 and 5), and the lacking discussion of errors and uncertainties (point 6), unfortunately it is impossible to recommend the manuscript for publication in the present form.

Response: We have addressed point-by-point the reviewer's comments relevant to this work and we appreciate those other clarifications about comparing remote sensing data to an atmospheric 3D model, which are out of the scope of this paper.

Please also note the supplement to this comment:
http://www.atmos-chem-phys-discuss.net/acp-2016-1031/acp-2016-1031-AC1-supplement.pdf

---

## Author Comment (AC2) · 16 Mar 2017

Revised manuscript with changes in lines and supplementary figures has been attached as a supplement PDF file.

Review of: Modelling the Inorganic Bromine Partitioning in the Tropical Tropopause over the Pacific Ocean by Navarro et al.

General Remarks: This paper uses global model stimulations to examine the inorganic bromine (Bry) budget of the TTL, building on the work of Navarro et al. (2015). In that work, the authors (a) presented measured (and modelled) vertical profiles of brominated very short-lived substances (VSLS), such as CHBr3 and CH2Br2, from recent NASA ATTREX flights, and (b) used a model that reproduces the observations well (CAM-Chem), to estimate the contribution of VSLS to Bry in the TTL (highlighting the

significance of that contribution). In the present work, the same approach is adopted as above, though the focus is more on understanding the modelled Bry speciation in the TTL, the Bry diurnal cycle, and differences between the West and East Pacific (where the ATTREX missions sampled). The model results from this work show that BrO and Br are the most abundant daytime species, while BrCl and BrNO2 are more important at night. The authors also discuss differences in modelled Bry partitioning between the West and East Pacific, and briefly the sensitivity to heterogeneous processes on ice. Overall, this paper is an interesting case study that provides an (incremental) advance on our understanding of Bry partitioning in the TTL over the Pacific. In the absence of new BrO measurement data being included in the manuscript, this advance is somewhat subtle when viewed alongside the modelling study of Fernandez et al. (2014, ACP) that also used CAM-Chem to look at TTL bromine partitioning, in some detail. I have outlined three major areas below that should be addressed before publication.

Response: We thank the reviewer for the helpful comments and technical corrections. Below we address point-by-point all his/her comments and suggestions.

Major Comments: 1. The authors should ensure that the Introduction clearly sets out which of the broad model findings have come before, in order to help determine what the main motivation and purpose of this paper is. For example, the model results on zones where the Br/BrO ratio is >1 in the UTLS are interesting, though have been discussed previously by Fernandez et al. (2014, ACP) and Saiz-Lopez and Fernandez (2016, GRL). The same can be said about the analysis of the Bry diurnal cycle and Bry speciation in the TTL, and their sensitivity to heterogeneous processes. Is the advance here that this is simply a CAM-Chem case study for the ATTREX campaign period? If so, that is fine, but the measurements of BrO (and NO2) from ATTREX would very much strengthen the paper and help corroborate the modelled fields. In the first paragraph of Results and Discussion, it is noted that "BrO and NO2 measurements from the ATTREX mission were still under examination by the time of this analysis". Is this still the case? It strikes me that it is quite odd that these data are not included here.

Response: We have amended the text of the introduction to clarify that the main motivation and purpose of this paper is to model the inorganic bromine partitioning derived from the different flights during the ATTREX campaign. A paragraph has been added to page 2 line 33 and now it reads: "Our study manly focuses on the difference in modelled Bry concentrations in the TTL over the Pacific throughout the ATTREX campaign flight tracks, and examines its temporal and spatial distributions. "Based on the reliable representation of the observed VSLorg by the CAM-Chem model on the study of Navarro et al., 2015, and as a follow up of this investigation regarding the chemistry of bromine tracers in the TTL, we estimated the partitioning of Bry over the tropical eastern and western Pacific during 2013 and 2014, respectively." "From this case study analysis, we also complement the finding of the diurnal Bry speciation in the TTL, and the Br/BrO ratio distribution in the Upper Troposphere-Lower Stratosphere (UTLS) found by Fernandez et al., 2014 and Saiz Lopez and Fernandez, 2016." Regarding to the measurements of BrO (and NO2) from ATRREX, we agree with the reviewer that it would strengthen the paper and help corroborate the model. However, this model study was running simultaneously with the study published by Navarro et al., 2015, when measurements of BrO and NO2 were still under examination, as we stated in Results and Discussion section. We clarify this point by adding the following sentence at the beginning of the Results section (page 5 line 28), which now read: "This modelling study was carried out simultaneously with the work published by Navarro et al., 2015. Only ozone and VSLorg abundances were available to validate model performance as BrO and NO2 measurements from the ATTREX mission, now published by Werner et al., (2017), were still under examination by the time of this analysis. Thus, once the model performance during ATTREX campaign is evaluated in Sect. 3.1, we step into a CAM-Chem modelling case study oriented to determine the Bry partitioning (Sect. 3.2) and efficiency of heterogeneous recycling reactions (Sect- 3.3) on the mostly unexplored eastern and western pacific." To the day, the BrO and NO2 measurements from ATTREX 2014 (Western Pacific) are still under review. Measurements of BrO and NO2 from ATTREX 2013 were published on the work of Werner et al., 2017,

but were not used in our study as the other manuscript was still under discussion by the time of our submission. However, our NO2 and BrO estimations are within the ranges observed by the measurements of Werner et al., manuscript. An additional statement have being added to our manuscript on page 7 line 5 and page 7 line 34 to clarify this information. The text now reads: "Our mean vertical distributions for the EP are in the lower edge of the reported ranges of Werner et al. (2017), who reported a measured range for BrO between $0.5 \pm 0.5$ ppt at the bottom of the TTL and about 5 ppt at $\theta = 400$ K, consistent with an inferred increase of Bry from a mean of $2.63 \pm 1.04$ ppt to $5.11 \pm 1.57$ ppt as we move upward in the TTL." "Our average range of NO2 mixing ratios is approximately $15 \pm 6$ ppt at 14 km, with slightly higher values over the tropopause, $22 \pm 24$ ppt at 17 km. These estimates within 1 standard deviation agree with the NO2 values presented by Stutz et al. (2016) and Werner et al. (2017) from observations made during ATTREX 2013 over the EP. As they report in their manuscript, their O3 scaling technique allowed retrieval of NO2 concentrations of $15 \pm 15$ ppt in the TTL, and a range of 70 up to 170 ppt in the mid-latitude lower stratosphere."

2. The most novel aspect of this work is the examination of differences between the W and E Pacific. The discussion of chlorine could be improved in this regard. If differences in Cly between the two regions can impact local Bry partitioning, some discussion on how well constrained the actual Cly simulation over the WP (average up to 84 ppt Cly in daylight) and EP (up to 181 ppt Cly in daylight) is needed. At the very least some more details of the chlorine simulation could be given. More broadly, I would suggest that the title of the paper should reflect that the emphasis of the paper is on the differences between W and E Pacific. Response: We appreciate the reviewer for highlighting this aspect, which has now been strengthened the revised manuscript. Although the sensitivity simulation described in Section 3.3 was introduced to highlight how the different atmospheric conditions between the EP and WP were affecting the bromine partitioning, the original manuscript mainly focused on the changes due to the high/low NOx regime prevailing in each region. The impact of Cly chemistry is maximized during the night, as the abundance of reservoir bromine species (i.e. BrONO2,

HOBr, HBr) maximizes after down, and in the presence of ice-crystals those species can react with HCl (which is the dominant Cly species throughout the troposphere). Thus, the following heterogeneous reacting sequence is amplified in the presence of large Surface Area Densities (SAD) in the TTL. BrONO2 –> HOBr + HNO3 HOCl + HBr –> BrCl + H2O HOBr + HCl –> BrCl + H2O In order to highlight the large impact that the heterogeneous recycling occurring on ice-crystal has on the nighttime partitioning, as well as validate the Cly abundance in CAM-Chem, we introduced the following sentences. Please note that additional information regarding chlorine chemistry is also given below in the answers to the general comments. Page 7 lines 14: "It is worth noting that even when the maximum inorganic chlorine levels are larger in the EP, BrCl is not the dominant night‐time reservoir, while in the WP, where BrCl dominates, maximum Cly mixing ratio is almost half the value found in the EP (see Table 1). Considering all flights, the maximum Cly abundances are < 85 pptv in the WP and < 182 for the EP, with a global mean tropical annual Cly mixing ratio of 50 pptv in agreement with previous reports (Marcy et al., 2004; Mébarki et al., 2010). This can be explained considering the faster vertical transport occurring in the western pacific region, which decreases the photochemical decomposition of VSL chlorocarbons (Saiz Lopez and Fernandez, 2016)." Page 8 lines 14: "Note that the differences in Cly abundance can reach factors as much as 5 times larger for the EP if the independent flights are considered (e.g., max. Cly ∼500 ppt for RF01, RF03 and RF04 performed in the EP during ATTREX 2013, while max. Cly for all flights except RF07 (< 400 pptv) remain below 100 ppt. However, the night time BrCl abundance is larger in the WP, representing more than 90% of the nighttime Bry partitioning for flights RF02 and RF04 (see Figure S1 & S2 in the Supplement). For these cases, BrCl mixing ratios between 1 and 2 pptv are formed within air parcels with a very low Cly abundance (of the order of 10 ppt). In order to understand this unexpected behavior, we performed a sensitivity simulation neglecting the inter halogen heterogeneous recycling occurring on upper tropospheric ice crystals, see Sect 3.3 below." Page 10 line 5: "Heterogeneous recycling reactions of reservoir species on ice crystals are relevant at UTLS levels, thus, a sensitivity test

was carried out to determine the influence of water–ice aerosols on the distribution of the inorganic species. Equations (1) to (6) shows the chlorine, bromine and inter halogen tropospheric heterogeneous reactions occurring on ice crystals (for a complete description of the implementation of heterogeneous reactions in CAM Chem, see Table S1 in supplementary online material of Fernandez et al., 2014). $BrONO_2$ –> $HOBr + HNO_3$ (1) $ClONO_2$ –> $HOCl + HNO_3$ (2) $HOCl + HCl$ –> $Cl_2 + H_2O$ (3) $HOCl + HBr$ –> $BrCl + H_2O$ (4) $HOBr + HCl$ –> $BrCl + H_2O$ (5) $HOBr + HBr$ –> $Br_2 + H_2O$ (6)" Page 10 line 28: "Thus, neglecting ice recycling reactions (1) to (6) suspend the heterogeneous conversion of $BrONO_2$ to BrCl, and gas–phase bromine nitrate (which is formed mainly by the termolecular reaction of $BrO + NO_2 + M$ during twilight) remain as the dominant $Br_y$ species during the night both under a high $NO_x$ regime (i.e., within the EP region) as well as under the low $NO_x$ regime (western pacific). But when the heterogeneous recycling reactions are activated, the model output predicts that the recycling efficiency depends mostly on the total surface area density of ice crystals in the upper troposphere (SAD ICE): even under very low $Cl_y$ concentrations (between 10 and 20 ppt), if SAD ICE is present in the TTL, the nighttime partitioning is displaced in favour to BrCl. Fernandez et al., (2014) found tropospheric SAD ICE levels within the western pacific upper TTL to be the largest of the whole tropical region, suggesting that BrCl abundance should be maximized in this region of the pacific." Finally, following the advice of the reviewer we change the title of the manuscript to emphasize the difference between W and E pacific. The title now reads: "Modelling the Inorganic Bromine Partitioning in the Tropical Tropopause over the Eastern and Western Pacific Ocean"

3. The writing is quite awkward in many places and the paper would benefit from a very thorough check/read through. In addition, although the paper is compact, it would benefit from some sub-headings, particularly in the Results and Discussion section (e.g. Model-measurement O3 comparison, Diurnal cycle in Bry partitioning, or something similar). Response: Following the advice of the reviewer, we have added some sub-headings to the results and discussion section. The text is now separated in the

following sections: 3.1 CAM-Chem model evaluation 3.2 Bry partitioning 3.2.1 Tropical ring of atomic Br: indications from this case study 3.3 Heterogeneous reactions: impact of water-ice recycling on Bry speciation/distribution

Other points: *Abstract* Sometimes the "E" in Eastern (Pacific) and the "W" in Western (Pacific) are capitalized and sometimes they are not. Please be consistent throughout manuscript (including in figures and captions). This has been corrected in the manuscript. *Introduction* P1, L33: "Bry" is defined early on in the manuscript but "inorganic bromine" is used in numerous places after that. I suggest changing the latter to the former where appropriate throughout the manuscript. Change has been made, although we kept "inorganic bromine" at a few places to facilitate the reading of congested sentences. P3, L2: Struck me that introducing the "proposed tropical ring of atomic bromine" here is odd. Could you not make mention of these papers earlier in the Introduction? A statement has been added to the introduction on page 2 lin 27. Now the text reads: "This study also introduced the concept of the "tropical ring of atomic bromine", a photochemical phenomenon that extends in the tropics from approximately 15 to 19 km where the abundance of Br atoms is favoured due to low temperatures (<200K) and low O3 abundances(<100 ppb)." P3, L3: "section 3" — "Section 3" Change has been made *Methods* I would separate out Methods into 2.1 Observations and 2.2 Modelling. This section seems quite unstructured in its present form. The Modelling section needs more details as to the ozone precursor emissions that were used in CAM-Chem + some brief information of the chlorine simulation. Change has been made. The methods section has been divided in section 2.1 Observations and 2.2 Modelling. In addition section 2.1 has been separated in 3 other sub-sections to improve the format and structure of the manuscript. Section 2.1 now includes: 2.1.1 ATTREX campaign, where we have added in the paragraphs of P5, L7 as it was suggested in your last comment. 2.1.2 VSLorg Observations, which briefly described the GWAS methodology explained on our previous publication: Navarro et. al., 2015. 2.1.3 O3 Observations, which described the O3 methodology. In addition, we extended the description of the ozone precursor inventory and the bromine and

chlorine emissions as follow (page 5 line 9): "The current setup is based on the bromo-carbon emission inventory of Ordoñez et al., (2012), which includes time dependent geographically distributed sources of CHBr3, CH2Br2, CH2BrCl, CHBr2Cl, CHBrCl2 and CH2IBr. Even when we do not consider here other chloro carbon sources like CH2Cl2 and C2Cl4, those species live long enough to be injected almost entirely as source gases to the stratosphere and do not contribute to the tropospheric inorganic chlorine (Cly) loading [Hossaini et al., 2015]. Additional Bry and Cly sources from sea‐salt heterogeneous dehalogenation in the lower troposphere are parameter-ized (Ordoñez et al., 2012; Fernandez et al, 2014). Prescribed surface volume mixing ratios of long lived chlorofluorocarbons (CFCs) and halons, as well as surface concen-tration of anthropogenic CO2, CH4, N2O and other ozone precursors are based on the long‐lived inventory of Meinshausen et al. (2011)." P3, L12: Has "very short-lived organic substances" not already been defined as VSLorg? Also, as the focus of this work is on VSLS, it would be better if some of the gases listed (e.g. CHBr3 etc.) are actually referred to earlier in the Introduction (maybe around line 31). Changes have been made, and now page 1 line 31 reads: "Many of these discuss the contribution of brominated very short‐lived organic substances (VSLorg) like bromoform (CHBr3), dibromomethane (CH2Br2) and/or bromochlorocarbons such as (CH2BrCl, CHBr2Cl, CHBrCl2, etc.), in addition to long‐lived halons and methyl bromide (CH3Br), as an important source of stratospheric bromine." P3, L17: The two sentences beginning "At the tropopause level" seems out of place. Is this not motivation/background for the present study and should it not appear in the Introduction? We believe that the phrase "At the tropopause level" may still be correct. As we mentioned before, this study is based on the previous publication of Navarro et. al., 2015 and the model results of the organic and inorganic bromine composition were found simultaneously. The study of Navarro et al., 2015 showed the organic bromine composition at the tropopause level (~17km). To clarify this point, we added a sentence on the introduction page 2 line 34 which now reads: "Based on the reliable representation of the observed VSLorg by the CAM ‐Chem model on the study of Navarro et al., 2015, and as a follow up of

this investigation regarding the chemistry of bromine tracers in the TTL, we estimated the partitioning of Bry over the tropical eastern and western Pacific during 2013 and 2014, respectively". P4, L4: CAM-Chem has already been defined. Change has been made P4, L21: "for both, the WP and EP respectively" — "for both the WP and EP". Change has been made P5, L7, Sentence beginning "During ATTREX" to the end of the paragraph. This text describes the different sampling times/paths of the observations/flights and would be better placed in Section 2. In the current location it disrupts the flow of results. Similarly, consider moving Figure 3 to the Measurements section. Changes have been made

References: Fernandez, R., Salawitch, R., Kinnison, D., Lamarque, J.-F., and Saiz-Lopez, A.: Bromine partitioning in the tropical tropopause layer: implications for stratospheric injection, Atmospheric Chemistry and Physics, 14, 13391-13410, 2014. Navarro, M. A., Atlas, E. L., Saiz-Lopez, A., Rodriguez-Lloveras, X., Kinnison, D. E., Lamarque, J.-F., Tilmes, S., Filus, M., Harris, N. R., and Meneguz, E.: Airborne measurements of organic bromine compounds in the Pacific tropical tropopause layer, Proceedings of the National Academy of Sciences, 112, 13789-13793, 2015. Saiz‐Lopez, A., and Fernandez, R. P.: On the formation of tropical rings of atomic halogens: Causes and implications, Geophysical Research Letters, 43, 2928-2935, 2016. Werner, B., Stutz, J., Spolaor, M., Scalone, L., Raecke, R., Festa, J., Colosimo, F., Cheung, R., Tsai, C., R.Hossaini, Chipperfield, M. P., Taverna, G. S., Feng, W., Elkins, J. W., Fahey, D. W., Gao, R.-S., Hintsa, E. J., Thornberry, T. D., Moore, F. L., Navarro, M. A., Atlas, E., Daube, B., Pittman, J., Wofsy, S., and Pfeilsticker, K.: Probing the subtropical lowermost stratosphere, tropical upper troposphere, and tropopause layer for inorganic bromine, Atmos. Chem. Phys, submitted, 2016.

Please also note the supplement to this comment:
http://www.atmos-chem-phys-discuss.net/acp-2016-1031/acp-2016-1031-AC2-supplement.pdf

[Figure]

[Figure]

**Supplement:**

[revised manuscript text omitted]

---

## Referee Report (RR1)

**Review for manuscript acp-2016-031**

The manuscript of Navarro et al., reports on Cam-Chem (Community Atmosphere Model with Chemistry) modelling of the partitioning of inorganic bromine in the tropical tropopause layer (TTL) for NASA ATTREX deployments of the Global Hawk over the eastern and western Pacific Ocean in 2013 and 2014. From the previous version the manuscript is now somewhat complemented including information on the heterogeneous processing of bromine and chlorine on ice particles. Due to the more vigorous deep convection and hence colder temperatures in the TTL and sustaining a larger occurrence of cirrus ice in the TTL of western as compared to the eastern Pacific, heterogeneous processes could indeed reveal some differences in the photochemistry in both areas. Including details of the heterogeneous processing and the irreversible removal of chlorine and bromine from the TTL by sedimentating particles may cover a novel aspect in the study, and as such could render the present study even more valuable to the readers of ACP.

Unfortunately, reactions to my previous comments (mostly, but not all related to how measurements and model predictions need to be compared in state-of-the-art studies) are mostly missing in the present manuscript due to any reason. In order to make it again clear, a state-of-the-art inter-comparison of measurements and model predictions would either involve (a) a comparison of the forward-simulated measurements including their specific features (here briefly called one-to-one inter-comparison), or (b) by using a Bayesian approach a comparison of measured and modelled probability density distributions (and their moments) of the targeted quantities. Both approaches have been widely used in past studies, and it is known by not so doing the information contained in measurements cannot be properly exploited, or is even misinterpreted (see minor comment 2). So it is up to the authors to provide a tighter inter-comparison of the various measurements with the model predictions than in the past and present a version of the manuscript which goes beyond what is already known about bromine in the TTL. Alternatively, the authors may mainly focus on modelling, but then a larger number of sensitivity runs on the relevant processes (c.f. heterogeneous activation and irreversible removal of the halogen and their relevant parameters) need to be presented.

While I tried to review the present manuscript as independently as possible from the previous version, I'm recalling here my major comments however in a less abstract but in a more illustrative way than previously put forward (see below). So once the authors will have decided on the focus of study/manuscript, they may either fully include my comments when it addresses measurement/model inter-comparison, or in case they focus their study on mostly modelling then they may circumvent them, while providing instead a series of sensitivity runs on the relevant parameters for their study.

Beside these more general comments, rather than only providing some major comments as done in my previous review, I did a more comprehensive review by adding a section called 'minor comments' which lists typos, and minor issues to be clarified as well as a section 'list of references' of which the references are worthwhile to be included in any revised version.

**Major comments:**

1. Comparison with measured data and 'averaging' (c.f. on page 5, line 3 and 4; These continuous measurements of ozone were averaged to match the location where GWAS samples were collected, and then compared to CAM-Chem outputs.):
   By referring to my comments 1 and 2 of the previous review, how was the averaging performed and inferred from what pdfs?
   In order to make my comment 2 (and in part 1) more clear my question points to the following: Since the sampling time of the NOAA-2 polarized O3 photometer is 1 Hz (corresponding to a column of air of about 180 m in length) but the CAM-Chem has a spatial resolution of 1° (longitude) x 1° (latitude) (page 5, line 19) or about 110 km x 110 km, how was the measured O3 averaged, and how were the higher moments of the pdf calculated and how do these quantities compare

with corresponding quantities calculated from the CAM-Chem simulation c.f. how were the errors bars in your Figure 2 calculated?

A similar question related to the averaging may arise when comparing the trace gases measured in AWAS air samples (where the sampling time was about 25 s at 14 km and 90 s at 18 km, corresponding to columns of air of being 4.5 km and 16.1 km long, respectively), and the model. Further since one can reasonably assume (e.g., when assuming a constant pumping rate), that the concentrations measured by the individual in-situ instruments are spatial averages of the probed air columns, it is not so clear how these statistical measures calculated from observed quantities compare to those (c.f. the average and the variance) calculated from modelled data? Noteworthy is here that the inferred mean/variances are obviously different in the measured and modelled ozone (see your Figure 2), while any respective information on the variances of modelled and measured brominated source gases is still missing in Figure 3.

Remark: The question related to the appropriate 'averaging' of geophysical parameters is largely justified since the concentrations of stratospheric trace gases and tracers may show long-range correlations which are likely different in the measurements than portrayed in the model (for details see Van Leuwen, 2009, and the references provided in my previous review). Moreover, quantities having long-range correlations (as they occur in geophysical fluids), the central limit theorem teaches (and mandates) that the inferred statistical quantities are scale-dependent and that accordingly the quantities (here concentrations) are in general not Gaussian distributed (for the refs see my previous review). Accordingly, the calculation of the relevant statistical quantities requires the knowledge of the underlying (non-Gaussian) pdf, which when known could alternatively to a one-to-one inter-comparison (as c.f., done in Werner et al., 2017) be used in a Bayesian approach in the inter-comparison (e.g., Van Leuwen, 2009, and the references provided earlier).

So some explanation (beyond the word 'average') is needed how the discussed quantities (ozone and the tracers) are inter-compared e.g., and how the mean and uncertainty/error bars in Figures 2 and 3 were calculated. (Comment: Here you may also anticipate the problem of inter-comparing remotely sensed quantities with modelled quantities since the former are usually not uniformly sampled over an air column like samples provided by in-situ measurements but via an observation dependent averaging kernel, see my remark/comment I. 3 in the previous review and in consequence the minor comment 2 below).

Finally, because the underlying statistic of the involved concentrations is non-Gaussian, care has to be taken that derivatives of inferred quantities, c.f. ratios as used to calculate the [Br]/[BrO] ratio, are not biased (see my comment 1, and comment II. 5, second paragraph in my previous review).

2. Model constraints and boundary conditions: Further if the manuscript attempts to provide novel insights into model's predictive skills on the budget and photochemistry of the considered species within the TTL, more information is needed than provided on how the model is constrained and on how the boundary conditions (for the relevant gases) are chosen.

Being more specific, it is unclear how Cly is constrained (or calculated) in the model due to the provided contradictory information. For example, on page 7, line 16 to 18, it is said that 'Considering all flights, the maximum Cly abundances are < 85 pptv in the WP and < 182 (????) for the EP, with a global mean tropical annual Cly mixing ratio of 50 pptv in agreement with previous reports (Marcy et al., 2004; Mébarki et al., 2010).' while in Table 1 a range of Cly (day) 1 - 515 pptv (WP) and 1 - 969 pptv (EP) is mentioned. Further, in order for reader to judge the predictive skills of the model, the results need to be compared with previous Cly observations (and eventually modelling) in the TTL beyond the Marcy et al., 2004 study actually mentioned in the manuscript, e.g., Marcy et al., 2007; Mébarki et al., 2010; von Hobe et al., 2011; Jurkat et al., 2014, and many others.

Here, similar than previously asked for (and again asked for here) the budget and partitioning of bromine (comment II., #5, first paragraph in the previous review), it would be worthwhile to

provide information on the altitude dependent partitioning of Cly (including chlorine the condensed phase) and how the budget is closed with respect to the total organic chlorine.

Next, since you state (c.f., page 7, line 34; 'The scenario over the EP is slightly different (from the WP) as levels of NO2 and O3 define a high NOx regime'), information on the implemented sources for NOx in the model is completely missing. Providing such information was already asked for in my previous review (comment II. 4.), primarily since measured and as well as the modelled NO2 in Werner et al. (2017) does not support the predicted 'high NOx regimes for the EP. In this context, your statement on Page 8, line 7 is not appropriate in this context (see my minor comment 2, and my remark 3 in the previous review). In consequence you have to face the findings of Werner et al., (2017) regarding NO2, i.e. measured and SLIMCAT modelled NO2 does not support the Cam-Chem prediction of a 'high NOx regime' in the TTL of the EP.

**Minor comments:**

1. Page 1, line 30 cont.: Your statement on the stratospheric sources of bromine does not include inorganic bromine being transported from the troposphere into the TTL and LS, for which previous experimental studies provided quite some evidences, c.f., Dorf et al., 2008, Laube et al. 2008; Brinckmann et al., 2012; Schmidt et al., 2016; Werner et al., 2017, and others. So please clarify.

2. Page 8, line 7: However, previous studies have shown large associated uncertainties in TTL NO2 measurements based on remote sensing instruments (Weidner et al., 2005; Butz et al., 2006; Bauer et al., 2012).

   This sentence demonstrates that the authors did not appreciate (or even understand) my comment 3 outlined in length in the previous review (see also comment 1 above).

   So recall my argument now expressed in more simple terms. First all three cited studies refer to measurements where the sensor was not deployed within the air mass of interest, but NO2 (and some other species) were measured somehow remotely (from a balloon or even satellite). Now in order to see the difference of the previous studies with the study of Werner et al., (2017) just make a drawing to compare the different observation geometries i.e. compare the line of sights for sensors looking (from a great distance) slant through the layer of interest (the former three studies) and those inspecting NO2 (horizontally) along the layer of interest (Werner et al., 2017, and inspect Figure 5 in Stutz et al., (2017). Then compare the different path lengths over which the skylight is absorbed and how they differ in length, since the path length is one decisive parameter in optical absorption measurements which determines the detection sensitivity. For the exercise you can fairly assume that all sensors have the same detection limit (in terms of optical density for NO2) since they all operate at the photon electron shot noise and receive about the same skylight radiance.

   Second, since the bulk of stratospheric NOx is located somewhere at around 30 km, sensors (satellites and balloons) which attempt to measure the comparable lower amounts of NO2 located below (or behind) this NO2 layer, i.e. NO2 in the UT/LS and TTL, are more affected by any NO2 changes within this high-located NO2 layer than a comparable sensors measuring below and horizontally along the layer of interest (c.f. on the Global Hawk). So as argued in my previous review, in remote sensing you cannot compare a detection limit inferred for a specific measurement with another measurement without considering the individual observation geometries (to which the remote sensors refer to averaging kernels), even when assuming the same instrument is used.

   By summarizing there are two observation-geometry related advantages of remote-sensing instruments being deployed within the layer of interest over those inspecting the studied air masses from 'far away'. In consequence, you have to appropriately considered the NO2 (and BrO) measurements, their detection limits and errors explained in length in Stutz et al., (2017), rather by assigning to them uncertainties which suite best to your study.

3. Page 8, line 10: These results are in good agreement with the partitioning of Bry found by (Werner et al., 2017) where BrO is the daylight dominant species over EP. In fact, your results and

conclusion (page 11, line 14 – 20) are suggesting something else. Accordingly, you need to reconsider this statement.

4. Page 9, line 29: These results are in good agreement with the statements of Fernandez et al. (2014), which suggested Br/BrO > 1 during strong convective periods over the WP warm pool region.
   This finding is not surprising given that the same model (with similar/same model parameters and inputs) were used. So the statement does not add new evidences on Br/BrO > 1 in the TTL et cetera.

5. Page 10, lines 20 to 22: … the absence of ice-crystal reactions increases the total inorganic fraction by 7% and 12% over the EP during the day and night, respectively. ….
   I see no specific reason why ice-crystals should change the amount of Bry (or Cly), except that Bry (or Cly) is heterogeneously removed from the gas phase, which (if the case) you should then mention and quantify in the manuscript. Again I emphasize (see comment 5, first paragraph) to show in a separate figure how (a) Bry (and Cly) is partitioned among all (organic and inorganic) gaseous bromine (and chlorinated) species (b) the fractions being up-taken by particle, and (c) the fraction being permanently removed by sedimentation (comment II.) #5, first paragraph in the previous review).

6. Page 11, lines 14 to 20: Reactive species like atomic Br become the dominant Bry species in large regions of the TTL during daylight, following the large variation of ozone abundance within these regions strongly influenced by deep convection……
   While I see motives for this statement when air masses are strongly influences by convection in in the TTL of the WP (and much less for the EP due to in general higher ozone there), how does the result relate to results of previous theoretical studies (beyond those of Fernandez et al., 2014)?

7. Page 11, lines 21 to 22: Why the contribution of inorganic bromine directly injected into the TTL is omitted here, e.g. Schmidt et al., 2016, Werner et al., 2017 and others

8. Page 11, line 25: …. to diminish the uncertainty of the amount of Bry that reaches the stratosphere, and properly constraint the global bromine budget.
   In fact, previous studies indicated that the amount of bromine in the stratosphere is less uncertain than how and in what form (i.e. the fraction of organic and inorganic) it is transported into stratosphere. Here you need to cite at least WMO 2014, and if you like to provide an informed list of bromine-related measurements, however only those performed tropical UT/TTL/LS, you need to cite the studies of Schauffler et al., 1993, 1998, and 1999; Dorf et al., 2008; Laube et al., 2008; Brinckmann et al., 2012; Sala et al., 2014; Wang et al., 2015; Werner et al., 2017; Stutz et al., 2017 and others.

**Typos and necessary clarifications:**

1. Page 2, line 30: ….. showed approximately 3 to 5 ppt for potential temperatures between 350 and 400 K in the TTL …change to…. showed approximately 3 to 5 ppt Bry for potential temperatures between 350 and 400 K in the TTL

2. Page 2, line 33: Our study manly focuses …change to … Our study mainly focuses

3. Page 4, lines 6 and 7: … with 90 custome-made stainless change to …. with 90 custom-made stainless…and again from … a custome inlet at 2 to 8 liters .. to .. a custom inlet at 2 to 8 liters

4. Page 4, line 31: …. the modelling estimates of the organic bromine fractions were similar for the entire Pacific (3.84 ± 0.64 and 3.18 ± 1.49 ppt from WP and EP, respectively). I guess here 'bromine tied in very short lived species' is meant?

5. Page 6, line 29: Over the EP, BrONO2 dominates the entire range of altitude from 14 to 18 km. Since this statement is certainly only correct for the inorganic bromine partitioning at night, the sentence should accordingly read. Over the EP, BrONO2 dominates the entire range of altitude from 14 to 18 km at night.

6. Page 6, line 29: The total Bry burden during daylight hours increases from … change to ..

7. During daylight hours, Bry increases from….

8. Page 7, line 8: …. of 2.63 ± 1.04 ppt to 5.11 ± 1.57 ppt as we move upward in the TTL …delete … as we move upward in the TTL.

9. Page 7, line 15: is almost half the value found in the EP ... change to ….. is almost half the concentration predicted for the EP (since a model predicts rather than finds (measures) something).

10. Page 7, line 16: ……………are < 85 pptv in the WP and < 182 for … change to … are < 85 pptv in the WP and < 182 pptv

11. Page 7, line 19 and elsewhere in the manuscript: … in the western pacific region, … change to … in the western Pacific ... since Pacific is a name and hence needs to be written with a capital letter.

12. Page 7, line 23: Figure 5 compares the mean abundances observed ….. change to Figure 5 compares the mean abundances of O3 (measured?), NO2 (modelled), and Cly (modelled)… since your study mostly reports on model results.

13. Page 7, line 28: As the SZA keeps on increasing, a decrease on photolysis as well as ozone concentrations… change to .... As the SZA is increasing, a decrease in the photolysis as well as ozone concentrations… and delete … as well as ozone concentrations…since there is no reasons why ozone concentrations should decrease with SZA.

14. Page 7, line 34: The scenario over the EP is slightly different as levels of NO2 and O3 define a high NOx regime … change to .. The scenario for the EP is slightly different from the WP as concentrations of NO2 and O3 define a high NOx regime there.

15. Page 8, line 2, and elsewhere in the text: Stutz et al. (2016) change to … Stutz et al. (2017)

16. Page 8, line 16: Note that the differences …change to … Note that the predicted differences

17. Page 8 line 16: the EP if the independent flights…change to … the EP if individual flights …

18. Page 8, line 28: Note, however, that atomic Br abundances surpass BrO mixing ratios at low SZA (close to noontime) and low ozone abundances (below 100 ppb, Fig. 5b and 5d) …. change to … Note, however, that modelled atomic Br abundances surpass BrO mixing ratios at low SZA (close to noontime) and low ozone abundances (below 100 ppb, Fig. 5b and 5d).

19. Page 9, line 2: In the EP, Br surpasses BrO mixing ratios at 60º SZA for flights RF04 and RF06, but …. change to … In the EP, modelled Br surpasses BrO mixing ratios at 60º SZA for flights RF04 and RF06, but

20. Page 9, line 8: …. which focused on ATTREX measurements taken exclusively over the EP and used an O3-scaling technique to retrieve their results, our model calculations support the fact that …change to .. which focused on ATTREX measurements taken over the EP, our model calculations indicate Br/BrO ratios (erase 1. exclusively since SF2 and SF5 lead to central Pacific), 2. erase … used an O3-scaling technique, since it falls of the context here, and 3. our model calculations support .. erase the fact that .. since a model can never produce facts but only more or less good predictions et cetera

21. Page 9, line 16: Over the EP, Br/BrO > 1 are observed as discrete masses, particularly at SZA between 40°and local noon … change to … Over the EP, Br/BrO > 1 are predicted (you did not observe it) in distinct air masses, particularly at SZA between 40°and local noon. …..by the way a result not supported by the results of the Werner et al., 2017 study.

22. Page 10, line 6: …on the distribution of the inorganic species .. change to .. on the partitioning of the inorganic species

23. Page 10, line 24: As explained by Aschmann et al. (2011) the increment in the amount of HBr at high altitude levels could be due to a slowly sedimentation following by evaporation as the adsorbed HBr is not washed out right away.

24. This sentence (as it is) does not really make sense to me, since if HBr is taken-up by particles it should reduce Bry at the condensing altitudes and upon evaporation (of the particles) should release HBr at lower altitudes (and not vice versa). So accordingly correct the sentence.

25. Page 10, line 33: (SAD-ICE) explain the acronym by words.

26. Page 11, line 10: …while BrCl and BrONO2 were found as the night-time dominant species over the WP and EP, respectively. …change to … while BrCl and BrONO2 were predicted to dominate TTL Bry at night-time.

27. Legend Figure 5: 'Average of inorganic bromine species (top panel) and main reactants of the inorganic chemistry (bottom panel) using the entire range of altitudes (14 to 18 km) over the

western Pacific (a and b) and eastern Pacific (c and d). Black boxes indicate the percentage of the dominant Bry species for day and night at 17 km.'

First I'm not sure what can be learned from 'averages' for the altitude range 14 – 18 km, when obviously information on the profiles (from the measurements and the model) of the shown quantities is available. Second, I really wonder why these 'averages' somewhat 'oscillate', actually more for the dark than the sun-lit hours. Third, I wonder whether these oscillations are also seen in the measured source gases?

**Additional references:**

Brinckmann, S., Engel, A., Bönisch, H., Quack, B., and Atlas, E.: Short-lived brominated hydrocarbons – observations in the source regions and the tropical tropopause layer, Atmos. Chem. Phys., 12, 1213–1228, doi:10.5194/acp-12-1213-2012, 2012.

Dorf, M., Butz, A., Camy-Peyret, C., Chipperfield, M. P., Kritten, L., and Pfeilsticker, K.: Bromine in the tropical troposphere and stratosphere as derived from balloon-borne BrO observations, Atmos. Chem. Phys., 8, 7265–7271, doi:10.5194/acp-8-7265-2008, 2008.

Jurkat, T., et al. , A quantitative analysis of stratospheric HCl, HNO3, and O3 in the tropopause region near the subtropical jet, *Geophys. Res. Lett.*, *41*, 3315–3321, doi:10.1002/2013GL059159, 2014

Laube, J. C., Engel, A., Bönisch, H., Möbius, T., Worton, D. R., Sturges, W. T., Grunow, K., and Schmidt, U.: Contribution of very short-lived organic substances to stratospheric chlorine and bromine in the tropics –a case study, Atmos. Chem. Phys., 8, 7325–7334, doi:10.5194/acp-8-7325-2008, 2008.

Hossaini, R., Chipperfield, M. P., Feng, W., Breider, T. J., Atlas, E., Montzka, S. A., Miller, B. R., Moore, F., and Elkins, J.: The contribution of natural and anthropogenic very short–lived species to stratospheric bromine, Atmos. Chem. Phys., 12, 371 - 380, doi:10.5194/acp-12-371-2012, 2012.

Marcy T. P. et al., Measurements of trace gases in the tropical tropopause layer, Atmospheric Environment 41, 34, 7253–7261, 2007.

Mébarki, Y., Catoire, V., Huret, N., Berthet, G., Robert, C., and Poulet, G.: More evidence for very short-lived substance contribution to stratospheric chlorine inferred from HCl balloon-borne in situ measurements in the tropics, Atmos. Chem. Phys., 10, 397-409, doi:10.5194/acp-10-397-2010, 2010.

Schauffler, S. M., Heidt, L. E., Pollock,W. H., Gilpin, T. M., Vedder, J. F., Solomon, S., Lueb, R. A., and Atlas, E. L.: Measurements of halogenated organic compounds near the tropical tropopause, Geophys. Res. Lett., 20, 2567–2570, doi:10.1029/93GL02840, 1993.

Schauffler, S. M., Atlas, E. L., Flocke, F., Lueb, R. A., Stroud, V., and Travnicek, W.: Measurement of bromine-containing organic compounds at the tropical tropopause, Geophys. Res. Lett., 25, 317–320, 1998.

Schauffler, S., Atlas, E., Blake, D., Flocke, F., Lueb, R., Lee-Taylor, J., Stroud, V., and Travnicek, W.: Distribution of brominated organic compounds in the upper troposphere and lower stratosphere, J. Geophys. Res., 104, 21513–21535, 1999.

Schmidt, J. A., Jacob, D. J., Horowitz, H. M., Hu, L., Sherwen, T., Evans, M. J., Liang, Q., Suleiman, R. M., Oram, D. E., Le Breton, M., Percival, C. J., Wang, S., Dix, B., and Volkamer, R.: Modeling the observed tropospheric BrO background: Importance of multiphase chemistry and implications for ozone, OH, and mercury, J. Geophys. Res.-Atmos., 121, 11819–11835, doi:10.1002/2015JD024229, 2016.

Sala, S., Bönisch, H., Keber, T., Oram, D. E., Mills, G., and Engel, A.: Deriving an atmospheric budget of total organic bromine using airborne in situ measurements from the western Pacific area during SHIVA, Atmos. Chem. Phys., 14, 6903–6923, doi:10.5194/acp-14-6903-2014, 2014.

Stutz, J., Werner, B., Spolaor, M., Scalone, L., Festa, J., Tsai, C., Cheung, R., Colosimo, S. F., Tricoli, U., Raecke, R., Hossaini, R., Chipperfield, M. P., Feng, W., Gao, R.-S., Hintsa, E. J., Elkins, J. W., Moore, F. L., Daube, B., Pittman, J., Wofsy, S., and Pfeilsticker, K.: A new Differential Optical Absorption Spectroscopy instrument to study atmospheric chemistry from a high-altitude unmanned aircraft, Atmos. Meas. Tech., 10, 1017-1042, doi:10.5194/amt-10-1017-2017, 2017.

Van Leuwen P. J., Particle Filtering in Geophysical Systems, Monthly, Weather Review, 137, 4089 - 4114, 2009.

von Hobe, M., Grooß, J.-U., Günther, G., Konopka, P., Gensch, I., Krämer, M., Spelten, N., Afchine, A., Schiller, C., Ulanovsky, A., Sitnikov, N., Shur, G., Yushkov, V., Ravegnani, F., Cairo, F., Roiger, A., Voigt, C., Schlager, H., Weigel, R., Frey, W., Borrmann, S., Müller, R., and Stroh, F.: Evidence for heterogeneous chlorine activation in the tropical UTLS, Atmos. Chem. Phys., 11, 241-256, doi:10.5194/acp-11-241-2011, 2011.

Wang, S., Schmidt, J. A., Baidar, S., Coburn, S., Dix, B., Koenig, T. K., Apel, E., Bowdalo, D., Campos, T. L., Eloranta, E., Evans, M. J., DiGangi, J. P., Zondlo, M. A., Gao, R.-S., Haggerty, J. A., Hall, S. R., Hornbrook, R. S., Jacob, D., Morley, B., Pierce, B., Reeves, M., Romashkin, P., ter Schure, A., and Volkamer, R.: Active and widespread halogen chemistry in the tropical and subtropical free troposphere, P. Natl. Acad. Sci. USA, 112, 9281 - 9286, doi:10.1073/pnas.1505142112, 2015.

WMO: Scientific assessment of ozone depletion: 2014, Global Ozone Research and Monitoring Project, World Meteorological Organisation (WMO), Geneve, Switzerland, 55, 416 pp., 2014.

---

## Author Response (AR3)

Review of: Modelling the Inorganic Bromine Partitioning in the Tropical Tropopause over the Pacific Ocean by Navarro et al.

*The manuscript of Navarro et al., reports on Cam-Chem (Community Atmosphere Model with Chemistry) modelling of the partitioning of inorganic bromine in the tropical tropopause layer(TTL) over the eastern and western Pacific Ocean. The modelling is compared with (averaged) observations of some key species, i.e. of the in-situ measured brominated source gases and O3 from which to the partitioning of inorganic bromine is concluded. Comparisons of measured with and modelling in particular for the yet underexplored TTL are per-se important and interesting. However, based the already-published literature and state knowledge of this field, the paper has major flaw in its present state. My criticism of the present study is based on 5 major deficits (2 more general and 3 more specific comments including one related remark, #3), which are detailed in the following:*

**Response:** We thank Prof. Pfeilsticker for his comments. We really appreciate and welcome any feedback that could be used to improve our manuscript.

*I.) Methodological deficits of the study*

*1. Using (spatial and temporal) averages for fast reacting species (radicals) in photochemical calculations:*

*For the modelling of the bromine partitioning, averaged in-situ O3 (together with bromine released from brominated VLSL) is used. Averaging over space and time of concentrations of photochemical reactive species (Figure 1) however is dangerous, since it may lead to incorrect and spurious results for the inferred quantities (for example concentration ratios). In order to see this please consider the rapidly established steady state of [Br] and [BrO] (both being a function of space x and time t) as well as of some (radical) species (e.g. O3, OH, HO2,..) at daytime, which is established through*

$$J_{BrO}(x,t) \cdot [BrO(x,t)] = k(T) \cdot [O_3(x,t)] \cdot [Br(x,t)] + \dots$$

or  eq. (1)

$$\frac{[BrO(x,t)]}{[Br(x,t)]} = \frac{k(T) \cdot [O_3(x,t)]}{J_{BrO}(x,t)} + \dots$$

*where in the present context irrelevant and missing terms are abbreviated by …. . Evidently in order for the equation to make sense k(T), O3(x,t), JBrO(x,t) need to be local (i.e. measured or calculated) quantities in the photochemical calculations. When using instead space and/or time-averaged quantities (the overbars denote either space or time averaging), the above mentioned equation would instead read as*

$$\overline{J_{BrO}(x,t) \cdot [BrO(x,t)]} = \overline{k(T) \cdot [O_3(x,t)] \cdot [Br(x,t)]} \cdot + \dots$$

$$\frac{\overline{[BrO(x,t)]}}{[Br(x,t)]} = \frac{\overline{k(T) \cdot [O_3(x,t)]}}{J_{BrO}(x,t)} + \overline{\dots}$$

*It can easily be seen, however, that the [BrO]/[Br] ratio calculated from averaged (space or time) quantities and from local quantities generally differ*

$$\frac{\overline{[BrO(x,t)]}}{[Br(x,t)]} = \frac{\overline{k(T) \cdot [O_3(x,t)]}}{J_{BrO}(x,t)} + \overline{\dots}$$

$$\neq \frac{[BrO(x,t)]}{[Br(x,t)]} = \left[ \frac{k(T) \cdot [O_3(x,t)]}{J_{BrO}(x,t)} + \dots \right]$$

*and accordingly only the latter gives the right answer for the photochemically established [BrO]/[Br] ratios in the atmosphere. In conclusion, when using space and time-averaged ozone concentrations (from the manuscript it is not clear as to whether (k(T), JBrO, [Br] and [BrO] were also averaged in the same manner or not, but the answer is somewhat irrelevant to my argument), the modelled [BrO]/[Br] ratio may depart more or less from the actual atmospheric [BrO]/[Br] ratio.*

**Response:** We agree with the reviewer regards the different results eq. (1) and (2) can give depending on the spatial or temporal average applied to either modelled and/or measured data. In doing so, please note the importance of assuming that the "*irrelevant and missing terms*" for the case of the [Br]/[BrO] ratio are negligible compared to the dominant production and loss channels for atomic bromine in the atmosphere. Only if those irrelevant terms can be neglected, then eq. (1) and (2) can be written in its simple ratio form (i.e., dependent only on the photodissociation rate constant $J_{BrO}$ and on the pseudo first-order reaction rate $k(T)*[O_3]$).

Having said this, we would like to make the following points clear:

- Photochemical production of bromine atoms in the Upper Troposphere (UT) is dominated by BrO photolysis ($BrO \xrightarrow{J_{BrO}} Br$), while chemical losses occur mainly through the bi-molecular thermal reaction ($Br + O_3 \xrightarrow{K(T)} BrO$), which itself represent 98% of the total atomic Br losses, (see Saiz-Lopez and Fernandez, GRL, (2016) for details). The much rapid reactivity of these two channels (respect to the neglected terms in eq. (1)) allows the establishment of a rapid pseudo steady-state between Br and BrO. Thus, if we accept that a rapid steady-state is reached between these two species, and also agree that neglected terms are irrelevant for this case, then it is evident that Br and BrO abundances must be related by a mathematical expression which considers only the reaction rates connecting those species.

- We do not mention at all the explicit relation between the [Br]/[BrO] ratio and $J_{BrO}$, $k(T)$ and $O_3$ in this manuscript. We only mentioned the relevance of computing the [Br]/[BrO] ratio in relation to the proposed tropical rings of atomic halogens, whose drivers are

described in a preceding paper (Saiz-Lopez and Fernandez, GRL, 2016). Anyhow, and being aware of the averaging issues mentioned by the reviewer, in that work we performed spatial and temporal averages of Br and BrO abundances, as well as to all rate constants affecting atomic bromine production and losses, and found an excellent correlation between instantaneous (e.g., hourly) and averaged (e.g., monthly) modelled output. Indeed, Fig. 3 in Saiz-Lopez and Fernandez, GRL, (2016), shows the vertical profile of the $[Br]/[BrO]$ ratio as well as the $J_{BrO}/k(T)[O_3]$ ratio obtained with CAM-Chem for an equivalent setup simulation as the one used in this work. The main panels show either annual values for the tropical (20ºN-20ºS) average (Fig.3D) or monthly values within the Tropical Western Pacific (TWP, Fig.3E), while the inset panels show the hourly output modelled linear correlation (daytime masked) between $[Br]/[BrO]$ and $J_{BrO}/k(T)[O_3]$. For the case of the Tropical region, $r^2 = 0.9782$, while for the TWP, $r^2 = 0.99695$, with ratio values spanning approximately from 0 to 3. Equivalent results were also obtained when individual model gridbox (lat,lon,z) were sampled, either hourly or monthly.

- The model output for the present simulations was instantaneous (i.e. hourly, half-hour could have been the highest possible resolution within CAM-Chem). The model was run in Specified Dynamic (SD) mode (i.e., considering the current meteorology prevailing during the campaign) and was further sampled at the correspondent latitude, longitude, height and time "gridbox(lat,lon,z,t)" that best matched the ATTREX flight-track. Thus, nor spatial neither temporal averaging of [Br], [BrO] and its ratio [Br]/[BrO] or other atmospheric quantities in CAM-Chem ($J_{BrO}$, $K(T)$, T, $O_3$, etc.) have been performed to extract the model output. As by the time of preparing this MS there were no other measurements available than ozone, we decided to present the validation of instantaneous $O_3$ measurements for all independent flights (Figure 1, now Fig. 2 in the revised version) and then all atmospheric model variables were averaged into 1 km height bins so the output correspondent to each independent flight could be compared with each other. Being this one a modelling paper (as it is clearly stated in the title), we found appropriate to also perform the spatial-temporal mean of all flight-tracks, which are then used to present a more general representation of the modelled state of the atmosphere in the rather yet unexplored tropical upper troposphere (e.g. the Vertical Profiles shown in Figs. 4, 7 and 8). We are aware that these mean vertical profiles are not descriptive of each of the independent flighst, but they are certainly representative and illustrative of the mean state of the tropical atmosphere within the Eastern and Western Pacific when sun photochemistry is turned on and off.

- For the case of Fig. 5, which shows the temporal evolution (i.e., SZA dependent) of the dominant bromine species and the main inorganic reactants during the day, twilight and night, we decided based on Prof. Pfeilsticker´s comment, to present in addition to the mean temporal profile, the independent results for each specific flight. In this way, the changes in partitioning of the dominant species can be addressed directly in response to the current abundance of ozone, $Cl_y$ and/or $NO_2$ prevailing during each flight track. This helps, for example, to support the large inhomogeneity we suggested in previous studies (see Saiz-Lopez and Fernandez, GRL, (2016) for details) for the [Br]/[BrO] ratio, which is modelled to be larger than 1 at a fixed SZA for one of the flights but not for the others: for this cases, as highlighted by the reviewer, performing the spatial-temporal mean of all flights does not illustrate the intrinsic variability found on the abundance of ozone, bromine and all related short-lived quantities.

Following the Prof. Pfeilsticker´s comments and the above responses, we have modified the text in as follows:

Page 5 line 20:
"*Model hourly output was sampled at exactly the same times and locations as the ATTREX measurements, without performing neither spatial nor temporal averaging on model grids. Once each independent flight track was extracted from the model output, all atmospheric quantities were averaged into 1 km altitude bins, to compare with measured data.*"

Page 7 line 23:
"*Figure 5 compares the mean abundances observed in the EP and WP considering all flights. Even when these results are not descriptive of each of the independent flight, they are representative and illustrative of the mean state of the tropical upper atmosphere within the eastern and western Pacific in the presence and absence of sunlight. Equivalent results but for each independent flight are show in the Supplementary online material*"

Page 9 line 1:
"*A closer inspection on each independent flight (Figs. S1 and S2) reveals the large inhomogeneity of the tropical rings of atomic bromine. In the EP, Br surpass BrO mixing ratios at 60° SZA for flights RF04 and RF06, but as the remaining flights sampled larger BrO mixing ratios, the mean EP abundances shown in Fig. 5c shows Br/BrO > 1 only at 20° SZA. Similarly, the mean results shown in Fig. 5a for the WP show BrO > Br at all times, but RF02 and RF03 show the ratio Br/BrO to be larger than one at 50° SZA. This highlight the importance of considering non-averaged (both spatially and temporal) model output to determine the concentration of photochemical reactive species or other atmospheric quantities such as the Br/BrO ratio.*"

Page 9 line 14:
"*Figure 7 shows the distribution of the Br/BrO ratio over the WP and EP, and its correlation with ozone concentrations and temperatures. The results are based on the mean 1 km binned data for all track flights, although equivalent conclusions can be reached for each independent transect.*"

*Further when inspecting the ozone concentrations measured by the NOAA instrument in the TTL during ATTREX, it can be seen that actual ozone concentrations may vary by up to a factor of 10 (mostly with height, less in the horizontal in the TLL, see Figure 1) and so the [BrO]/[Br] should cover a similar dynamical range (keeping all the other parameters the same, see Figure 8 in Fernandez et al., (2014)), a behaviour not really recovered when using 1 km binned averages for ozone (Figure 1).*

*As consequence, the modelled [BrO]/[Br] may not well represent actual [BrO]/[Br] ratio met in the atmosphere, and as thus may not really provide a meaningful information to reader.*

**Response:** The sensitivity of the [Br] and [BrO] abundances to ozone mixing ratio shown in Fig. 8 of Fernandez et al., (2014), was performed using a box-model constrained with many chemical parameters (not relevant to described here) and also constant temperature (T=190 K). Note that Fig. 9 of the same paper, shows an additional sensitivity of bromine abundances to temperature. As the bi-molecular thermal reaction $Br + O_3 \xrightarrow{K(T)} Br$ decrease with increasing temperature

$(K(T) = 1.60 \times 10^{-11} e^{-780/T})$, we do not expect the modelled [Br]/[BrO] ratio to cover a similar dynamical range than the ozone variations: it would also depend on the temperature change associated to the air parcels considered (K(T) changes a factor ~ 1.5 between 190 and 210 K). Also note that in the current modelling approach, ATTREX ozone measurements has been used to validate CAM-Chem performance (Fig. 1), but individual measurements values have not been used to compute the [Br]/[BrO] ratio. The modelled [Br]/[BrO] ratio shown in Fig. 7A was computed considering CAM-Chem ozone (Fig. 7B) and temperature (Fig. 7C) fields. Modelled ozone abundances change between 100 ppb and 600 ppb and [Br]/[BrO] ratios between 0.35 and 2.0, so the modeled range between maximum and minimum values span approximately a factor of 6.

With regards to the last sentence, in any case, spatial/temporal average of the modelled $[Br]/[BrO]$ ratio may depart more or less from the actual modelled $J_{BrO}/K(T)[O_3]$ ratio, but not such a strong affirmation can be made respect to the atmospheric *[Br]/[BrO]*. There are no means we can compare here modelled *[Br]/[BrO]* with atmospheric *[Br]/[BrO]* ratios as that would have implied the simultaneous atmospheric measurements of atomic Br and BrO. Even when Br atoms (as well as atmospheric $Br_y$) can be inferred from BrO measurements, this procedure also implies including a detailed chemical mechanism for bromine. Thus, any modelled ratio (such as *[Br]/[BrO], [Br]/[Bry]* or the more commonly used *[BrO]/[Bry]* partitioning) can be compared to atmospheric ratios as long as the chemical mechanism considered is appropriate to represent the chemistry of that specific portion of the atmosphere. We are quite confident that bromine chemistry in CAM-Chem is very well represented (as in many other global models) and that all the main chemical reactions reported in the literature are up-to-date in our setup. Thus, we found quite interesting to compute atmospheric ratios between the major species to establish which ones are the dominant species, and in this way, validate them against measurements to properly constrain chemistry-climate models.

*Finally, the ozone measured by the NOAA instrument and plotted in Figure 1 (right panel) appears to be spikier (due to any reason, but this could also be visual illusion) than the same ozone plotted in Figures 3 to 8 (panel c) in Werner et al., (2017) for the Eastern Pacific.*

**Response:** Measured ozone values have been processed as described in Section 2.1. We found quite difficult to compare the Vertical Profiles shown here in Figure 2 (note that Fig. 1 was shifted to Fig. 2 in the revised manuscript) with the temporal timeseries shown for each flight in Werner et al., 2017, Figs. 3-8c. In any case, the spiker representation of $O_3$ measurements presented here might explain the large range between the maximum and minimum [Br]/[BrO] ratios you expected to find.

*2. Averaging (over the space and time domains) concentrations for longer lived species:*

*For some selected measurements (which ones?) 6 O3 averages and corresponding averages of CHBr3, and CH2Br2 (out of in total 745 in-situ samples from the EP according to the information provided in Figures 2 and 3a) are inter-compared with the respective model predicted parameters. Averaging over time (or space) for species of different photochemical lifetimes is somewhat problematic.*

*In order to see this let's consider species of different photochemical lifetimes $\tau i$ ($i = 1, 2, 3 ...$) with a common timescale against atmospheric transport $\tau m$. Here remember that in general photochemical and dynamical time scales for individual air masses are distributed in space and time (e.g., Waugh and Hall, 2002; Waugh, 2009; for TTL distributions of O3 see c.f., Pan et al., 2014). For the moment, however I skip these complications. Then the joint timescale for photochemical processing and transport is given by*

$$\tau_{eff,i} = \left( \frac{1}{\tau_m} + \frac{1}{\tau_i} \right)^{-1}$$

*where for the sake simplicity, it is assumed that both photochemical and dynamical processes lead to exponentially decaying concentrations. With these simplifications in mind, the time averaged concentration is then obtained from*

$$\overline{c(x)} = \frac{\int\limits_{0}^{\infty} c(x,t) \cdot \exp(-\frac{t}{t_{eff,i}}) \cdot dt}{\int\limits_{0}^{\infty} \exp(-\frac{t}{t_{eff,i}}) \cdot dt} = -\frac{1}{t_{eff,i}} \cdot \int\limits_{0}^{\infty} c(x,t) \cdot \exp(-\frac{t}{t_{eff,i}}) \cdot dt$$

*(which is a Laplace transform of c(x,t)). Averaging samples using an appropriate kernel (here exp(- t/teff, i) is of course different from the (geometrical) average taken over individual samples of ck(x,t), i.e.*

$$\overline{c(x,t)} = \frac{1}{m} \cdot \sum\limits_{k=1}^{m} c_k(x,t)$$

*since in the latter calculation any kernel (whether appropriate, or not see below) to calculated averages is discarded.*
*While for photochemical processes an exponential decay is a reasonable assumption, for dynamical processes in the atmosphere, it is certainly not a good assumption due to the turbulent transport (2-D in the stratosphere). Accordingly, the kernel for dynamical averages (often also called probability density functions, or pdf) does not follow an exponential but rather a power law (e.g., Min et al., 1996; Pierrehumbert and Yang, 1993; Minschwaner et al., 1996; Seo and Bowman, 2000; and for the statistics of actual field data of O3, ClO, and others e.g., see Tuck et al., 2003; Tuck, 2008; Pan et al., 2014). As a consequence, the resulting air mass age spectrum (from which the average age can approximately calculated) is then (approximately) represented by Γ-type functions for the concentrations, which again depend on the time and location in the atmosphere (Hall and Plumb, 1994, Waugh and Hall, 2002).*

*In consequence, the comparison of modelled and measured averages (for ozone in Figure 1 and 2, and ozone CHBr3 and CH2Br2 shown in Figure 2) does not really make sense, if the pdfs for the atmospheric and modelled samples are not the same in a statistical sense. To put it into simple terms, when averaging over (limited) samples one has to prove that the sampling from the real atmosphere and from the modelled atmosphere are made from the same statistical distributed event in space-time manifold in order for comparisons to make any sense. So certainly the way that the measured and modelled parameters are averaged deserves much more attention in the manuscript.*

*Finally, noteworthy is that averages over temporally and spatially distributed 'fluctuations' only give the same result for the inferred moments (averages, variance, et cetera) if the system is ergodic, which unfortunately in atmospheric dynamics is mostly not the case. Moreover, the samples need to be huge in order to fulfil one requirement of the central limiting theorem (CLT), that both samples (taken from the atmosphere and the model) converge to the same pdf (given they are the same which needs separately to be proven).*

**Response:** We appreciate the explanation but we believe that the concerns found by the reviewer in this point are not relevant for this study.

*3. Comparing remotely sensed and modelled concentrations:*

*Moreover, the kernels to calculate averages (and used further on in inter-comparison exercises, see below) in remote sensing applications and in inverse modelling are strongly instrument and measurement-dependent (Rodgers, 2000). Fortunately, they often mask the above described effects due to their limited spatial or time resolution, i.e. their inherent averaging. In fact, in the latter applications these 'kernels' are called 'averaging kernels (AK)' of the observation and in colloquial English the averaging kernels can be called the 'glasses' by/through which the remote sensing observations were made. So the characteristics of 'the glasses' need to be considered in some way in inter-comparison exercises with modelled quantities (see below).*

*For some examples of actual AKs, please inspect Figures 5 and 10 (for the weighting of the probed concentrations in the horizontal) in Stutz et al., (2016), Figure 3.5 in Rodgers (2000), Figure 1 (below), or any other study on remote sensing. Chapter 3 in Rodgers, (2000) also discusses the different error sources of the traditional inversion methods used in remote sensing and inverse modelling. It also describes how remotely sensed quantities (here called co(i), where i is the retrieval grid number somehow representing the vertical resolution of the measurement) need to be compared with modelled results (cm(i)), i.e. by comparing the inferred co(i) with the product AK· cm(i), where AK is a tensor, of which the columns (or rows) a filled with the individual averaging kernels, displayed for example in Figure 1.*

[Figure]

Figure 1: Calculated Averaging Kernels (AK) to infer BrO profiles (Rodgers, 2000) from limb observations during NASA ATTREX using optimal inversion (Stutz et al., 2016, and Werner et al, 2017).

*In order to avoid these complications using traditional inversion methods for the interpretation of remotely sensed quantities (and in the particular case those arising from multiple scattering due to the a priori unknown spatial distribution and optical properties of aerosol and cloud particles), Stutz et al., (2016) describes a novel (scaling) method for the interpretation data. In effect, the scaling method uses additional information gained from simultaneously in-situ measured gases (i.e. O3) in order to assist the interpretation of remotely sensed NO2, and BrO in the TTL. Therefore, the scaling method has to be considered as a hybrid method (since it uses information collected by remote sensing and in-situ measurement), which comes with some advantages (and disadvantages) over traditional remote sensing methods. For example, it provides a higher accuracy than methods purely relying on remotely sensed information. Evidently the major disadvantage of the scaling method arises from the need of in-situ information of the probed air masses, i.e. it is suitable for applications from satellites, or high flying balloons. Further the scaling method still requires to carefully consider (by RT calculations simulating the observations) in order simulate how the information (the measured absorption) is obtained.*

*Accordingly, when applying the scaling technique to their remotely sensed data, Stutz et al., (2016) and Werner et al., (2017) actually simulated each individual observation by modelling the actual RT (and the predicted absorption of the targeted species) by considering instrumental and other details of the measurements as well as predicted curtains of the targeted species, obtained from CTM modelling (TOMCAT/SLIMCAT). This approach (as in any traditional remote sensing application) thus carries over to the analysis any relevant instrumental and observation-related features in the forward modelling of the observation. Evidently, the scaling method (as any traditional inversion method) then allows very close inter-comparisons of the predicted quantities (e.g. trace gas concentrations) with the observations, including a correct attribution of the fraction of the measured absorption (or slant column) to parts of the atmosphere not directly probed by the observation, however only if the averaging kernels are appropriately considered.*

*Here please also note that the latter approach to inter-compare remotely sensed data and CTM modelling is not new at all, but e.g., it has been used by our group for more than 2 decades. Further using the scaling method, the calculation of absolute concentrations is achieved using a simultaneously in-situ measured and remotely sensed gas (e.g., O3), together with an appropriate consideration (by RT modelling) of the different sensitivities for detection of the targeted and scaling gas (see equation 14 in Stutz et al., 2016). In effect, the accuracy of the inferred quantities is arguably much better (Stutz et al., 2016) than only relying on remotely sensed quantities for the*

*retrieval of concentrations. Accordingly check your statement on page 6 (lines 14 and 15) for correctness.*

**Response:** we thank the valuable information the reviewer gives us about remote sensing techniques and the scaling method, but our manuscripts is not based on them. We are not using the CAM-Chem model to compare with any remote sensing data. The manuscript is based on the discrete measurements taken with *in situs* GWAS, which CAM-Chem reproduced very well.

*II.) Comparison with available measured data*

*Further, I'm really curious why the authors did not attempt to compare their modelling work with actual measured NO2 and BrO data (potentially) available to the first author for more than a year and which now have been published (Werner et al, 2017). However, when using remotely sensed data in inter-comparison exercises, the kernel for horizontal averaging (see Figure 10 in Stutz et al. (2016)) has to be appropriately taken into account for the modelled data (see my remark #3 above). Further, given that the Werner et al., (2017) manuscript (which the first author of the present article co-authored) was submitted earlier (July 17, 2016) than the present manuscript (Nov. 18, 2016), the statement on page 4 (line 18) is not well based. By being more specific, the lack of a tight comparison of the modelled results with existing measured data give rise to some more deficits of the present study:*

*4. Simulated NO2:*
*For the Eastern Pacific TTL, the CAM-Chem model predicts NO2 between 0.7 – 343 ppt at daytime (Table 1 and Figure 6). No reasons are provided for the elevated NO2 in the TTL over the EP, except that modelled air masses are affected by 'pollution'. However, no other indication (neither from, for example, measured CO during NASA-ATTREX (UCATS) nor any further evidence inferred from the model) is provided that in fact polluted air masses were reaching the TTL over the EP in early 2013. In fact, the NO2 mixing ratios reported by Werner et al., (2017) were < 20 ppt in the TTL, and they agree well (within the error bars +-/10 ppt) with the predictions of the TOMCAT/SLIMCAT simulations assuming no contribution from 'pollution'. In all these respects, and in particular with respect to the discussion provided above under point #3, the statement on page 6 (lines 14 and 15) is not well founded.*
*Accordingly, in any further study information has to be provided why for the EP TTL the Cam-Chem model predicts NO2 concentrations much large than observed. In addition, coherent evidences both from observations and modelling has to be provided (for example from CO/O3 and CH4/ O3) that indeed the TTL over the EP is affected by 'pollution'.*

*5. Simulated inorganic bromine, its partitioning and spatial patchiness:*
*A major part of the study is devoted to model the bromine partitioning. First, I found it hard understand why the model does not really reproduce the increase in total inorganic bromine with increasing height (potential temperature) within the TTL, mainly caused by the destruction of brominated VSLS. This is somehow curious since the bromine concentrations at the lower boundary reported by Navarro et al., (2015) (page 3, line 18 and 19; VSLS; 3.84 ± 0.64 and 3.18 ± 1.49 ppt from WP and EP, respectively, and inorganic bromine 3.02 ± 1.90 ppt of Bry over the EP and 1.97 ± 0.21 ppt over WP) are in reasonable agreement with the data for the EP TTL,*

*reported by Werner et al., (2017). Moreover, in the Cam-Chem model inorganic bromine (in gaseous form) barely increase from ~2 ppt (from the lower boundary at 14 km) to ~3 ppt at 18 km (Figures 4 and 8), in stark contrast with the observations presented in the Werner et al., study for the upper levels of the TTL over EP. Here, depending on the flight, inorganic bromine ranges from (2.63 ± 1.04) ppt (range from 0.5 ppt to 5.25 ppt) to 5.1 ±1.57 ppt (at Θ = 390 - 400 K) to 6.74±1.79 ppt (at Θ > 400 K), in agreement with the measured destruction of brominated VSLS species (Navarro et al., 2015, and Figure 14 in Werner et al., 2017). So the obvious question is: Does the model either not efficiently destroy the brominated VSLS, and/or does the missing bromine reside in/on particles? If the latter is the case, the bromine up-taken by particles need to be rather large (2 – 3 ppt) in order close the bromine budget. So some information has to be provided how the bromine budget is closed in the model, and in particular on how much bromine is up-taken by the particles.*

*Next even though the modelled absolute amount of gaseous inorganic bromine likely may not affect the Bry partitioning, the modelled [Br]/[BrO] (cited: (1) .... the modelled Br/BrO maximizes at 17 km from page 7, line 7 to 17 and in Figure 7 and (2) .....that Br/BrO may become as large as 2 in the TTL of the EP, see Figure 7) deviates from expectations based on the amount of ozone and its increase with height (see Figure 1 left panel, and Figure 3 to 8 in Werner et al., 2017), and the modelled bromine partitioning in the TTL as function of ozone (Fernandez et al., 2014, Figure 8). In fact, these findings largely contrast with early findings based on the Br/BrO ratio in TTL (at 17 km) during daytime c.f., by Fernandez et al., 2014 (Figure 1, left panel where Br/BrO < 0.6 at 17 km during tropical noon), Schmidt et al., (2016) (Figure 1), or lately the model results presented in Werner et al., (2017) (inspect Figure 3 – 8, Br/BrO < 0.6 at 17 km). Reasons for this discrepancy, including a discussion how the averaging of the ozone and the source gas concentrations and of other quantities impacts the modelled Br/BrO ratio (see points 1 and 2 above) certainly need to be addressed in any future study.*

*Finally, the model predicts a certain patchiness (on spatial scales of some hundred kilometres) of the modelled Br/BrO ratio at 17 km for the EP (and WP), with [Br]/[BrO] ratios ranging from below < 0.5 to about 2. No further reason for this patchiness is provided in the manuscript. If air masses entrained by mesoscale convection into the TTL are responsible for this patchiness, then it also needs to be seen in other gases (e.g. CO, CH4...), but again no evidence for this is provided in the manuscript. The predicted patchiness also contrasts with measured O3, NO2, and BrO, in particular since the remote sensing measurements can easily resolve horizontal variations of the measured quantities on the hundreds of kilometre scale (e.g., Stutz et al., (2016) figure 9, and Werner et al., (2017), figures 3 to 8). Further, since at daytime a rapid steady state is established between Br and BrO (see above) as function of the solar illumination and O3 concentration, it is difficult to infer from measured data any reason for the predicted patchiness in the [Br]/[BrO] ratio.*

As answered to reviewer 2 (see major comment 1) the manuscript has been modified to emphasize the fact that this model study was performed simultaneously to the study published by Navarro et al., 2015. We prefer not to go into further details about the issue of data availability and, although we really thank Prof. Pfeilsticker for his insightful comments, we would also like to think that the soundness of our modelling paper is not solely based on how well or not we compare to measurements and TOMCAT/SLIMCAT model results in Werner et al., 2017, as it seems to transpire throughout this review. Nevertheless, and when relevant, we have also modified the manuscript to state how our results compare to the work of Wener et al., 2017.

A final point, so far and to the best of our knowledge, the BrO and $NO_2$ measurements from ATTREX 2014 (Western Pacific) are still being reviewed. Werner et al., 2017 report measurements of BrO and $NO_2$ from ATTREX 2013, although that paper was still in discussion by the time of our submission.

*6. Error and uncertainties:*
*Finally, as an experimentalist who devotes 85 % of his efforts in the interpretation of data to get a handle on a reliable (thus justifiable) errors and uncertainties of the measured quantities, I always find it curious if studies lack a proper discussion of errors and uncertainties of the presented results. In modelling studies, this could for example be done by (1) inspecting respective Jacobians of the relevant quantities, (2) investigate differences in the modelled fields from 'on and off' runs, and (c) perform ensemble runs et cetera. So also in this respect, the present study largely lacks this requirement for robust science.*

**Response:** This is a good point - in this work, we are using the model and chemical mechanism employed in the Navarro et al., 2015 paper. This mechanism was already tested, tuned and validated for tropical vertical profiles of speciated $Br_y$ (Fernandez et al., 2014; Ordoñez et al., 2012). The details about the development of the chemical mechanism, all sensitivity tests performed to tune the model along with uncertainties estimation can be found in those previous works.

*Summary*
*Given the above described methodological deficits (points 1 and 2), the lacking comparison of the modelled results with actual measured data (points 4 and 5), and the lacking discussion of errors and uncertainties (point 6), unfortunately it is impossible to recommend the manuscript for publication in the present form.*

**Response:** We have addressed point-by-point the reviewer´s comments relevant to this work and we appreciate those other clarifications about comparing remote sensing data to an atmospheric 3D model, which are out of the scope of this paper.

Review of: Modelling the Inorganic Bromine Partitioning in the Tropical Tropopause over the Pacific Ocean by Navarro et al.

*General Remarks: This paper uses global model stimulations to examine the inorganic bromine (Bry) budget of the TTL, building on the work of Navarro et al. (2015). In that work, the authors (a) presented measured (and modelled) vertical profiles of brominated very short-lived substances (VSLS), such as CHBr3 and CH2Br2, from recent NASA ATTREX flights, and (b) used a model that reproduces the observations well (CAM-Chem), to estimate the contribution of VSLS to Bry in the TTL (highlighting the significance of that contribution).*
*In the present work, the same approach is adopted as above, though the focus is more on understanding the modelled Bry speciation in the TTL, the Bry diurnal cycle, and differences between the West and East Pacific (where the ATTREX missions sampled).*
*The model results from this work show that BrO and Br are the most abundant daytime species, while BrCl and BrNO2 are more important at night. The authors also discuss differences in modelled Bry partitioning between the West and East Pacific, and briefly the sensitivity to heterogeneous processes on ice.*
*Overall, this paper is an interesting case study that provides an (incremental) advance on our understanding of Bry partitioning in the TTL over the Pacific. In the absence of new BrO measurement data being included in the manuscript, this advance is somewhat subtle when viewed alongside the modelling study of Fernandez et al. (2014, ACP) that also used CAM-Chem to look at TTL bromine partitioning, in some detail. I have outlined three major areas below that should be addressed before publication.*

**Response:** We thank the reviewer for the helpful comments and technical corrections. Below we address point-by-point all his/her comments and suggestions.

*Major Comments:*

*1. The authors should ensure that the Introduction clearly sets out which of the broad model findings have come before, in order to help determine what the main motivation and purpose of this paper is. For example, the model results on zones where the Br/BrO ratio is >1 in the UTLS are interesting, though have been discussed previously by Fernandez et al. (2014, ACP) and Saiz-Lopez and Fernandez (2016, GRL). The same can be said about the analysis of the Bry diurnal cycle and Bry speciation in the TTL, and their sensitivity to heterogeneous processes. Is the advance here that this is simply a CAM-Chem case study for the ATTREX campaign period? If so, that is fine, but the measurements of BrO (and NO2) from ATTREX would very much strengthen the paper and help corroborate the modelled fields. In the first paragraph of Results and Discussion, it is noted that "BrO and NO2 measurements from the ATTREX mission were still under examination by the time of this analysis". Is this still the case? It strikes me that it is quite odd that these data are not included her*e.

**Response:** We have amended the text of the introduction to clarify that the main motivation and purpose of this paper is to model the inorganic bromine partitioning derived from the different flights during the ATTREX campaign.
A paragraph has been added to page 2 line 33 and now it reads:

*"Our study manly focuses on the difference in modelled $Br_y$ concentrations in the TTL over the Pacific throughout the ATTREX campaign flight tracks, and examines its temporal and spatial distributions.*

*"Based on the reliable representation of the observed $VSL_{org}$ by the CAM-Chem model on the study of Navarro et al., 2015, and as a follow up of this investigation regarding the chemistry of bromine tracers in the TTL, we estimated the partitioning of $Br_y$ over the tropical eastern and western Pacific during 2013 and 2014, respectively."*

*"From this case study analysis, we also complement the finding of the diurnal Bry speciation in the TTL, and the Br/BrO ratio distribution in the Upper Troposphere-Lower Stratosphere (UTLS) found by Fernandez et al., 2014 and Saiz-Lopez and Fernandez, 2016."*

Regarding to the measurements of BrO (and $NO_2$) from ATRREX, we agree with the reviewer that it would strengthen the paper and help corroborate the model. However, this model study was running simultaneously with the study published by Navarro et al., 2015, when measurements of BrO and $NO_2$ were still under examination, as we stated in Results and Discussion section. We clarify this point by adding the following sentence at the beginning of the Results section (page 5 line 28), which now read:

*"This modelling study was carried out simultaneously with the work published by Navarro et al., 2015. Only ozone and $VSL_{org}$ abundances were available to validate model performance as BrO and $NO_2$ measurements from the ATTREX mission, now published by Werner et al., (2017), were still under examination by the time of this analysis. Thus, once the model performance during ATTREX campaign is evaluated in Sect. 3.1, we step into a CAM-Chem modelling case study oriented to determine the $Br_y$ partitioning (Sect. 3.2) and efficiency of heterogeneous recycling reactions (Sect- 3.3) on the mostly unexplored eastern and western pacific."*

To the day, the BrO and $NO_2$ measurements from ATTREX 2014 (Western Pacific) are still under review. Measurements of BrO and $NO_2$ from ATTREX 2013 were published on the work of Werner et al., 2017, but were not used in our study as the other manuscript was still under discussion by the time of our submission. However, our $NO_2$ and BrO estimations are within the ranges observed by the measurements of Werner et al., manuscript. An additional statement have being added to our manuscript on page 7 line 5 and page 7 line 34 to clarify this information. The text now reads:

*"Our mean vertical distributions for the EP are in the lower edge of the reported ranges of Werner et al. (2017), who reported a measured range for BrO between 0.5 ± 0.5 ppt at the bottom of the TTL and about 5 ppt at $\vartheta$ = 400 K, consistent with an inferred increase of $Br_y$ from a mean of 2.63 ± 1.04 ppt to 5.11 ± 1.57 ppt as we move upward in the TTL."*

*"Our average range of $NO_2$ mixing ratios is approximately 15 ± 6 ppt at 14 km, with slightly higher values over the tropopause, 22 ± 24 ppt at 17 km. These estimates within 1 standard deviation agree with the $NO_2$ values presented by Stutz et al. (2016) and Werner et al. (2017) from observations made during ATTREX 2013 over the EP. As they report in their manuscript,*

*their O₃ scaling technique allowed retrieval of NO₂ concentrations of 15 ± 15 ppt in the TTL, and a range of 70 up to 170 ppt in the mid-latitude lower stratosphere."*

*2. The most novel aspect of this work is the examination of differences between the W and E Pacific. The discussion of chlorine could be improved in this regard. If differences in Cly between the two regions can impact local Bry partitioning, some discussion on how well constrained the actual Cly simulation over the WP (average up to 84 ppt Cly in daylight) and EP (up to 181 ppt Cly in daylight) is needed. At the very least some more details of the chlorine simulation could be given. More broadly, I would suggest that the title of the paper should reflect that the emphasis of the paper is on the differences between W and E Pacific.*

**Response:** We appreciate the reviewer for highlighting this aspect, which has now been strengthened the revised manuscript. Although the sensitivity simulation described in Section 3.3 was introduced to highlight how the different atmospheric conditions between the EP and WP were affecting the bromine partitioning, the original manuscript mainly focused on the changes due to the high/low NOx regime prevailing in each region. The impact of $Cl_y$ chemistry is maximized during the night, as the abundance of reservoir bromine species (i.e. $BrONO_2$, HOBr, HBr) maximizes after down, and in the presence of ice-crystals those species can react with HCl (which is the dominant Cly species throughout the troposphere). Thus, the following heterogeneous reacting sequence is amplified in the presence of large Surface Area Densities (SAD) in the TTL.

$$BrONO_2 \rightarrow HOBr + HNO_3$$
$$HOCl + HBr \rightarrow BrCl + H_2O$$
$$HOBr + HCl \rightarrow BrCl + H_2O$$

In order to highlight the large impact that the heterogeneous recycling occurring on ice-crystal has on the nighttime partitioning, as well as validate the Cly abundance in CAM-Chem, we introduced the following sentences. Please note that additional information regarding chlorine chemistry is also given below in the answers to the general comments.

Page 7 lines 14:
*"It is worth noting that even when the maximum inorganic chlorine levels are larger in the EP, BrCl is not the dominant night-time reservoir, while in the WP, where BrCl dominates, maximum $Cl_y$ mixing ratio is almost half the value found in the EP (see Table 1). Considering all flights, the maximum $Cl_y$ abundances are < 85 pptv in the WP and < 182 for the EP, with a global mean tropical annual $Cl_y$ mixing ratio of 50 pptv in agreement with previous reports (Marcy et al., 2004; Mébarki et al., 2010). This can be explained considering the faster vertical transport occurring in the western pacific region, which decreases the photochemical decomposition of VSL chlorocarbons (Saiz-Lopez and Fernandez, 2016)."*

Page 8 lines 14:
*"Note that the differences in $Cl_y$ abundance can reach factors as much as 5 times larger for the EP if the independent flights are considered (e.g., max. Cly ~500 ppt for RF01, RF03 and RF04 performed in the EP during ATTREX-2013, while max. Cly for all flights except RF07 (< 400 pptv) remain below 100 ppt. Howeverl, the night-time BrCl abundance is larger in the WP, representing more than 90% of the night-time $Br_y$ partitioning for flights RF02 and RF04 (see Figure S1 & S2 in the Supplement). For these cases,*

*BrCl mixing ratios between 1 and 2 pptv are formed within air-parcels with a very low $Cl_y$ abundance (of the order of 10 ppt). In order to understand this unexpected behavior, we performed a sensitivity simulation neglecting the inter-halogen heterogeneous recycling occurring on upper tropospheric ice-crystals, see Sect 3.3 below."*

Page 10 line 5:

*"Heterogeneous recycling reactions of reservoir species on ice-crystals are relevant at UTLS levels, thus, a sensitivity test was carried out to determine the influence of water-ice aerosols on the distribution of the inorganic species. Equations (1) to (6) shows the chlorine, bromine and inter-halogen tropospheric heterogeneous reactions occurring on ice-crystals (for a complete description of the implementation of heterogeneous reactions in CAM-Chem, see Table S1 in supplementary online material of Fernandez et al., 2014).*

$$BrONO_2 \rightarrow HOBr + HNO_3 \qquad (1)$$
$$ClONO_2 \rightarrow HOCl + HNO_3 \qquad (2)$$
$$HOCl + HCl \rightarrow Cl_2 + H_2O \qquad (3)$$
$$HOCl + HBr \rightarrow BrCl + H_2O \qquad (4)$$
$$HOBr + HCl \rightarrow BrCl + H_2O \qquad (5)$$
$$HOBr + HBr \rightarrow Br_2 + H_2O \qquad (6)"$$

Page 10 line 28:

*"Thus, neglecting ice-recycling reactions (1) to (6) suspend the heterogeneous conversion of $BrONO_2$ to BrCl, and gas-phase bromine nitrate (which is formed mainly by the termolecular reaction of $BrO + NO_2 + M$ during twilight) remain as the dominant $Br_y$ species during the night both under a high-NOx regime (i.e., within the EP region) as well as under the low-NOx regime (western pacific). But when the heterogeneous recycling reactions are activated, the model output predicts that the recycling efficiency depends mostly on the total surface area density of ice-crystals in the upper troposphere (SAD-ICE): even under very low $Cl_y$ concentrations (between 10 and 20 ppt), if SAD-ICE is present in the TTL, the night-time partitioning is displaced in favour to BrCl. Fernandez et al., (2014) found tropospheric SAD-ICE levels within the western pacific upper TTL to be the largest of the whole tropical region, suggesting that BrCl abundance should be maximized in this region of the pacific."*

Finally, following the advice of the reviewer we change the title of the manuscript to emphasize the difference between W and E pacific. The title now reads:

"*Modelling the Inorganic Bromine Partitioning in the Tropical Tropopause over the Eastern and Western Pacific Ocean*"

*3. The writing is quite awkward in many places and the paper would benefit from a very thorough check/read through. In addition, although the paper is compact, it would benefit from some sub-headings, particularly in the Results and Discussion section (e.g. Model-measurement O3 comparison, Diurnal cycle in Bry partitioning, or something similar).*

**Response:** Following the advice of the reviewer, we have added some sub-headings to the results and discussion section. The text is now separated in the following sections:

3.1 CAM-Chem model evaluation
3.2 Br$_y$ partitioning
       3.2.1 Tropical ring of atomic Br: indications from this case study
3.3 Heterogeneous reactions: impact of water-ice recycling on Br$_y$ speciation/distribution

*Other points:*
*\*Abstract\* Sometimes the "E" in Eastern (Pacific) and the "W" in Western (Pacific) are capitalized and sometimes they are not. Please be consistent throughout manuscript (including in figures and captions).*

This has been corrected in the manuscript.

*\*Introduction\* P1, L33: "Bry" is defined early on in the manuscript but "inorganic bromine" is used in numerous places after that. I suggest changing the latter to the former where appropriate throughout the manuscript.*

Change has been made, although we kept "inorganic bromine" at a few places to facilitate the reading of congested sentences.

*P3, L2: Struck me that introducing the "proposed tropical ring of atomic bromine" here is odd. Could you not make mention of these papers earlier in the Introduction?*

A statement has been added to the introduction on page 2 lin 27. Now the text reads:
*"This study also introduced the concept of the "tropical ring of atomic bromine", a photochemical phenomenon that extends in the tropics from approximately 15 to 19 km where the abundance of Br atoms is favoured due to low temperatures (<200K) and low O3 abundances(<100 ppb)."*

*P3, L3: "section 3" — "Section 3"*

Change has been made

*\*Methods\* I would separate out Methods into 2.1 Observations and 2.2 Modelling. This section seems quite unstructured in its present form. The Modelling section needs more details as to the ozone precursor emissions that were used in CAM-Chem + some brief information of the chlorine simulation.*

Change has been made. The methods section has been divided in section 2.1 Observations and 2.2 Modelling.  In addition section 2.1 has been separated in 3 other sub-sections to improve the format and structure of the manuscript. Section 2.1 now includes:
       2.1.1 ATTREX campaign, where we have added in the paragraphs of P5, L7 as it was suggested in your last comment.

2.1.2 VSL$_{org}$ Observations, which briefly described the GWAS methodology explained on our previous publication: Navarro et. al., 2015.

2.1.3 O$_3$ Observations, which described the O$_3$ methodology.

In addition, we extended the description of the ozone precursor inventory and the bromine and chlorine emissions as follow (page 5 line 9):

*"The current setup is based on the bromo- carbon emission inventory of Ordoñez et al., (2012), which includes time-dependent geographically distributed sources of CHBr$_3$, CH2Br$_2$, CH$_2$BrCl, CHBr$_2$Cl, CHBrCl$_2$ and CH$_2$IBr. Even when we do not consider here other chloro-carbon sources like CH$_2$Cl$_2$ and C$_2$Cl$_4$, those species live long enough to be injected almost entirely as source gases to the stratosphere and do not contribute to the tropospheric inorganic chlorine (Cl$_y$) loading [Hossaini et al., 2015]. Additional Br$_y$ and Cl$_y$ sources from sea-salt heterogeneous dehalogenation in the lower troposphere are parameterized (Ordoñez et al., 2012; Fernandez et al, 2014). Prescribed surface volume mixing ratios of long-lived chlorofluorocarbons (CFCs) and halons, as well as surface concentration of anthropogenic CO2, CH4, N2O and other ozone precursors are based on the long-lived inventory of Meinshausen et al. (2011)."*

*P3, L12: Has "very short-lived organic substances" not already been defined as VSLorg? Also, as the focus of this work is on VSLS, it would be better if some of the gases listed (e.g. CHBr3 etc.) are actually referred to earlier in the Introduction (maybe around line 31).*

Changes have been made, and now page 1 line 31 reads:

*"Many of these discuss the contribution of brominated very short-lived organic substances (VSL$_{org}$) like bromoform (CHBr$_3$), dibromomethane (CH$_2$Br$_2$) and/or bromochlorocarbons such as (CH$_2$BrCl, CHBr$_2$Cl, CHBrCl$_2$, etc.), in addition to long-lived halons and methyl bromide (CH$_3$Br), as an important source of stratospheric bromine."*

*P3, L17: The two sentences beginning "At the tropopause level" seems out of place. Is this not motivation/background for the present study and should it not appear in the Introduction?*

We believe that the phrase "At the tropopause level" may still be correct. As we mentioned before, this study is based on the previous publication of Navarro et. al., 2015 and the model results of the organic and inorganic bromine composition were found simultaneously. The study of Navarro et al., 2015 showed the organic bromine composition at the tropopause level (~17km). To clarify this point, we added a sentence on the introduction page 2 line 34 which now reads:

*"Based on the reliable representation of the observed VSL$_{org}$ by the CAM-Chem model on the study of Navarro et al., 2015, and as a follow up of this investigation regarding the chemistry of bromine tracers in the TTL, we estimated the partitioning of Br$_y$ over the tropical eastern and western Pacific during 2013 and 2014, respectively".*

*P4, L4: CAM-Chem has already been defined.*

Change has been made

Change has been made

Changes have been made

The manuscript of Navarro et al., reports on Cam-Chem (Community Atmosphere Model with Chemistry) modelling of the partitioning of inorganic bromine in the tropical tropopause layer (TTL) for NASA ATTREX deployments of the Global Hawk over the eastern and western Pacific Ocean in 2013 and 2014. From the previous version the manuscript is now somewhat complemented including information on the heterogeneous processing of bromine and chlorine on ice particles. Due to the more vigorous deep convection and hence colder temperatures in the TTL and sustaining a larger occurrence of cirrus ice in the TTL of western as compared to the eastern Pacific, heterogeneous processes could indeed reveal some differences in the photochemistry in both areas. Including details of the heterogeneous processing and the irreversible removal of chlorine and bromine from the TTL by sedimentating particles may cover a novel aspect in the study, and as such could render the present study even more valuable to the readers of ACP.

Unfortunately, reactions to my previous comments (mostly, but not all related to how measurements and model predictions need to be compared in state-of-the-art studies) are mostly missing in the present manuscript due to any reason. In order to make it again clear, a state-of-the-art intercomparison of measurements and model predictions would either involve (a) a comparison of the forward-simulated measurements including their specific features (here briefly called one-to-one inter-comparison), or (b) by using a Bayesian approach a comparison of measured and modelled probability density distributions (and their moments) of the targeted quantities. Both approaches have been widely used in past studies, and it is known by not so doing the information contained in measurements cannot be properly exploited, or is even misinterpreted (see minor comment 2). So it is up to the authors to provide a tighter inter-comparison of the various measurements with the model predictions than in the past and present a version of the manuscript which goes beyond what is already known about bromine in the TTL. Alternatively, the authors may mainly focus on modelling, but then a larger number of sensitivity runs on the relevant processes (c.f. heterogeneous activation and irreversible removal of the halogen and their relevant parameters) need to be presented.

While I tried to review the present manuscript as independently as possible from the previous version, I'm recalling here my major comments however in a less abstract but in a more illustrative way than previously put forward (see below). So once the authors will have decided on the focus of study/manuscript, they may either fully include my comments when it addresses measurement/model inter-comparison, or in case they focus their study on mostly modelling then they may circumvent them, while providing instead a series of sensitivity runs on the relevant parameters for their study.

Beside these more general comments, rather than only providing some major comments as done in my previous review, I did a more comprehensive review by adding a section called 'minor comments' which lists typos, and minor issues to be clarified as well as a section 'list of references' of which the references are worthwhile to be included in any revised version.

**Response:** We would like to thank Prof. Pfeilsticker for the time spent on reevaluating this manuscript, and for the supportive comments that help to improve the text. Here, we address their concerns:

Major comments:

1. Comparison with measured data and 'averaging' (c.f. on page 5, line 3 and 4; These continuous measurements of ozone were averaged to match the location where GWAS samples were collected, and then compared to CAM-Chem outputs.):

By referring to my comments 1 and 2 of the previous review, how was the averaging performed and inferred from what pdfs?

In order to make my comment 2 (and in part 1) more clear my question points to the following:

Since the sampling time of the NOAA-2 polarized O3 photometer is 1 Hz (corresponding to a column of air of about 180 m in length) but the CAM-Chem has a spatial resolution of 1° (longitude) x 1° (latitude) (page 5, line 19) or about 110 km x 110 km, how was the measured O3 averaged, and how were the higher moments of the pdf calculated and how do these quantities compare with corresponding quantities calculated from the CAM-Chem simulation c.f. how were the errors bars in your Figure 2 calculated?

A similar question related to the averaging may arise when comparing the trace gases measured in AWAS air samples (where the sampling time was about 25 s at 14 km and 90 s at 18 km, corresponding to columns of air of being 4.5 km and 16.1 km long, respectively), and the model.

Further since one can reasonably assume (e.g., when assuming a constant pumping rate), that the concentrations measured by the individual in-situ instruments are spatial averages of the probed air columns, it is not so clear how these statistical measures calculated from observed quantities compare to those (c.f. the average and the variance) calculated from modelled data? Noteworthy is here that the inferred mean/variances are obviously different in the measured and modelled ozone (see your Figure 2), while any respective information on the variances of modelled and measured brominated source gases is still missing in Figure 3.

Remark: The question related to the appropriate 'averaging' of geophysical parameters is largely justified since the concentrations of stratospheric trace gases and tracers may show long-range correlations which are likely different in the measurements than portrayed in the model (for details see Van Leuwen, 2009, and the references provided in my previous review). Moreover, quantities having long-range correlations (as they occur in geophysical fluids), the central limit theorem teaches (and mandates) that the inferred statistical quantities are scale-dependent and that accordingly the quantities (here concentrations) are in general not Gaussian distributed (for the refs see my previous review). Accordingly, the calculation of the relevant statistical quantities requires the knowledge of the underlying (non-Gaussian) pdf, which when known could alternatively to a one-to-one inter-comparison (as c.f., done in Werner et al., 2017) be used in a Bayesian approach in the inter-comparison (e.g., Van Leuwen, 2009, and the references provided earlier).

So some explanation (beyond the word 'average') is needed how the discussed quantities (ozone and the tracers) are inter-compared e.g., and how the mean and uncertainty/error bars in Figures 2 and 3 were calculated. (Comment: Here you may also anticipate the problem of inter-comparing remotely sensed quantities with modelled quantities since the former are usually not uniformly sampled over an air column like samples provided by in-situ measurements but via an observation dependent averaging kernel, see my remark/comment I. 3 in the previous review and in consequence the minor comment 2 below).

Finally, because the underlying statistic of the involved concentrations is non-Gaussian, care has to be taken that derivatives of inferred quantities, c.f. ratios as used to calculate the [Br]/[BrO] ratio, are not biased (see my comment 1, and comment II. 5, second paragraph in my previous review).

**Response:**

As we emphasized before, this model study was performed simultaneously to the study published by Navarro et al., 2015. Thus, average of measurements and model is the appropriate procedure in order to be consistent with the previous methodology. The methodology is supported by

1- The average of continues measurements of ozone over the whole air sampling times based on previous studies that employed a Whole Air Sampler (e.g. Blake et al., 1997;Blake et al., 2001;Blake et al., 2003;Blake et al., 1999;Blake et al., 2004;Schroeder et al., 2014)

2- The study of Olson et al., 2012, which showed measurement-model inter-comparisons of binned data in altitude profiles used to characterize continues measurements of acetaldehyde (taken with TOGA and PTR-MS instruments) that were interpolated to match the whole air sampling times regardless the differences of the WAS filling time (8 s at 150 m and 90 s at 12 km).  Similarly, the study of Kormann et al., 2003 that showed the observation-modeled inter-comparison of averages in the altitude profile of formaldehyde in the marine boundary layer of the Mediterranean Sea.

3- An analysis of annual ozone climatology and ozone probability distribution functions (PDFs) on 12 different regions around the world using NCAR CAM-Chemo model. The climatology evaluate the model performance regarding ozone averages, seasonality, inter-annual variability and the shape of ozone distribution. For the region of Japan for example, model and observations showed good agreement in PDFs for two stations at 500 hPa, and a good reproduction of a bimodal distribution of one of the stations at 800 hPa (Tilmes et al., 2012).

4- A study (unpublished yet) of CAM-Chem ozone PDFs along all the flight track of CONTRAST campaign (carried out at the same time of ATTREX over WP). The following figures illustrate the reasonable agreement between model and observations, particularly at relative humidity (RH) conditions of 45%-100%.

**CAM-Chem along the GV flight track ±2 grid points ± 1 vertical level (1x1 degree run)**

[Figure]

Figure 1: four figures on top: observations–modelled inter-comparison of ozone concentrations during CONTRAST 2014. Four figures on the bottom: densities inter-comparison.

As the reviewer mentioned the average of geophysical parameters is largely justified for stratospheric trace gases and tracers that show long-range correlations. However, careful considerations have to been taken on the way of treating discrete and continues variables. We agree with the reviewer that the central limit theorem and the Bayesian approach is appropriated for the model-measurements inter-comparison, but these conditions only apply for continues variables, not for the discrete measurements we obtained once ozone concentrations were average over the whole air sampling times. Wilks, 2011 pointed out the different parametric probabilities distribution, where there is a clear definition of the discrete a continuous distribution of atmospheric variables. Based on this information, the statistical approach used for our discrete distributions are reduced to find the expected values E(x):

$$\mu = E(x) = \sum xp(x)$$

Where $\mu$= mean, x= discrete variable, and p(x)= corresponding probability of the discrete variable.

Which by definition, it is equivalent to:

$$\mu = \frac{1}{n} \sum_{i=1}^{n} xi$$

Similarly, for discrete random variables the standard deviation was calculated by summing the product of the square of the difference between the value of the random variable and the expected value, and the associated probability of the value of the random variable, taken over all of the values of the random variable, and finally taking the square root.

$$\sigma = \sqrt{(x - \mu)^2 p(x)}$$

Which equivalent formula is,

$$\sigma = \sqrt{E(x^2) - [E(x)]^2}$$

Since modelled and measurements were later on binned into 1km of altitude (as we follow the methodology of Navarro et al., 2015) the standard deviations showed in figure 2 represent the $\pm 1\sigma$ variation from the binned values.

Regarding to the errors in figure 3, we omitted the illustration of the $\pm 1\sigma$ variation from the binned values to avoid cumbersome plots. Instead, errors of slope and interception of the lineal regression are presented in each plot as the goal of the figure is to show the correlation derived from these regressions.

We took into consideration this comment from the reviewer. However, statistical approaches are out of the scope of this manuscript. We modified the statement on page 6 line 7 to clarify the methodological approach of our discrete measurements. The text now reads:

*"The current modelling study was conducted as part of the work described by Navarro et al., (2015), and it follows the same methodology and statistical analysis for discrete variables."*

Also appropriate references were added to the methodology of $O_3$ observations in page 5, line 4-8. Now the text reads:

*"To merge the measurements taken over different time scales, these high-rate measurements of ozone were averaged to match the sample collection times of each GWAS sample (~30 – 90 sec) (Blake et al., 1997;Blake et al., 2001;Blake et al., 2003;Blake et al., 1999;Blake et al., 2004;Schroeder et al., 2014), and then the merged data were compared to CAM-Chem outputs (Kormann et al., 2003;Olson et al., 2012)."*

Similarly, we emphasized the meaning of the error bars of figure 2, by adding a sentences to page 5 line 32 to page 6 line 2. Now the text reads:

*"Once each independent flight track was extracted from the model output, all atmospheric quantities were averaged into 1 km altitude bins, standard deviations were calculated, and the model was compared with measured data."*

And we added another sentences to page 6, line 24-26 to point out the errors of the lineal regression of figure 3. Now the text reads:

*"Figure 3 shows the correlation between average measurements and model outputs of the 1 km of altitude bins for $O_3$, $CHBr_3$ and $CH_2Br_2$ over the WP and EP, as well as the linear regression equations with and associated uncertainties in slopes and intercepts."*

2. Model constraints and boundary conditions: Further if the manuscript attempts to provide novel insights into model's predictive skills on the budget and photochemistry of the considered species within the TTL, more information is needed than provided on how the model is constrained and on how the boundary conditions (for the relevant gases) are chosen.
Being more specific, it is unclear how Cly is constrained (or calculated) in the model due to the provided contradictory information. For example, on page 7, line 16 to 18, it is said that 'Considering all flights, the maximum Cly abundances are < 85 pptv in the WP and < 182 (????) for the EP, with a global mean tropical annual Cly mixing ratio of 50 pptv in agreement with previous reports (Marcy et al., 2004; Mébarki et al., 2010).' while in Table 1 a range of Cly (day) 1 - 515 pptv (WP) and 1 - 969 pptv (EP) is mentioned. Further, in order for reader to judge the predictive skills of the model, the results need to be compared with previous Cly observations (and eventually modelling) in the TTL beyond the Marcy et al., 2004 study actually mentioned in the manuscript, e.g., Marcy et al., 2007; Mébarki et al., 2010; von Hobe et al., 2011; Jurkat et al., 2014, and many others.
Here, similar than previously asked for (and again asked for here) the budget and partitioning of bromine (comment II., #5, first paragraph in the previous review), it would be worthwhile to provide information on the altitude dependent partitioning of Cly (including chlorine the condensed phase) and how the budget is closed with respect to the total organic chlorine.
Next, since you state (c.f., page 7, line 34; 'The scenario over the EP is slightly different (from the WP) as levels of NO2 and O3 define a high NOx regime'), information on the implemented sources for NOx in the model is completely missing. Providing such information was already asked for in my previous review (comment II. 4.), primarily since measured and as well as the modelled NO2 in Werner et al. (2017) does not support the predicted 'high NOx regimes for the EP. In this context, your statement on Page 8, line 7 is not appropriate in this context (see my

minor comment 2, and my remark 3 in the previous review). In consequence you have to face the findings of Werner et al., (2017) regarding NO2, i.e. measured and SLIMCAT modelled NO2 does not support the Cam-Chem prediction of a 'high NOx regime' in the TTL of the EP.

**Response:** We appreciate the reviewer for highlighting some inconsistencies in the manuscript and for pointing out model descriptions or configurations that were unintentionally omitted and may shed light respect to how relevant processes were implemented in our study.

a) $Cl_y$ is calculated explicitly in the model and solved at each timestep ($Cl_y = Cl + 2 \times Cl_2 + ClO + OClO + 2 \times Cl_2O_2 + HCl + HOCl + ClONO_2 + ClNO_2 + BrCl$). Source constraints are based on the CFCs and HCFCs surface mixing ratios following (Meinshausen et al., 2011) as well as on emission of VSL chlorinated species included in the model ($CH_2BrCl$, $CHBr_2Cl$, $CHBrCl_2$). Accordingly we added the following lines to Section 2.2 (page 5, line 23-24):

*"It is worth noting that all inorganic halogen species (i.e., $Cl_y$ and $Br_y$) are not constrained but explicitly solved at each timestep."*

Following the Editor suggestion, we have now moved the additional Figures (required by Prof. Pfeilsticker during previous round of review) from the supplement into the main text (new Figures 5 and 6), and relate them to the original Figure 5 (now Figure 7) (page 7, line 33 to page 8 line 6). Thus, now it is possible to observe the inferred $Br_y$ and $Cl_y$ mixing ratios for each independent flight, as well as the campaign average for both WP and EP. To avoid the contradictory information presented in previous draft, and to relate our findings to previously published results regarding inorganic chlorine in the UTLS, we have modified the text as follows (page 7, lines 21-26):

*"It is worth noting that even when the maximum inorganic chlorine levels for individual flights are larger in the EP, BrCl is not the dominant night-time reservoir, while in the WP, where BrCl dominates, maximum $Cl_y$ mixing ratio is almost half the concentration predicted for the EP (see Table 1 and Figures 5 and 6). Maximum $Cl_y$ abundances averaged for all flights within each region show $< 85$ pptv in the WP and $< 182$ pptv for the EP (see Fig. 7), with a global mean tropical annual $Cl_y$ mixing ratio of 50 pptv in agreement with previous reports (Marcy et al., 2004; Fernandez et al., 2014;Hobe et al., 2011;Jurkat et al., 2014;Mébarki et al., 2010)."*

b) Regarding the altitude dependent partitioning for $Br_y$ and $Cl_y$, Figure 4 in the main text show the vertical extent of the inorganic bromine partitioning within the WP and EP. The respective organic (i.e. $VSL_{org}$) profiles have already been described in Navarro et al., (2015), so it is not necessary to repeat them here. Finally, as current work focus strictly on bromine and not on chlorine, we are confident that knowledge of the total $Cl_y$ atmospheric levels is enough for the purposes of current work. Indeed, HCl dominates inorganic chlorine partitioning in the region of study (Marcy et al. (2004); Hobe et al., 2011) and is the main species affecting heterogeneous recycling of $Br_y$ reservoirs (Fernandez et al., 2014). Thus, we added the following sentence (page 7, line 28-30):

*"Note that within the TTL, HCl dominates the $Cl_y$ partitioning, with modelled mixing ratios up to 1 order of magnitude larger than those found for HOCl and ClONO_2 (see Fig. 10 in Fernandez*

*et al., 2014). Further knowledge of the complete partitioning between inorganic chlorine species is beyond the scope of this work."*

Please see also the answer to minor comment 5 regarding the role of heterogeneous reactions in controlling the $Br_y$ and $Cl_y$ removal in the TTL.

c) We added information on the NOx sources in the Methodology section (page 5, lines 21-23):

*"Global emissions of important ozone precursors (NOx, CO, VOCs) were obtained through a harmonization exercise of reactive emissions between years 2000 and 2005 for different RCP (Representative Concentration Pathways) scenarios (Meinshausen et al., 2011, Lamarque et al., 2011)."*

Even though is not necessary to include the following information in the manuscript, it is worth noting here that these studies showed consistent impacts on tropospheric ozone and determined that the different scenarios for methane mixing ratios will strongly impact tropospheric ozone, much more so than the difference in NOx emissions between RCPs.

Regarding the distinction between a High-$NO_x$ and a Low-$NO_x$ regime, we agree with the reviewer that our previous comparison between EP and WP $NO_x$ conditions was erroneous and misleading. $NO_x$ levels in the EP are somewhat higher than in the WP, but they increase at most by a factor of 2, and always remain below 50 ppt, well into a Lox-$NO_x$ regime. Thus, even the different $NO_x$ abundances produce a different proportion of $BrONO_2$ partitioning between EP and WP, there is not a mechanistic change from a Low-$NO_x$ regime to a High-$NO_x$ regime: in both regions the Low-$NO_x$ regime prevails. In addition to the small $NO_x$ differences, there are large SAD-ICE changes between the EP and WP which affect the overall reaction rate of heterogeneous reactions (1) to (6) within each region. The largest the SAD-ICE, the faster the conversion from $BrONO_2$ to BrCl, Thus, in the WP where SAD-ICE levels are ~1 order of magnitude larger than in the EP, $BrONO_2$ mixing ratios are smaller and BrCl mixing ratios are larger. We have corrected this misinterpreted feature at several sentences in the manuscript, which now reads:

*"The western-to-eastern differences in the partitioning of inorganic bromine are explained by different abundances of ozone ($O_3$), nitrogen dioxide ($NO_2$), total inorganic chlorine ($Cl_y$) and the efficiency of heterogeneous reactions of bromine reservoirs (mostly $BrONO_2$ and HBr) occurring on ice-crystals."* (page 1, lines 26-28)

*"The scenario over the EP is slightly different as levels of $NO_2$ and $O_3$ are larger, while the Surface Area Density of ice-crystals (SAD-ICE) is reduced (see Section 3.3)."* (page 8, line 12-13)

*"Hence, the EP daytime average concentrations of ozone (up to ~ 300 ppb), $Cl_y$ (max $Cl_y$ ~ 181 ppt) and $NO_2$ (max $NO_2$ ~ 48 ppt) are almost twice as high as those over the WP (Fig 7d), while SAD-ICE levels are up to 1 order of magnitude smaller (see Fernandez et al., 2014). Higher concentrations of ozone were associated with enhanced production of BrO (Fig 7c). Meanwhile, during dark hours, the higher $NO_2$ concentrations and the slower rate of heterogeneous*

*reactions of bromine reservoirs (see Section 3.3), lead to the predominant formation of BrONO$_2$ and the reduction of BrCl levels over the EP (Fig 7c)."* (page 8, line 22-26)

*"Also, Br shows a smooth variation during the day, slightly decreasing its abundance as the SZA increases, while the temporal evolution of BrO is more variable, mostly under the higher NO$_x$ levels prevailing in the EP (Fig. 7)."* (page 9, line 12-14)

*"Thus, neglecting ice-recycling reactions (1) to (6) prevents the heterogeneous conversion of BrONO$_2$ to BrCl, and gas-phase bromine nitrate (which is formed mainly by the termolecular reaction of BrO + NO$_2$ + M during twilight) remain as the dominant Br$_y$ species during the night for both EP and WP regions."* (page 11, line 22-26)

*"The smaller impact of turning off heterogeneous reactions in the EP can be explained considering the less efficient inorganic bromine recycling occurring on the smaller SAD-ICE prevailing in this region."* (page 11, line 29-31)

*"Indeed, Hobe et al., (2011) suggested that the coupling of chlorine and nitrogen compounds in the tropical UTLS may not be completely understood, which would also impact on the bromine burden."* (page 11, line 32-34).

Minor comments:
1. Page 1, line 30 cont.: Your statement on the stratospheric sources of bromine does not include inorganic bromine being transported from the troposphere into the TTL and LS, for which previous experimental studies provided quite some evidences, c.f., Dorf et al., 2008, Laube et al. 2008; Brinckmann et al., 2012; Schmidt et al., 2016; Werner et al., 2017, and others. So please clarify.

**Response:** The original initial sentence of the introduction points out to the general role played by bromine (without separating the organic and inorganic fraction) in the stratosphere, and requires no further clarification. The next paragraphs extend specifically on (first) organic source gases and (later) on product gas sources produced in the troposphere. Thus, is only when reaching page 3, line 1, that we added the following sentence to make explicit mention to the Product Gas contribution:

*"These previous studies, as well as many others (Dorf et al., 2008, Brinckmann et al., 2012, Liang et al., 2010, etc.), highlighted the importance of Br$_y$ product gas transportation from the lower troposphere into the TTL and lower stratosphere as well as its impact on ozone."*

Having said this, we would like to highlight a couple of sentences from the original manuscript (one of them also in the introduction) which clearly shows that the direct Br$_y$ transport from the troposphere into the TTL has been considered and analyzed in this study.

*"For example, the location and timing of emissions, the transport dynamics and dehydration processes in the tropical tropopause layer (TTL) (Liang et al., 2010), and the occurrence of heterogeneous recycling reactions on sea-salt aerosol and ice-crystals (Fernandez et al., 2014)*

*"Additional Br$_y$ and Cl$_y$ sources from sea-salt heterogeneous dehalogenation in the lower troposphere are parameterized (Ordóñez et al., 2012; Fernandez et al., 2014)."* (page 5, line 18-19)

2. Page 8, line 7: However, previous studies have shown large associated uncertainties in TTL NO2 measurements based on remote sensing instruments (Weidner et al., 2005; Butz et al., 2006; Bauer et al., 2012).

This sentence demonstrates that the authors did not appreciate (or even understand) my comment 3 outlined in length in the previous review (see also comment 1 above).

So recall my argument now expressed in more simple terms. First all three cited studies refer to measurements where the sensor was not deployed within the air mass of interest, but NO2 (and some other species) were measured somehow remotely (from a balloon or even satellite). Now in order to see the difference of the previous studies with the study of Werner et al., (2017) just make a drawing to compare the different observation geometries i.e. compare the line of sights for sensors looking (from a great distance) slant through the layer of interest (the former three studies) and those inspecting NO2 (horizontally) along the layer of interest (Werner et al., 2017, and inspect Figure 5 in Stutz et al., (2017). Then compare the different path lengths over which the skylight is absorbed and how they differ in length, since the path length is one decisive parameter in optical absorption measurements which determines the detection sensitivity. For the exercise you can fairly assume that all sensors have the same detection limit (in terms of optical density for NO2) since they all operate at the photon electron shot noise and receive about the same skylight radiance.

Second, since the bulk of stratospheric NOx is located somewhere at around 30 km, sensors (satellites and balloons) which attempt to measure the comparable lower amounts of NO2 located below (or behind) this NO2 layer, i.e. NO2 in the UT/LS and TTL, are more affected by any NO2 changes within this high-located NO2 layer than a comparable sensors measuring below and horizontally along the layer of interest (c.f. on the Global Hawk). So as argued in my previous review, in remote sensing you cannot compare a detection limit inferred for a specific measurement with another measurement without considering the individual observation geometries (to which the remote sensors refer to averaging kernels), even when assuming the same instrument is used.

By summarizing there are two observation-geometry related advantages of remote-sensing instruments being deployed within the layer of interest over those inspecting the studied air masses from 'far away'. In consequence, you have to appropriately considered the NO2 (and BrO) measurements, their detection limits and errors explained in length in Stutz et al., (2017), rather by assigning to them uncertainties which suite best to your study.

**Response:** Details of remote sense deployments and functionalities are beyond the scope of this manuscript. However, the text has been modified to point out the reviewer's comments. The text now reads (page 8, line 18-21):

*"However, previous studies have shown large associated uncertainties in NO$_2$ measurements based on remote sensing instruments, which also depends on the individual observation*

*geometries and instrument operation times (e.g. 30% of total relative error of NO₂ measurements below 25Km (Weidner et al., 2005), and 50 % for satellite measurements from SAGE II bellow 25 km (Bauer et al., 2012))."*

3. Page 8, line 10: These results are in good agreement with the partitioning of Bry found by (Werner et al., 2017) where BrO is the daylight dominant species over EP. In fact, your results and conclusion (page 11, line 14 – 20) are suggesting something else. Accordingly, you need to reconsider this statement.

**Response:** Starting with minor comment 3, and extending through minor comments 4 and 6, we understand the reviewer is worried about our results are suggesting that the tropical rings of atomic bromine is a constant feature of the tropical atmosphere. However, we clearly show in Figures 4 and 5 that within our CAM-Chem modelling studies BrO is usually the dominant species during daytime (not atomic Br), and made it clear in the text since the first draft version (Page 7, lines 16-20).

*"At the tropopause level (~17 km) and integrated over all flights and SZAs, the inorganic partitioning showed ~ 43 % (0.79 ppt) of abundance of BrO during daylight and ~61 % of BrCl (0.94 ppt) during night-time over WP. On the other hand, 48 % (1.43 ppt) of Bry is presented as BrO during daylight and 56 % (1.41 ppt) as BrONO2 at night-time over EP (Fig 4). Atomic bromine is the second most abundant species during the day, with mean daytime values of 0.64 ppt and 0.57 ppt for the WP and EP, respectively.*

The specific statement highlighted by the reviewer (Page 8, line 26-29 of current version) states:

*"These results are in good agreement with the partitioning of Br$_y$ found by Werner et al. (2017) where BrO is the daylight dominant species over EP, and the estimates of Fernandez et al. (2014), which suggested BrO and BrONO₂ as the dominant species in the TTL over the entire tropics during daytime and midnight, respectively."*

and we do not find any contradictory message respect to what has been said before and/or the referenced papers.

Regarding the only sentence in our conclusions that the reviewer mention to be "suggesting something else" (page 11, line 14-20 (old) and page 12, line 10-14 (new)), we have slightly modified it as follows, so there is no space left for the reader to infer in between line the message we are trying to give.

*"Reactive species like atomic Br become the dominant Br$_y$ species in patchy regions of the Eastern and Western Pacific TTL during daylight, following the large inhomogeneity of ozone abundances within these regions strongly influenced by deep convection. The low ozone and cold conditions, in combination with the rapid photochemical equilibrium between BrO and Br, favour Br/BrO > 1 for patchy regions of the TTL and are consistent to previous results about the proposed tropical ring of atomic bromine."*

Additionally, the first line in the Conclusions now states (page 12, line 4-6):

*"Our estimates of the $Br_y$ partitioning in the TTL over the Pacific Ocean showed that mostly BrO and in a lesser extent atomic Br are the dominant species during daytime hours, while BrCl and $BrONO_2$ are predicted to dominate the TTL $Br_y$ at night-time over the WP and EP."*

4. Page 9, line 29: These results are in good agreement with the statements of Fernandez et al. (2014), which suggested Br/BrO > 1 during strong convective periods over the WP warm pool region. This finding is not surprising given that the same model (with similar/same model parameters and inputs) were used. So the statement does not add new evidences on Br/BrO > 1 in the TTL etcetera.

**Response:** Even when the halogen chemical mechanisms used in Fernandez et al. (2014) and in current work are similar (for both gas phase and heterogeneous processes), we disagree that current study and Fernandez et al., used equivalent model configurations or inputs. Fernandez et al. was configured with 1.9°×2.5° lat-lon spatial resolution and 26 vertical levels, considered only monthly and annual output and used meteorological fields from a free running (FR) simulation representative of the 2000[th] decade. In the present study, CAM-Chem was executed on the specified dynamics (SD) mode using the meteorological fields prevailing at the time of the campaigns (years 2013 and 2014) with a spatial resolution of 1° (longitude) × 1° (latitude) and 56 vertical levels (from the surface to ~ 3.5 hPa). In addition, the model provided hourly output that was sampled at exactly the same times and locations as the ATTREX measurements. These features, explained in the methodology of the model, add the originality to this work, thus we consider this a major outcome as we were able to confirm the previous results of Fernandez's study. See also answer to minor comments 3 and 6.

5. Page 10, lines 20 to 22: … the absence of ice-crystal reactions increases the total inorganic fraction by 7% and 12% over the EP during the day and night, respectively. …. I see no specific reason why ice-crystals should change the amount of Bry (or Cly), except that Bry (or Cly) is heterogeneously removed from the gas phase, which (if the case) you should then mention and quantify in the manuscript. Again I emphasize (see comment 5, first paragraph) to show in a separate figure how (a) Bry (and Cly) is partitioned among all (organic and inorganic) gaseous bromine (and chlorinated) species (b) the fractions being up-taken by particle, and (c) the fraction being permanently removed by sedimentation (comment II.) #5, first paragraph in the previous review).

**Response:** There is not any non-stoichiometric heterogeneous reaction removing directly HBr from the gas phase in our model setup. To make it clear how heterogeneous reactions can affect the washout of bromine in the TTL, we added the following sentence in Section 3.3 (page 11, line 1-5):

*"Including heterogeneous reactions in the chemical mechanism changes the relative partitioning between $Br_y$ species, and consequently, the abundance of the dominants species that controls effective removal are altered. Thus, turning on and off heterogeneous reactions will change the bromine sinks within the UT and TTL, as the relative efficiency of effective washout for each independent $Br_y$ species are different (i.e., individual Henry's Laws constant are considered for each species)."*

We also added the following sentence in the methodology section to describe the washout parameterization implemented in CAM-Chem (page 5 line 24-28).

*"Losses in CAM-Chem are parameterized following a large-scale precipitation scavenging algorithm that includes a physical treatment of scavenging through improvements in the formulation of the removal in sub-grid-scale cloudy environments, and includes washout as well as ice phase uptake of soluble inorganic bromine species (each of them with an independent Henry's Law constant) within the water column (see Neu and Prather, 2012 and Fernandez et a., (2014) for details)."*

From the three independent points emphasized by the reviewer, item a) requested Bry (and Cly) inorganic (and organic) vertical profile information. The answer has been already given in the response to mayor comment 2 above. Regarding items b) and c), including additional figures showing the bromine fraction being uptake by water and/or ice particles, as well as the fraction being removed by sedimentation, is not the focus of current work. Expanding on these issues here will enlarge enormously the manuscript and change the main focus of this paper, which is to escort the Navarro et al., (2015) study by addressing the inorganic bromine abundance consistent with the previous reported measurements. We agree with the reviewer that heterogeneous uptake and washout is a very important issue, and as such it will be address in another forthcoming paper specifically focused on comparing the different type of halogen removal parameterizations. Here, we have just expanded the description of washout processes of the dominant species, and computed the change in the total inorganic bromine burden for the full scheme compared to the sensitivity case that neglected heterogeneous reactions. The original text (page 10, lines 20-22) now reads: (page 11, lines 6-19).

*"Relative to the results from the complete mechanism (Fig. 4), at 17 km, the absence of ice-crystal reactions increases the total inorganic fraction by 7% and 12% over the EP during the day and night, respectively. On the other hand, $Br_y$ increases by 29% and 40% over the WP during day and night, respectively. This relative increase of total $Br_y$ is mainly due to changes in the amount of HBr during the day and an enhancement of both HBr and $BrONO_2$ during the night. As BrCl is only produced by equations (4) and (5), it does not accumulate during night hours within the sensitivity study where heterogeneous reactions have been turned off (Fig.10). The mixing ratios of all other species remain very similar when comparing Figure 4 and Figure 10. Our model results show that turning off heterogeneous reactions reduces the total amount of $Br_y$ washed out at 17 km by ~0.5 pptv and ~0.3 pptv for the WP and EP, respectively. This value is of the same magnitude but opposite direction to the results obtained by Aschmann et al., (2011). HBr is highly soluble and it would be expected that a relative increase in HBr partitioning would imply a more efficient washout. But, as explained by Aschmann et al. (2011) it is possible that a significant part of the adsorbed HBr at high altitude levels can re-evaporate within the TTL (and eventually reach the stratosphere) before being washed out. This is because the removal process does not occur right away and residence times are large in the TTL. Indeed, they found a local HBr maximum at around 17 km within an equivalent sensitivity simulation that neglected heterogeneous activation for HBr."*

6. Page 11, lines 14 to 20: Reactive species like atomic Br become the dominant Bry species in large regions of the TTL during daylight, following the large variation of ozone abundance within these regions strongly influenced by deep convection……

While I see motives for this statement when air masses are strongly influences by convection in in the TTL of the WP (and much less for the EP due to in general higher ozone there), how does the result relate to results of previous theoretical studies (beyond those of Fernandez et al., 2014)?

**Response:** As we mentioned in previous answers, this study is a follow up of the study carried out in 2015 (Navarro et al., 2015), which focused on the ATTREX campaign, and not the one carried out in 2014 (Fernandez et al., 2014), which first described the tropical ring of atomic bromine. Thus, we've focus on the modelled product gas distributions within the EP and WP consistent with the VSL source gas measurements reported in Navarro et al., instead of providing other model and/or measurement evidences regarding the existence of the tropical rings hypothesis on this work. Certainly, current results are expected to lie in line to our previous studies, but there are also other studies performed with other models and by other authors, that have suggested the prevalence of atomic bromine in the TTL. For Example:

*"This enhancement of Br in the tropical upper troposphere was previously identified by Holmes et al. [2006] for its importance in Hg(0) oxidation and is a consistent feature of models [Fernandez et al., 2014]"*. (Schmidt et al., GRL, 2016. Model GEOS-Chem).

*"Thus, a small amount of BrO observed in the TFT and TTL can reflect the presence of a much higher amount of $Br_y$. Consequently, the full assessment of PGI requires observations of a wider suite of inorganic bromine species in the tropics" … "The ratio of [Br]/[BrO] increased with altitude and is significantly greater than 1 in the TTL, making Br atom the dominant daytime photoreactive $Br_y$ component from ~11 km to the highest altitude we sampled (~15 km)"*. (Chen et al., JGR, 2016. See their Figure 10, panel c.)

In order to make this point clear we have modified the text as follows (Page 9, line 24-26):

*"This enhancement of Br atoms in the tropical tropopause layer has also been identified in other studies and seems to be a consistent feature in global models including a complete treatment of halogen chemistry in the troposphere (Chen et al., 2016;Holmes et al., 2006; Schmidt et al., 2016;)."*

See also answer to minor comments 3 and 4.

7. Page 11, lines 21 to 22: Why the contribution of inorganic bromine directly injected into the TTL is omitted here, e.g. Schmidt et al., 2016, Werner et al., 2017 and others.

**Response:** As we have already clarified in the answer to minor comment 1, our study considers and discusses the direct contribution of $Br_y$ into the TTL. We find not necessary to cite other works here, as we have done it before and because in the lines highlighted by the reviewer we are summarizing only the concluding remarks of our study. Within the referred sentence, when we mention *"their transport into the TTL, as well as the efficiency of heterogeneous reactions*

*involving ice-aerosols"* we are pointing out to the inorganic bromine fraction prevailing in the lower troposphere, which is transported into the TTL. But, as we see the current wording had confused the reviewer, and could also be misunderstanding to other readers, we have modified the sentence to make it clear and explicit (page 12, line 18-22):

*"The variable photodecomposition of VSL$_{org}$, the transport of inorganic degradation products from the lower troposphere into the TTL, as well as the efficiency of heterogeneous reactions involving ice aerosols, play an important role in the overall upper tropospheric Br$_y$ loading and the consequent stratospheric bromine injection."*

8. Page 11, line 25: …. to diminish the uncertainty of the amount of Bry that reaches the stratosphere, and properly constraint the global bromine budget.
In fact, previous studies indicated that the amount of bromine in the stratosphere is less uncertain than how and in what form (i.e. the fraction of organic and inorganic) it is transported into stratosphere. Here you need to cite at least WMO 2014, and if you like to provide an informed list of bromine-related measurements, however only those performed tropical UT/TTL/LS, you need to cite the studies of Schauffler et al., 1993, 1998, and 1999; Dorf et al., 2008; Laube et al., 2008; Brinckmann et al., 2012; Sala et al., 2014; Wang et al., 2015; Werner et al., 2017; Stutz et al., 2017 and others.

**Response:** The lines highlighted by the reviewer is the last sentence of the paper, and is oriented to call the scientific community to keep on going and reinforcing their efforts on measuring the speciation of inorganic bromine species in the TTL. That is why we prefer not to cite previous campaigns at this time (most of which, on the other side, have already been cited in previous sections of the manuscript).

Typos and necessary clarifications:
1. Page 2, line 30: ….. showed approximately 3 to 5 ppt for potential temperatures between 350 and 400 K in the TTL …change to…. showed approximately 3 to 5 ppt Bry for potential temperatures between 350 and 400 K in the TTL
Done
2. Page 2, line 33: Our study manly focuses …change to … Our study mainly focuses
Changed to "focuses mainly" as suggested by a native English speaker author.
3. Page 4, lines 6 and 7: … with 90 custome-made stainless change to …. with 90 custom-made stainless…and again from … a custome inlet at 2 to 8 liters .. to .. a custom inlet at 2 to 8 liters
Done
4. Page 4, line 31: …. the modelling estimates of the organic bromine fractions were similar for the entire Pacific (3.84 ± 0.64 and 3.18 ± 1.49 ppt from WP and EP, respectively). I guess here 'bromine tied in very short lived species' is meant?
The phrase very short-lived species was added to the text.
5. Page 6, line 29: Over the EP, BrONO2 dominates the entire range of altitude from 14 to 18 km.

Since this statement is certainly only correct for the inorganic bromine partitioning at night, the sentence should accordingly read. Over the EP, BrONO2 dominates the entire range of altitude from 14 to 18 km at night.
Done

6. Page 6, line 29: The total Bry burden during daylight hours increases from … change to ..
During daylight hours, Bry increases from….
Done

8. Page 7, line 8: …. of 2.63 ± 1.04 ppt to 5.11 ± 1.57 ppt as we move upward in the TTL …delete … as we move upward in the TTL.
Done

9. Page 7, line 15: is almost half the value found in the EP ... change to ….. is almost half the concentration predicted for the EP (since a model predicts rather than finds (measures) something).
Done

10. Page 7, line 16: ……………are < 85 pptv in the WP and < 182 for … change to … are < 85 pptv in the
WP and < 182 pptv
Done

11. Page 7, line 19 and elsewhere in the manuscript: … in the western pacific region, … change to … in the western Pacific ... since Pacific is a name and hence needs to be written with a capital letter.
Changed to WP to keep consistency along the text.

12. Page 7, line 23: Figure 5 compares the mean abundances observed ….. change to Figure 5 compares the mean abundances of O3 (measured?), NO2 (modelled), and Cly (modelled)… since your study mostly reports on model results.
Since figure 5 and 6 were added to the manuscript, the text in page 7 line 33 to page 8 line 3 now reads: *"Figures 5 and 6 compare the Br$_y$ partitioning as well as the modelled O$_3$, NO$_2$ and Cl$_y$ abundances along all flights in the EP and WP, respectively. Here, it can be clearly observed how the dominant species changes from BrO during daytime, to BrONO$_2$ or BrCl during night-time. Figure 7 shows the mean abundances for all species including all flights in the WP and EP."*

13. Page 7, line 28: As the SZA keeps on increasing, a decrease on photolysis as well as ozone concentrations… change to .... As the SZA is increasing, a decrease in the photolysis as well as ozone concentrations… and delete … as well as ozone concentrations…since there is no reasons why ozone concentrations should decrease with SZA.
Changed to "As the SZA increases, a decrease of photolysis…" as suggested by a native English speaker author.

14. Page 7, line 34: The scenario over the EP is slightly different as levels of NO2 and O3 define a high NOx regime … change to .. The scenario for the EP is slightly different from the WP as concentrations of NO2 and O3 define a high NOx regime there.
Changed to *"The scenario over the EP is slightly different as levels of NO$_2$ and O$_3$ are larger, while the Surface Area Density of ice-crystals (SAD-ICE) is reduced (see Section 3.3)."* as explained in major comment 2.

15. Page 8, line 2, and elsewhere in the text: Stutz et al. (2016) change to … Stutz et al. (2017)
Done

16. Page 8, line 16: Note that the differences …change to … Note that the predicted differences

Done
17. Page 8 line 16: the EP if the independent flights…change to … the EP if individual flights …
Done
18. Page 8, line 28: Note, however, that atomic Br abundances surpass BrO mixing ratios at low SZA (close to noontime) and low ozone abundances (below 100 ppb, Fig. 5b and 5d) …. change to …Note, however, that modelled atomic Br abundances surpass BrO mixing ratios at low SZA (close to noontime) and low ozone abundances (below 100 ppb, Fig. 5b and 5d).
Done, and as the number of the figures changed now it reads: *"However, modelled atomic Br abundances surpass BrO mixing ratios at low SZA (close to noontime) and low ozone abundances (below 100 ppb, Fig. 7b and 7d)."*
19. Page 9, line 2: In the EP, Br surpasses BrO mixing ratios at 60º SZA for flights RF04 and RF06, but…. change to … In the EP, modelled Br surpasses BrO mixing ratios at 60º SZA for flights RF04 and RF06, but
Done
20. Page 9, line 8: …. which focused on ATTREX measurements taken exclusively over the EP and used an O3-scaling technique to retrieve their results, our model calculations support the fact that …change to .. which focused on ATTREX measurements taken over the EP, our model calculations indicate Br/BrO ratios (erase 1. exclusively since SF2 and SF5 lead to central Pacific),
Done
2. erase … used an O3-scaling technique, since it falls of the context here,
We have not modified the sentence as we believe that this is actually one of the main differences between Werner et al., 2017 study and ours.
and 3. our model calculations support .. erase the fact that .. since a model can never produce facts but only more or less good predictions et cetera
Word fact changed to prediction.
21. Page 9, line 16: Over the EP, Br/BrO > 1 are observed as discrete masses, particularly at SZA between 40°and local noon … change to … Over the EP, Br/BrO > 1 are predicted (you did not observe it) in distinct air masses, particularly at SZA between 40°and local noon. …..by the way a result not supported by the results of the Werner et al., 2017 study.
Change made. However, comment regarding to comparison with Werner et al., 2017 manuscript has been omitted here as they have been mentioned just a few lines above (page 9, line 21). Once again, we want to emphasis that our modelling paper is not based on how well or not we compare to measurements and TOMCAT/SLIMCAT model results in Werner et al., 2017.
22. Page 10, line 6: …on the distribution of the inorganic species .. change to .. on the partitioning of the inorganic species
Done
23. Page 10, line 24: As explained by Aschmann et al. (2011) the increment in the amount of HBr at high altitude levels could be due to a slowly sedimentation following by evaporation as the adsorbed HBr is not washed out right away. This sentence (as it is) does not really make sense to me, since if HBr is taken-up by particles it should reduce Bry at the condensing altitudes and upon evaporation (of the particles) should release HBr at lower altitudes (and not vice versa). So accordingly correct the sentence.
We have modified the sentence as follows. For a complete description of HBr washout see also the answer to minor comment 5 above.

*"HBr is highly soluble and it would be expected that a relative increase in HBr partitioning would imply a more efficient washout. But, as explained by Aschmann et al. (2011) it is possible that a significant part of the adsorbed HBr at high altitude levels can re-evaporate within the TTL (and eventually reach the stratosphere) before being washed out."*

25. Page 10, line 33: (SAD-ICE) explain the acronym by words.

SAD-ICE= Surface Area Density of ICE. The acronyms was described before put it between parentheses. We capitalized the initials of the phrase to make it clear.

26. Page 11, line 10: …while BrCl and BrONO2 were found as the night-time dominant species over the WP and EP, respectively. …change to … while BrCl and BrONO2 were predicted to dominate TTL Bry at night-time.

Done

27. Legend Figure 5: 'Average of inorganic bromine species (top panel) and main reactants of the inorganic chemistry (bottom panel) using the entire range of altitudes (14 to 18 km) over the western Pacific (a and b) and eastern Pacific (c and d). Black boxes indicate the percentage of the dominant Bry species for day and night at 17 km.'

First I'm not sure what can be learned from 'averages' for the altitude range 14 – 18 km, when obviously information on the profiles (from the measurements and the model) of the shown quantities is available. Second, I really wonder why these 'averages' somewhat 'oscillate', actually more for the dark than the sun-lit hours. Third, I wonder whether these oscillations are also seen in the measured source gases?

We appreciate the reviewer for addressing the erroneous caption description of old Figure 5, which is now Fig. 7 in the current version. Perhaps it is clearer to interpret now that an equivalent plot for each individual flight has been included in the new main Figures 5 and 6. Each independent panel in Figures 5 and 6 shows the modelled CAM-Chem bromine partitioning sampled at exactly the same times and locations (lat, lon, lev) as the ATTREX measurements. Only results lying between 14 and 18 km are considered. Current Figure 7a and 7b then shows the mean SZA dependence (averaged within ± 5 SZA) for all flights developed in the WP and EP, respectively. We have then modified the text and caption as follows:

*"Figure 5 and 6 compares the $Br_y$ partitioning as well as the modelled $O_3$, $NO_2$ and $Cl_y$ abundances along all flights in the EP and WP, respectively. Here it can be clearly observed how the dominant species changes from BrO during daytime, to $BrONO_2$ or BrCl during nigh-time. Figure 7 shows the mean abundances for all species including all flights in the WP and EP. Even though the mean results do not simulate differences observed in each flight, they are representative and illustrative of the average state of the tropical upper atmosphere within the EP and WP in the presence and absence of sunlight, and should provide relevant information about the dominant processes occurring in each region."* (page 7 line 33 to page 8 line 6)

*"The CAM-Chem output along the ATTREX flights, indicates that Br and BrO alternate as the dominant daytime species, indicating a large inhomogeneity for the tropical ring of Br, mainly due to the large ozone/T variability of the air parcels within the convective tropical WP and EP."* (page 12, line 14-16)

*"Figure 5: Inorganic bromine species and main reactants of the inorganic chemistry sampled at exactly the same times and locations as the ATTREX flights developed over the western Pacific.*

*Each separate panel show SZA dependent results for flights RF01, RF03, RF04, RF05, RF06, RF07 and RF08. Only output sampled between 14 and 18 km is considered.* (page 22, caption)

*"Figure 6: Figure 6: Inorganic bromine species and main reactants of the inorganic chemistry sampled at the same times and locations as the ATTREX flights over the eastern Pacific. Each separate panel show SZA dependent results for flights RF01, RF03, RF04, RF05 and RF06. Only output sampled between 14 and 18 km is considered."* (page 23, caption)

*"Figure 7: Average of inorganic bromine species (top panel) and main reactants of the inorganic chemistry (bottom panel) for all ATTREX flights developed over the western Pacific (a and b) and eastern Pacific (c and d). The output from each flight has been sampled only between 14 and 18 km, and that the average has been performed within ± 5 SZA bins. Black boxes indicate the percentage of the dominant $Br_y$ species for day and night at 17 km."* (page 24, caption)

Additional references:

Brinckmann, S., Engel, A., Bönisch, H., Quack, B., and Atlas, E.: Short-lived brominated hydrocarbons – observations in the source regions and the tropical tropopause layer, Atmos. Chem. Phys., 12, 1213– 1228, doi:10.5194/acp-12-1213-2012, 2012.

Dorf, M., Butz, A., Camy-Peyret, C., Chipperfield, M. P., Kritten, L., and Pfeilsticker, K.: Bromine in the tropical troposphere and stratosphere as derived from balloon-borne BrO observations, Atmos. Chem. Phys., 8, 7265–7271, doi:10.5194/acp-8-7265-2008, 2008.

Jurkat, T., et al. , A quantitative analysis of stratospheric HCl, HNO3, and O3 in the tropopause region near the subtropical jet, Geophys. Res. Lett., 41, 3315–3321, doi:10.1002/2013GL059159, 2014

Laube, J. C., Engel, A., Bönisch, H., Möbius, T., Worton, D. R., Sturges, W. T., Grunow, K., and Schmidt, U.: Contribution of very short-lived organic substances to stratospheric chlorine and bromine in the tropics –a case study, Atmos. Chem. Phys., 8, 7325–7334, doi:10.5194/acp-8-7325-2008, 2008.

Hossaini, R., Chipperfield, M. P., Feng, W., Breider, T. J., Atlas, E., Montzka, S. A., Miller, B. R., Moore, F., and Elkins, J.: The contribution of natural and anthropogenic very short–lived species to stratospheric bromine, Atmos. Chem. Phys., 12, 371 - 380, doi:10.5194/acp-12-371-2012, 2012.

Marcy T. P. et al., Measurements of trace gases in the tropical tropopause layer, Atmospheric Environment 41, 34, 7253–7261, 2007.

Mébarki, Y., Catoire, V., Huret, N., Berthet, G., Robert, C., and Poulet, G.: More evidence for very shortlived substance contribution to stratospheric chlorine inferred from HCl balloon-borne in situ measurements in the tropics, Atmos. Chem. Phys., 10, 397-409, doi:10.5194/acp-10-397-2010, 2010.

Schauffler, S. M., Heidt, L. E., Pollock,W. H., Gilpin, T. M., Vedder, J. F., Solomon, S., Lueb, R. A., and Atlas, E. L.: Measurements of halogenated organic compounds near the tropical tropopause, Geophys. Res. Lett., 20, 2567–2570, doi:10.1029/93GL02840, 1993.

Schauffler, S. M., Atlas, E. L., Flocke, F., Lueb, R. A., Stroud, V., and Travnicek, W.: Measurement of bromine-containing organic compounds at the tropical tropopause, Geophys. Res. Lett., 25, 317–320, 1998.

Schauffler, S., Atlas, E., Blake, D., Flocke, F., Lueb, R., Lee-Taylor, J., Stroud, V., and Travnicek, W.: Distribution of brominated organic compounds in the upper troposphere and lower stratosphere, J.

Geophys. Res., 104, 21513–21535, 1999.

Schmidt, J. A., Jacob, D. J., Horowitz, H. M., Hu, L., Sherwen, T., Evans, M. J., Liang, Q., Suleiman, R. M., Oram, D. E., Le Breton, M., Percival, C. J., Wang, S., Dix, B., and Volkamer, R.: Modeling the observed tropospheric BrO background: Importance of multiphase chemistry and implications for ozone, OH, and mercury, J. Geophys. Res.-Atmos., 121, 11819–11835, doi:10.1002/2015JD024229, 2016.

Sala, S., Bönisch, H., Keber, T., Oram, D. E., Mills, G., and Engel, A.: Deriving an atmospheric budget of total organic bromine using airborne in situ measurements from the western Pacific area during SHIVA, Atmos. Chem. Phys., 14, 6903–6923, doi:10.5194/acp-14-6903-2014, 2014.

Stutz, J., Werner, B., Spolaor, M., Scalone, L., Festa, J., Tsai, C., Cheung, R., Colosimo, S. F., Tricoli, U., Raecke, R., Hossaini, R., Chipperfield, M. P., Feng, W., Gao, R.-S., Hintsa, E. J., Elkins, J. W., Moore, F. L., Daube, B., Pittman, J., Wofsy, S., and Pfeilsticker, K.: A new Differential Optical Absorption Spectroscopy instrument to study atmospheric chemistry from a high-altitude unmanned aircraft, Atmos. Meas. Tech., 10, 1017-1042, doi:10.5194/amt-10-1017-2017, 2017.

Van Leuwen P. J., Particle Filtering in Geophysical Systems, Monthly, Weather Review, 137, 4089 - 4114, 2009.

von Hobe, M., Grooß, J.-U., Günther, G., Konopka, P., Gensch, I., Krämer, M., Spelten, N., Afchine, A., Schiller, C., Ulanovsky, A., Sitnikov, N., Shur, G., Yushkov, V., Ravegnani, F., Cairo, F., Roiger, A., Voigt, C., Schlager, H., Weigel, R., Frey, W., Borrmann, S., Müller, R., and Stroh, F.: Evidence for heterogeneous chlorine activation in the tropical UTLS, Atmos. Chem. Phys., 11, 241-256,

doi:10.5194/acp-11-241-2011, 2011.

Wang, S., Schmidt, J. A., Baidar, S., Coburn, S., Dix, B., Koenig, T. K., Apel, E., Bowdalo, D., Campos, T. L., Eloranta, E., Evans, M. J., DiGangi, J. P., Zondlo, M. A., Gao, R.-S., Haggerty, J. A., Hall, S. R., Hornbrook, R. S., Jacob, D., Morley, B., Pierce, B., Reeves, M., Romashkin, P., ter Schure, A., and Volkamer, R.: Active and widespread halogen chemistry in the tropical and subtropical free troposphere, P. Natl. Acad. Sci. USA, 112, 9281 - 9286, doi:10.1073/pnas.1505142112, 2015.

WMO: Scientific assessment of ozone depletion: 2014, Global Ozone Research and Monitoring Project,World Meteorological Organisation (WMO), Geneve, Switzerland, 55, 416 pp., 2014.

Reference used in this review:

[revised manuscript text omitted]